# DDX5 plays essential transcriptional and post-transcriptional roles in the maintenance and function of spermatogonia

Julien M.D. Legrand [1,2], Ai-Leen Chan[1,2], Hue M. La [1,2], Fernando J. Rossello[1,3], Minna-Liisa Änkö[1,4], Frances V. Fuller-Pace[5] & Robin M. Hobbs [1,2]

Mammalian spermatogenesis is sustained by mitotic germ cells with self-renewal potential known as undifferentiated spermatogonia. Maintenance of undifferentiated spermatogonia and spermatogenesis is dependent on tightly co-ordinated transcriptional and post-transcriptional mechanisms. The RNA helicase DDX5 is expressed by spermatogonia but roles in spermatogenesis are unexplored. Using an inducible knockout mouse model, we characterise an essential role for DDX5 in spermatogonial maintenance and show that *Ddx5* is indispensable for male fertility. We demonstrate that DDX5 regulates appropriate splicing of key genes necessary for spermatogenesis. Moreover, DDX5 regulates expression of cell cycle genes in undifferentiated spermatogonia post-transcriptionally and is required for cell proliferation and survival. DDX5 can also act as a transcriptional co-activator and we demonstrate that DDX5 interacts with PLZF, a transcription factor required for germline maintenance, to co-regulate select target genes. Combined, our data reveal a critical multi-functional role for DDX5 in regulating gene expression programmes and activity of undifferentiated spermatogonia.

[1] Australian Regenerative Medicine Institute, Monash University, Melbourne, VIC 3800, Australia. [2] Development and Stem Cells Program, Monash Biomedicine Discovery Institute and Department of Anatomy and Developmental Biology, Monash University, Melbourne, VIC 3800, Australia. [3] University of Melbourne Centre for Cancer Research, The University of Melbourne, Melbourne, Victoria 3000, Australia. [4] Centre for Reproductive Health and Centre for Cancer Research, Hudson Institute of Medical Research and Department of Molecular and Translational Science, Monash University, Melbourne, VIC 3168, Australia. [5] Jacqui Wood Cancer Centre, Ninewells Hospital and Medical School, University of Dundee, Dundee DD1 9SY, UK. Correspondence and requests for materials should be addressed to R.M.H. (email: robin.hobbs@monash.edu)

The maintenance of adult tissue integrity and function is often dependent upon a population of resident stem cells that self-renews and generates differentiating cells. The balance between stem cell self-renewal and differentiation can be regulated through extrinsic stimuli, including growth factors produced in a supportive niche, as well as cell intrinsic factors such as transcription factors and post-transcriptional gene regulatory mechanisms. The mammalian testis is an example of such a tissue where tightly co-ordinated transcriptional and post-transcriptional mechanisms harmonise to ensure lifelong maintenance of fertility through the continual production of mature spermatozoa[1].

In mammals, spermatozoa are produced through the complex process of spermatogenesis that is maintained by a population of cells with self-renewal potential referred to as undifferentiated spermatogonia. During spermatogenesis, undifferentiated spermatogonia undergo mitotic divisions followed by induction of differentiation that ultimately results in production of meiotic spermatocytes. After two meiotic reduction divisions, round spermatids are formed that undergo significant morphological changes during a process known as spermiogenesis to ultimately produce haploid spermatozoa[1]. The entire process of spermatogenesis occurs within seminiferous tubules of the testis and is supported by Sertoli cells, essential somatic components that produce GDNF, a growth factor critical for spermatogonial maintenance[2]. In homoeostatic testis, only a proportion of undifferentiated spermatogonia function as stem cells, while the remainder act as differentiation-committed progenitors. However, progenitor populations can revert to a stem cell state following germ cell depletion and induction of a regenerative response[3,4]. Self-renewing cells within the steady-state undifferentiated spermatogonial pool are marked by expression of Gfra1, Nanos2, and Id4 while committed progenitors express Ngn3, Sox3, and Rarg[3–8]. Spermatogonial differentiation is marked by induction of the cell surface receptor c-KIT and a series of coordinated mitotic divisions prior to meiosis[2,9]. Progression of germ cells through the differentiation pathway is associated with incomplete cytokinesis following cell division, a conserved feature of germline systems between species[7]. Within the undifferentiated compartment, the most primitive cells generally exist as single cells termed $A_{single}$ ($A_s$) that can divide to produce $A_s$ daughters for self-renewal or pairs ($A_{paired}$ or $A_{pr}$) and subsequently longer chains of cells that reach 16 cells or more ($A_{aligned}$ or $A_{al}$) and are typically differentiation-primed[7] (Fig. 1a). Importantly, in the presence of GDNF and bFGF growth factors, undifferentiated spermatogonia can be cultured long-term in vitro while maintaining stem cell potential[10].

In addition to extrinsic stimuli, the maintenance and function of undifferentiated spermatogonia are dependent on a complex network of cell intrinsic factors that co-ordinately regulate transcription. Amongst these, transcription factors such as PLZF and SALL4 are important for spermatogonial self-renewal and differentiation, respectively[11–14]. PLZF maintains self-renewal capacity by enhancing sensitivity of undifferentiated spermatogonia to GDNF through indirect inhibition of mTORC1 signalling[14–16]. mTORC1 is a key regulator of mRNA translation, as well as cellular metabolism and aberrant activation of this pathway drives differentiation commitment of spermatogonia in vivo[14,17]. SALL4 was demonstrated to modulate PLZF function in undifferentiated spermatogonia and is required for their differentiation[13]. More recently, we have demonstrated that SALL4 is also required in a PLZF-independent manner for long-term maintenance of undifferentiated spermatogonia activity and regeneration following germline depletion[12]. Other important transcriptional regulators of spermatogonial self-renewal and differentiation have been identified such as ETV5[18] and

SOHLH1/2[8], respectively; however, despite the uncovering of numerous factors, much remains to be understood regarding their interplay in this intricate process.

Besides regulation at the transcriptional level, periods of transcriptional quiescence occurring in spermatogenesis during meiotic homologous recombination and spermiogenesis necessitates post-transcriptional regulation of gene expression[1]. Numerous regulators of alternative pre-mRNA splicing, mRNA export and stability, as well as translational control have been shown to be essential for timely gene expression during spermatogenesis[1]. The RNA-binding protein NANOS2 maintains the undifferentiated state of spermatogonia by recruiting differentiation-associated mRNAs to ribonucleoprotein complexes and inhibiting their translation[19]. NANOS2 also directly inhibits mTORC1 through sequestration of the mTOR kinase, thus robustly suppressing translation of mRNAs required for differentiation[19]. More recently, BCAS2 has been shown to be an essential regulator of alternative pre-mRNA splicing in spermatogonia[20]. Loss of BCAS2 in male mice results in infertility as a consequence of aberrant splicing of genes necessary for meiotic initiation[20]. While spermatogonial fate is regulated by both transcriptional and post-transcriptional mechanisms, involvement of specific RNA processing factors in the maintenance and function of undifferentiated spermatogonia is poorly appreciated.

DEAD box helicase 5 (DDX5) is an ATP-dependent RNA helicase originally described as important for the unwinding of RNA secondary structure, but now known to be involved in multiple aspects of RNA processing including splicing, mRNA export, transcript stability, rRNA biogenesis, and microRNA processing[21–25]. DDX5 belongs to the DEAD box protein family, which is characterised by a conserved Asp-Glu-Ala-Asp (DEAD) motif within a helicase core[22]. Although the helicase core is largely conserved amongst DEAD box family members, highly variable N-terminal and C-terminal regions confer additional roles to these proteins including the ability to bind DNA and act as transcriptional co-factors in conjunction with partner transcription factors[22,26]. In particular, DDX5 acts as a transcriptional co-factor for MYOD and RUNX2, key regulators of myogenesis and skeletal development, respectively[27,28]. DDX5 has also been implicated in progression of breast and prostate cancer due to its ability to act as a co-activator of oestrogen and androgen receptor signalling[22,29].

Previous studies have suggested a role for DDX5 in the male germline; DDX5 was found to be specifically expressed in human spermatogonia through validation of single cell RNA-sequencing (RNA-seq), and the RNA-binding protein RBM5 was found to interact with DDX5 in mouse round spermatids and be required for spermatogenesis[30,31]. DDX5 has also been suggested to be a negative regulator of WNT signalling in the GC-1spg germ cell line[32]. Despite its expression within the male germline, a precise role for DDX5 has not been elucidated and its involvement in maintenance of spermatogenesis is unknown. In this study, we identified DDX5 as an interacting partner of PLZF in spermatogonia and found that DDX5 is indispensable for maintenance of spermatogenesis in mice. We demonstrate that DDX5 has a multifaceted role within undifferentiated spermatogonia that includes both transcriptional and post-transcriptional functions. We show that DDX5 is important for multiple RNA processing steps including pre-mRNA splicing of key genes required for spermatogenesis, as well as the regulation of mRNA nuclear export and transcript stability of cell cycle-related genes. In addition, we provide evidence that DDX5 is involved in transcriptional regulation together with PLZF. Our data uncovers an essential role for DDX5 within undifferentiated and differentiating spermatogonia, and characterises DDX5 as a critical regulator of male fertility.

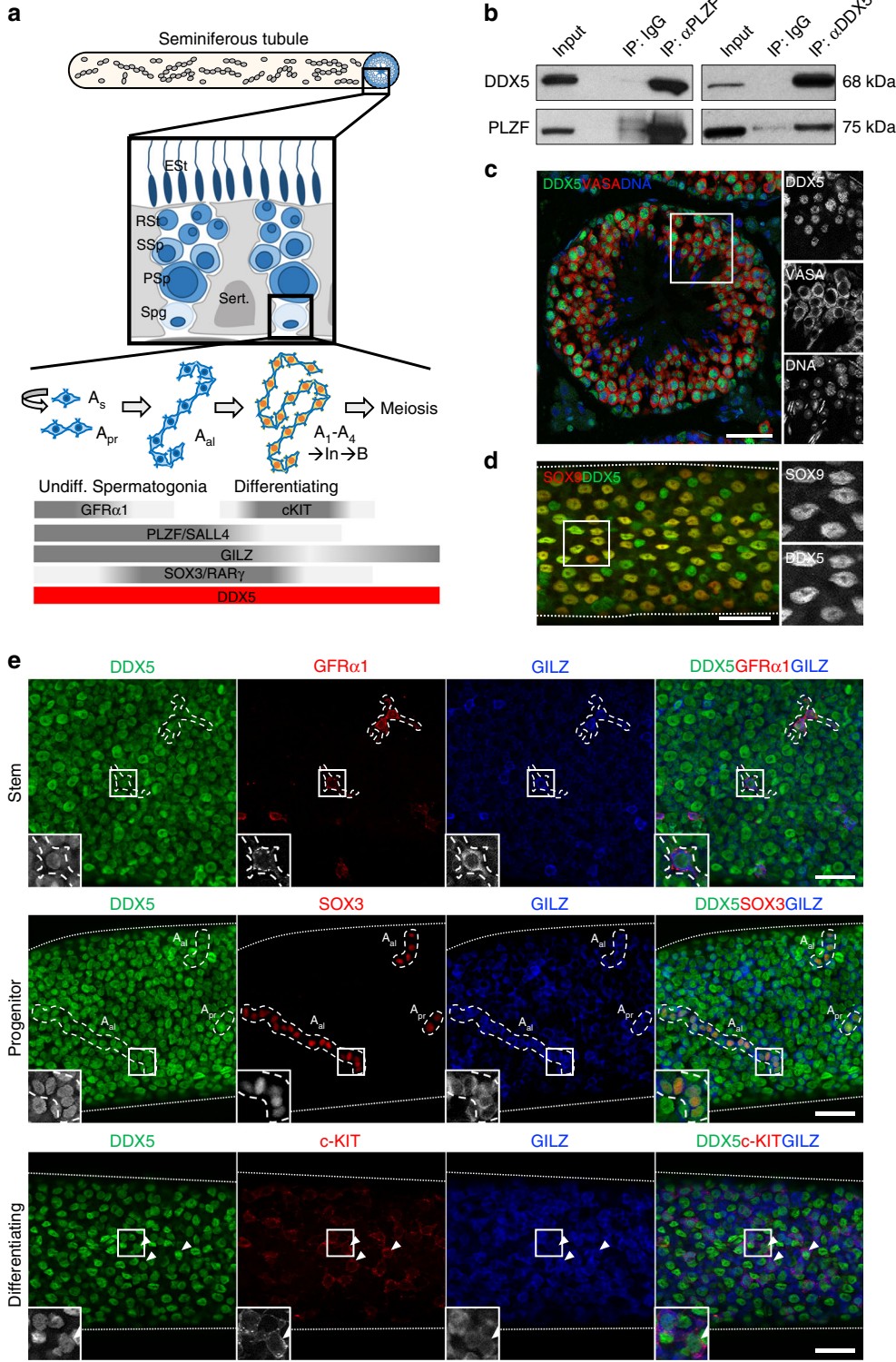

**Fig. 1** DDX5 associates with PLZF and is expressed throughout the male germline. **a** Schematic depicting the mouse seminiferous tubule and populations of undifferentiated spermatogonia ($A_s$, $A_{pr}$ and $A_{al}$), as well as differentiating spermatogonia ($A_1$–$A_4$, In, B) residing on the basement membrane. Commonly used markers expressed within each population are shown below. DDX5 is ubiquitously expressed within spermatogonial populations. Other spermatogenic cells are depicted in a cross-section schematic (large black square). Spg: spermatogonia, PSp: primary spermatocyte, SSp: secondary spermatocyte, RSt: round spermatid, ESt: elongated spermatid, Sert.: Sertoli cells. **b** Reciprocal co-immunoprecipitation followed by western blot showing pull down of DDX5 and PLZF using antibodies against each versus IgG control pull down. **c** Adult mouse seminiferous tubule cross-section showing nuclear expression of DDX5 in all VASA-positive germ cells. **d** Mouse seminiferous tubule whole mount immunofluorescence showing nuclear expression of DDX5 in SOX9-positive Sertoli cells. **e** Adult mouse seminiferous tubule whole mount immunofluorescence showing expression of DDX5 in stem/self-renewing (GFRα1-positive), progenitor (SOX3-positive) and differentiating (c-KIT-positive) spermatogonia populations. GILZ used as a broad marker of spermatogonia. Representative cells are demarcated with dashed lines and white arrowheads and shown in zoom insets. Seminiferous tubule border marked with dotted lines. All scale bars = 50 μm

## Results

**DDX5 is expressed in the testis and associates with PLZF.**
Previous studies have demonstrated that PLZF is important for self-renewal of undifferentiated spermatogonia in mice[14]. To understand further the role of PLZF within the male germline, we performed PLZF immunoprecipitation (IP) on lysates from cultured wildtype murine undifferentiated spermatogonia followed by mass spectrometry (MS) analysis to identify interacting proteins[12]. Including the known PLZF-interacting protein SALL4[13], we identified putative PLZF binding partners (Supplementary Data 1) with DDX5 amongst the top hits defined by presence in each of three independent samples. We confirmed this interaction by reciprocal co-IP and western blot where we observed bands at expected molecular weights when probing for PLZF (75 kDa) and DDX5 (68 kDa) (Fig. 1b). Next, we assessed in vivo expression of DDX5 in mouse seminiferous tubules by immunofluorescence (IF) of testis sections and found nuclear DDX5 expression within all spermatogenic cells (spermatogonia, spermatocytes, spermatids) marked by the germ cell marker VASA[33,34] and distinguished according to their location within the seminiferous tubule[35] (Fig. 1a, c). In addition, we examined expression of DDX5 in non-human primate (marmoset) and human testis sections and observed a comparable expression pattern in spermatogenic cells to that of mouse, including within PLZF-positive cells (Supplementary Fig. 1a, b).

As spermatogonia reside adjacent to the seminiferous tubule basement membrane, whole mount IF can be used to distinguish subsets of spermatogonia through the presence of specific markers plus according to their morphology and topology[7]. Using this method, we examined expression of DDX5 within self-renewing, committed progenitor, and differentiating spermatogonia. First, we found DDX5 expressed within supporting Sertoli cells as shown by co-localisation with the Sertoli cell marker, SOX9[36] (Fig. 1d). Next, we observed DDX5 expression localised to the nuclei of GFRα1-positive (self-renewing, $A_s$ and $A_{pr}$)[3], SOX3-positive (committed progenitor, $A_{al}$)[8], and c-KIT-positive (differentiating)[1,9] spermatogonia, with GILZ used as a broad spermatogonial marker[37] (Fig. 1a, e). We concluded that DDX5 is expressed throughout the male germline and in supporting somatic cells.

**DDX5 is essential for the maintenance of spermatogenesis.** To define the functional role of DDX5 in the adult male germline, we sought to ablate *Ddx5* using a conditional knockout model. Previously, we have used transgenic mice containing a tamoxifen-inducible Cre recombinase under control of the *ubiquitin c* promoter (UBC-Cre$^{ERT2}$)[38] to drive efficient Cre-LoxP-mediated gene recombination in spermatogonia, while meiotic and testis somatic cells remain mostly unaffected[12]. We crossed UBC-Cre$^{ERT2}$ mice with previously described *Ddx5*$^{flox/flox}$ mice[39] to generate a tamoxifen-inducible *Ddx5* knockout line (*Ddx5*$^{flox/flox}$; UBC-Cre$^{ERT2}$). We treated 8-week-old to 12-week-old male *Ddx5*$^{flox/flox}$; UBC-Cre$^{ERT2}$ mice (*Ddx5*$^{TAM-KO}$) and *Ddx5*$^{flox/flox}$ control mice with tamoxifen (TAM) and collected testis samples 5, 7, 14, and 30 days (D) post-treatment. Analysis of testis sections by IF revealed a notable loss of PLZF-positive spermatogonia by D7 following TAM-induced *Ddx5* ablation (Fig. 2a). To verify loss of all PLZF-positive spermatogonial subsets, we stained testis sections for markers of self-renewing (GFRα1), progenitor (SOX3) and differentiating (c-KIT) cells (Supplementary Fig. 2). We did not observe any *Ddx5*-ablated cells remaining in these spermatogonial populations by IF at D7 post-TAM treatment, besides few c-KIT-positive cells adjacent to the basement membrane of seminiferous tubules (Supplementary Fig. 2c). In addition, we quantified *Ddx5* ablation and total cell numbers for

Sertoli cells, spermatocytes, and round spermatids by IF at D7 (Supplementary Fig. 3). We found no significant difference in the number of Sertoli cells, spermatocytes or round spermatids between control and TAM-treated *Ddx5*$^{TAM-KO}$ testis cross-sections. Moreover, we found no significant differences in the proportion of those cells positive for DDX5, suggesting the model used in this study does not result in significant *Ddx5* ablation within testis cells other than spermatogonia (Supplementary Fig. 3). Interestingly, in both control and TAM-treated *Ddx5*$^{TAM-KO}$ testis we noted that DDX5 was downregulated in a subset of SYCP3-positive meiotic cells proximal to the basement membrane, suggesting dynamic regulation of DDX5 during meiosis (Supplementary Fig. 3b). Next, we analysed populations of spermatogonia isolated from whole testis samples up to D14 post-TAM by intracellular staining and flow cytometry according to expression of PLZF and c-KIT[12,13,17]. We observed significant reduction of undifferentiated (PLZF$^+$ c-KIT$^-$), early differentiating (PLZF$^+$ c-KIT$^+$) and late differentiating spermatogonia (PLZF$^{low}$ c-KIT$^+$) in *Ddx5*$^{TAM-KO}$ testes by D5 post-TAM compared with control, with these significant reductions persisting through to D14 (Supplementary Fig. 4a, b).

By D30 post-*Ddx5* ablation, analysis of testis cross-sections by IF revealed seminiferous tubules completely devoid of germ cells as indicated by the absence of VASA-positive cells and a Sertoli cell-only phenotype (Fig. 2a). Whole mount IF of seminiferous tubules at D30 post-*Ddx5* ablation confirmed significant loss of PLZF-positive spermatogonia, with only *Ddx5*-retaining Sertoli cells visible in most TAM-treated *Ddx5*$^{TAM-KO}$ tubules (Fig. 2b). Only tubules where rare *Ddx5*-retaining PLZF-positive spermatogonia were present showed indications of germ cell repopulation (Supplementary Fig. 4c). At D30, TAM-treated *Ddx5*$^{TAM-KO}$ testes weighed significantly less than control testes as anticipated (0.12 ± 0.01% of total body mass versus 0.30 ± 0.01%, mean ± SEM, $n = 18$ control, $n = 20$ TAM, $P < 0.0001$, two-tailed unpaired $t$-test) (Fig. 2c). Quantification of PLZF-positive cells per tubule cross-section demonstrated significant reduction in numbers of spermatogonia at D5, D7, D14, and D30 in TAM-treated *Ddx5*$^{TAM-KO}$ testes compared with controls (Fig. 2d). Besides this testis phenotype, no other overt phenotype was observed in *Ddx5*$^{TAM-KO}$ mice and no significant difference in body mass was discerned at any time point (Fig. 2e). This is of interest given that UBC-Cre$^{ERT2}$ is broadly expressed in adults[38], and TAM treatment of *Ddx5*$^{TAM-KO}$ mice would ablate *Ddx5* in multiple tissues besides the testis. Our data indicate that DDX5 plays critical roles in maintenance of spermatogenesis and its loss results in rapid and profound depletion of adult spermatogonia.

**DDX5 is indispensable for the maintenance of spermatogonia.**
Having demonstrated the requirement of DDX5 in maintenance of spermatogonia in vivo, we sought to explore mechanisms underlying DDX5 function and confirm its cell-autonomous role in the germline using an in vitro system[4,14]. Therefore, we established cultures of undifferentiated spermatogonia from untreated *Ddx5*$^{TAM-KO}$ mice and induced *Ddx5* ablation by treatment with 4-hydroxytamoxifen (TAM)[12]. Cultured *Ddx5*$^{TAM-KO}$ spermatogonia treated with vehicle alone were used as controls. *Ddx5* was efficiently ablated in *Ddx5*$^{TAM-KO}$ spermatogonia following TAM and resulted in pronounced depletion of cultured cells such that only few small colonies of PLZF-positive *Ddx5*-ablated cells remained at D2 post-treatment (Fig. 3a). These results are consistent with a critical cell-autonomous role for DDX5 in spermatogonial maintenance. In contrast, mouse embryonic fibroblasts (MEFs) established from *Ddx5*$^{TAM-KO}$ embryos and treated with TAM persisted in culture and continued to proliferate despite loss of *Ddx5*

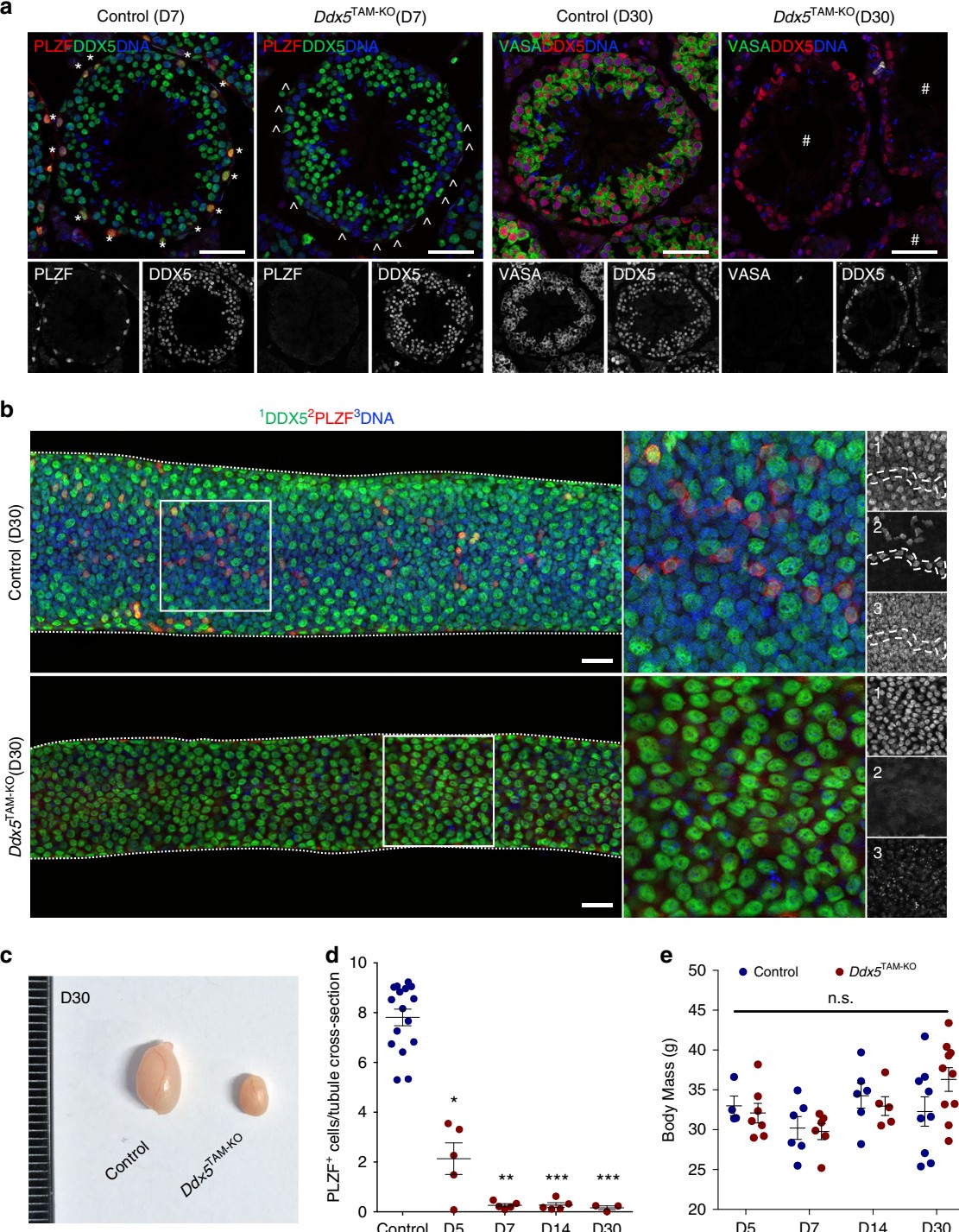

**Fig. 2** DDX5 is essential for the maintenance of spermatogenesis. **a** Adult mouse seminiferous tubule cross-sections showing tamoxifen-induced UBC-Cre-mediated deletion of *Ddx5* (*Ddx5*[TAM-KO]) results in loss of PLZF-positive spermatogonia by 7 days post-tamoxifen (D7) and all VASA-positive spermatogenic cells by 30 days (D30). Only DDX5-retaining Sertoli cells are present at D30 in *Ddx5*[TAM-KO] seminiferous tubules. *: PLZF-positive spermatogonia; ^: Sertoli cells; #: seminiferous tubules devoid of germ cells, with only DDX5-retaining Sertoli cells visible. **b** Adult mouse seminiferous tubule whole mounts showing control and *Ddx5*[TAM-KO] tubules at D30 post-tamoxifen. Chains of PLZF-positive undifferentiated spermatogonia (red) are clearly interspersed amongst Sertoli cells (visible by their characteristic pattern and morphology) in control tubules, whereas no spermatogonia are present upon ablation of *Ddx5*. Zoom insets are shown for control (top middle) and *Ddx5*[TAM-KO] (bottom middle) conditions, as well as single channel images for DDX5 (1), PLZF (2) and DAPI nuclear counterstain (3). All scale bars = 50 μm. **c** Gross testis morphology shows reduction of testis size in *Ddx5*[TAM-KO] mice at D30 post-tamoxifen. **d** Quantification of PLZF-positive cells per seminiferous tubule cross-section upon loss of *Ddx5* at D5, D7, D14, and D30. Control: $n = 16$ mice; D5: $n = 5$ mice; D7: $n = 5$ mice; D14: $n = 5$ mice; D30: $n = 3$ mice; Kruskal–Wallis test with Dunn's multiple comparisons test. *$P < 0.05$; **$P < 0.01$; ***$P < 0.001$ all versus control. Mean ± SEM shown. **e** Comparison of body mass between control and *Ddx5*[TAM-KO] mice at D5, D7, D14, and D30 post-tamoxifen shows no difference between conditions at any time point. D5: $n = 4$ mice control, $n = 7$ mice KO; D7: $n = 6$ mice control, $n = 6$ mice KO; D14: $n = 6$ mice control, $n = 5$ mice KO; D30: $n = 9$ mice control, $n = 10$ mice KO. Mean ± SEM shown. n.s: not significant

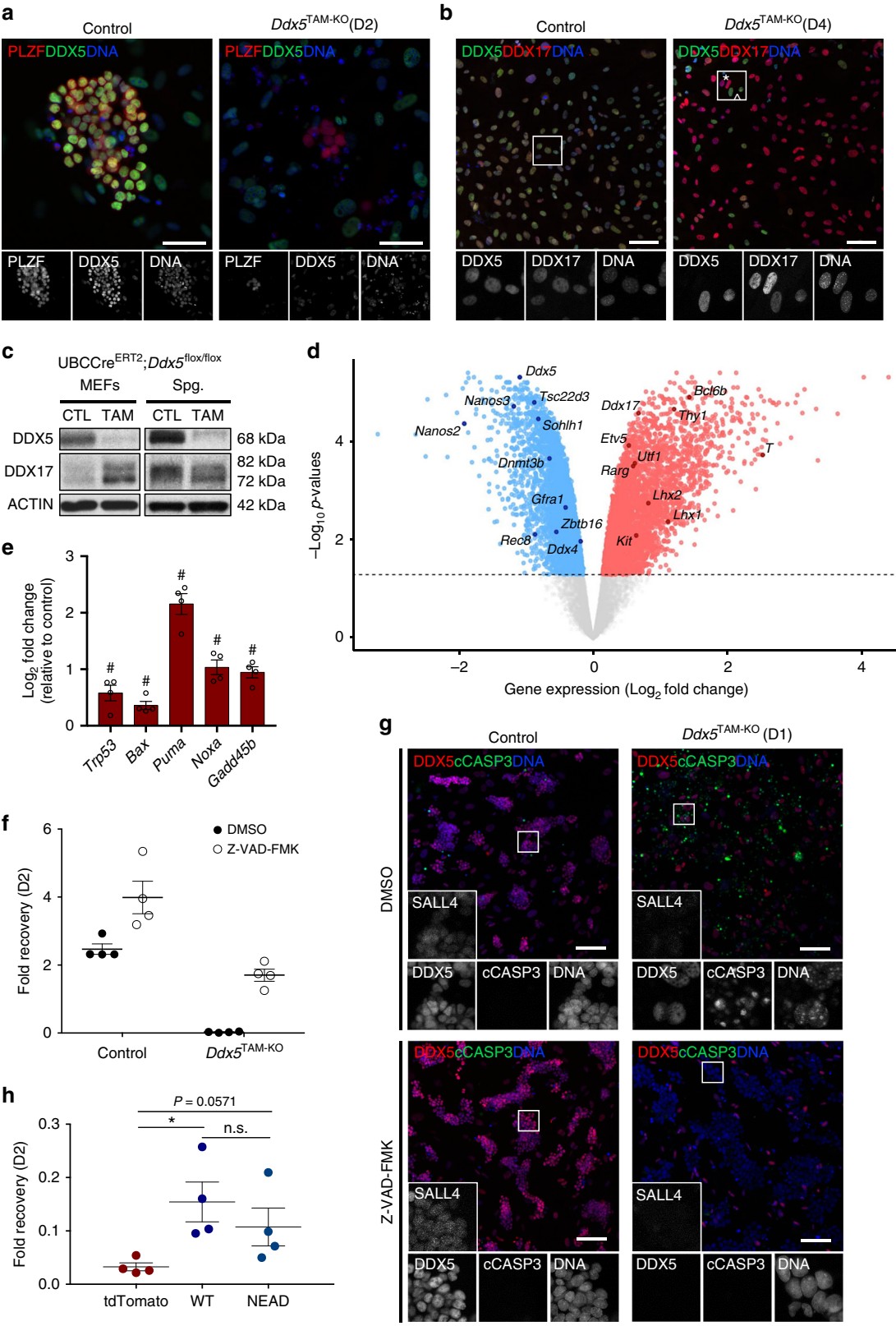

suggesting a specific requirement for DDX5 within spermatogonia (Fig. 3b). It was noted that expression of DDX17, a functionally co-operative paralog of DDX5[26], was upregulated in $Ddx5^{\text{TAM-KO}}$ MEFs following TAM when compared with vehicle-treated MEFs (Fig. 3b, c). However, no corresponding increase in DDX17 was observed in $Ddx5^{\text{TAM-KO}}$ spermatogonia and although the ratio of p72 and p82 isoforms appeared slightly altered upon $Ddx5$ loss, this was not statistically significant (Fig. 3b, c and Supplementary Fig. 5). These data suggest that loss of DDX5 function in MEFs may be compensated for through upregulation of DDX17, whereas its function is indispensable in spermatogonia.

**Fig. 3** DDX5 is required for maintenance of undifferentiated spermatogonia in vitro. **a** Immunofluorescence showing 4OH-tamoxifen-induced UBC-Cre-mediated deletion of *Ddx5* (*Ddx5*[TAM-KO]) results in loss of murine cultured undifferentiated spermatogonia by 2 days post-tamoxifen (D2). PLZF is used as a marker of undifferentiated spermatogonia in culture and DAPI used as a nuclear counterstain (DNA). **b** Immunofluorescence showing loss of *Ddx5* in cultured mouse embryonic fibroblasts (MEFs) (*Ddx5*[TAM-KO]) results in a compensatory upregulation of DDX17 by D4 post-tamoxifen. *: representative cells showing DDX5 loss and elevated DDX17. ^: DDX5-retaining MEFs with comparable levels of DDX17. **c** Representative western blots depicting DDX17 protein levels upon loss of *Ddx5* in 4OH-tamoxifen-treated (TAM) MEFs and spermatogonia (Spg.) compared with vehicle-treated control (CTL) cells in a tamoxifen-inducible cre/lox model (UBC-Cre[ERT2];*Ddx5*[flox/flox]). Blots are representative of *n* = 4 independent experiments. **d** Volcano plot of differentially expressed genes determined by RNA-sequencing analysis of vehicle-treated versus tamoxifen-treated *Ddx5*[TAM-KO] cultured spermatogonia. Dotted line represents false discovery rate (FDR) cut-off <0.05. *n* = 4 biologically independent samples per condition. **e** Differentially expressed genes determined by RNA-sequencing analysis related to p53-mediated apoptosis. #: indicates FDR <0.05. *n* = 4 independent samples per condition. Mean ± SEM shown.
**f** Quantification of cell fold recovery following tamoxifen-induced *Ddx5* ablation (*Ddx5*[TAM-KO]) in cultured spermatogonia at D2 versus vehicle-treated control (Control) and effects of treatment with pan-caspase inhibitor Z-VAD-FMK (with DMSO used as a control). All differences between groups are statistically significant (*P* < 0.05, Mann–Whitney *U* test, *n* = 4 biologically independent cell lines per condition, mean ± SEM shown).
**g** Immunofluorescence of cultured spermatogonia upon *Ddx5* ablation at D1 depicting an increase in caspase-mediated apoptosis. Cleaved caspase-3 (cCASP3) is used as a marker of apoptotic cells, with SALL4 used as a marker of spermatogonia. Inhibition of apoptosis using the pan-caspase inhibitor Z-VAD-FMK prevents loss of spermatogonia upon *Ddx5* ablation. Nuclei are counterstained with DAPI (DNA). All scale bars = 100 μm. **h** Quantification of cell fold recovery at D2 in cultured murine spermatogonia transduced with wildtype DDX5 (WT), helicase-inactive mutant DDX5 (NEAD) or tdTomato control constructs prior to tamoxifen-induced *Ddx5* ablation at D0. *P < 0.05; n.s.: not significant. Mann–Whitney *U* test, *n* = 4 biologically independent cell lines per condition. Mean ± SEM shown

Despite significant loss of cultured spermatogonia upon *Ddx5* ablation, we were able to extract RNA from remaining *Ddx5*-ablated cells for gene expression analysis. TAM-treated and vehicle-treated *Ddx5*[TAM-KO] spermatogonia were FACS-purified at D2 based on expression of EpCAM to eliminate potential feeder cell contamination[12]. We performed whole transcriptome analysis using RNA-seq to gain insight into the consequences of *Ddx5* loss in undifferentiated spermatogonia. We found that *Ddx5* loss resulted in differential expression of 6934 genes (false discovery rate <0.05) (Fig. 3d and Supplementary Data 2). We confirmed downregulation of *Ddx5* in TAM-treated samples and found aberrant expression of a number of key genes required for maintenance and function of spermatogonia. Key stem-associated and progenitor-associated genes such as *Nanos2*, *Gfra1*, *Sohlh1*, *Plzf* (*Zbtb16*), and *Gilz* (*Tsc22d3*) were downregulated and these changes confirmed by RT-qPCR in independent samples (Fig. 3d, Supplementary Fig. 6a and Supplementary Data 2). However, a number of spermatogonia-associated and spermatogenesis-associated genes were either upregulated (e.g., *Rarg*, *Kit*, *Thy1*, *T*, *Lhx1*), downregulated (*Nanos3*, *Rec8*, *Dnmt3b*, *Sycp3*), or unchanged (*Sall4*, *Pou5f1*), suggesting an overall dysregulation of gene expression rather than transition to a particular state (e.g., stem, differentiating, meiotic induction) (Fig. 3d, Supplementary Fig. 6a and Supplementary Data 2). We also noted upregulation of *Ddx17* in TAM-treated *Ddx5*[TAM-KO] spermatogonia by RNA-seq analysis; however, this could not be confirmed by RT-qPCR and corresponding levels of DDX17 protein were unchanged (Fig. 3b–d, Supplementary Fig. 5 and Supplementary Fig. 6a). Moreover, the severity of *Ddx5*[TAM-KO] phenotype in spermatogonia indicates that DDX17 cannot compensate for DDX5 function in undifferentiated spermatogonia.

Concerning the general mechanism by which spermatogonia were lost following *Ddx5* ablation, we observed upregulation of *Trp53* and genes encoding effectors of p53-mediated apoptosis (*Bax*, *Puma*, *Noxa*, *Gadd45b*)[40,41] by RNA-seq analysis (Fig. 3e) and we confirmed upregulation of *Puma* and *Gadd45b* by RT-qPCR in independent samples (Supplementary Fig. 6b). To confirm loss of *Ddx5*[TAM-KO] spermatogonia occurred by apoptosis, we treated cells with TAM in combination with the pan-caspase inhibitor, Z-VAD-FMK. At D2 post-treatment, TAM-treated cells were almost completely lost, whereas significantly more cells remained when treated in combination with Z-VAD-FMK. As expected, control cells treated with vehicle alone or in combination with Z-VAD-FMK showed increased cell numbers (Fig. 3f). By IF, we confirmed successful gene ablation in TAM-treated *Ddx5*[TAM-KO] spermatogonia under all conditions and increased levels of apoptotic marker cleaved capsase-3 compared with controls (Fig. 3g). Importantly, we confirmed that the substantial increase in cleaved caspase-3-positive cells evident upon *Ddx5* deletion was inhibited by Z-VAD-FMK (Fig. 3f, g). Although we could not confirm upregulation of all genes identified in our RNA-seq analysis by RT-qPCR, these data indicate that following *Ddx5* ablation, undifferentiated spermatogonia are lost due to p53-mediated apoptosis.

Next, we aimed to determine whether the helicase activity of DDX5 is required for survival of spermatogonia, as previous studies investigating the role of DDX5 in other systems have shown helicase-dependent and independent functions[22]. We transduced *Ddx5*[TAM-KO] cultured spermatogonia with lentiviral constructs containing wildtype DDX5 (DDX5-WT) or a helicase-inactive form of DDX5[42] (DDX5-NEAD) prior to TAM treatment. Although we were unable to elicit expression of DDX5 constructs to the same level as endogenous DDX5 and observed substantial apoptosis in all conditions (Supplementary Fig. 6c), we nevertheless observed ~5-fold higher cell numbers remaining in the DDX5-WT condition and ~3-fold higher in the DDX5-NEAD condition compared with control transduced cells at D2 post-TAM (Fig. 3h). These findings suggest that although the helicase activity of DDX5 is not entirely dispensable for survival of undifferentiated spermatogonia, there exists a helicase-independent function for DDX5 within spermatogonia. Moreover, *Ddx5* is specifically required for survival of undifferentiated spermatogonia, with its loss leading to aberrant expression of spermatogonial genes and rapid induction of apoptosis.

**DDX5 is required for splicing of key genes in spermatogonia.** DDX5 is involved in numerous transcriptional and post-transcriptional gene regulatory processes across various biological systems through interactions with other proteins[21–24,26]. To gain additional insight into DDX5 function in undifferentiated spermatogonia, we characterised DDX5 interacting proteins by IP and MS analysis from lysates of cultured wildtype cells. We identified 445 candidate proteins that interacted with DDX5 (Supplementary Data 3) and subjected these results to gene ontology (GO) term overrepresentation analysis according to classification by *molecular function* and *biological process* using PANTHER[43]. For GO molecular function and GO biological

process, we found significant overrepresentation of terms related to RNA binding and post-transcriptional processes, respectively (Fig. 4a and Supplementary Data 4 and 5). To confirm interactions identified by MS analysis that implicate DDX5 in RNA processing steps, we performed DDX5 IP from cultured wildtype spermatogonia followed by western blot analysis. We confirmed interaction of DDX5 with ELAVL1 (mRNA stability), PABP1 (mRNA stability/export), and SRSF3 (splicing/mRNA export) (Fig. 4b). Our data indicates that DDX5 is primarily involved in RNA processing in undifferentiated spermatogonia.

Given these findings and that DDX5 is implicated in splicing within other systems[21,25,44], we sought to establish a role for DDX5 in splicing within undifferentiated spermatogonia. We used the MISO (mixture-of-isoforms) probabilistic framework[45] to analyse our RNA-seq data for identification of alternative splicing events upon loss of Ddx5. We discovered 667 alternative splicing events upon loss of Ddx5 in spermatogonia (Bayes factor >10, ΔPSI ≥0.20) with the majority of events (= 406/667) annotated as cassette exon, where differential expression of specific exons was identified between control and TAM-treated Ddx5TAM-KO samples (Fig. 4c and Supplementary Data 6). From these events, we selected candidates based on involvement in spermatogenesis (Dazl, Ehmt2)[46–48] and key cellular processes such as DNA repair/homologous recombination (Rad50, Rad18)[49,50] and autophagy (Becn1)[51]. Loss of Ddx5 resulted in differential expression of exon 8 in Dazl (increased exon inclusion), exon 10 in Ehmt2 (increased exon skipping), exon 15 in Rad50 (increased exon skipping), exon 10 in Rad18 (increased exon inclusion), and exon 3 in Becn1 (increased exon skipping) (Fig. 4d). We confirmed these candidates were bound to DDX5 by RNA immunoprecipitation (RIP)-PCR (Fig. 4e), suggesting DDX5 is directly involved in their splicing, and we validated these alternative splicing events by PCR (Fig. 4f). Other candidates were confirmed to be differentially spliced upon Ddx5 loss (e.g., Fmr1) but were not detectably bound to DDX5, suggesting their splicing is indirectly regulated (Supplementary Fig. 7a–c). In addition, we confirmed that Hprt was not bound to DDX5 as a negative RIP control (Supplementary Fig. 7c).

Interestingly, the DNA repair genes Rad18 and Rad50 are required for maintenance of spermatogenesis in mice and loss of function of either gene results in infertility[49,50]. When we assessed gene expression of mis-spliced candidates, we found Rad50 to be substantially downregulated upon Ddx5 loss according to RNA-seq (Fig. 4g), and we confirmed this finding by RT-qPCR (Supplementary Fig. 7d). By coding sequence analysis of Rad50 mRNA, we found that skipping of exon 15 results in the introduction of a premature termination codon (Supplementary Fig. 7e) and is likely to result in nonsense mediated decay of the transcript[52]. Accordingly, we confirmed a decrease in RAD50 protein in TAM-treated Ddx5TAM-KO spermatogonia compared with controls by western blot (Fig. 4h) and IF (Fig. 4i). These findings demonstrate that DDX5 is an important trans-acting regulator of splicing in undifferentiated spermatogonia and its loss results in dysregulated splicing of key genes required for spermatogenesis. Moreover, p53-mediated apoptosis of spermatogonia following ablation of Ddx5 may be a direct consequence of splicing defects[53].

**DDX5 regulates cyclin mRNA export and stability**. To more fully understand the transcriptional consequences of Ddx5 loss in spermatogonia, we performed pathway analysis using the Ingenuity platform on differentially expressed genes identified by RNA-seq. The top two pathways identified for top canonical pathways and top tox lists (i.e., top biological "toxicity" associated with the observed gene expression changes) were found to be cell

cycle-related pathways (Fig. 5a and Supplementary Data 7). We observed aberrant expression of a number of genes involved in cell cycle regulation upon Ddx5 ablation in spermatogonia, including significant upregulation of the cell cycle arrest gene Cdkn1a (p21) (Fig. 5b and Supplementary Fig. 8a). However, core cell cycle-related genes were not predicted to be aberrantly spliced (Supplementary Data 6). Cell cycle analyses of surviving TAM-treated Ddx5TAM-KO cells by flow cytometry revealed a significant decrease in the proportion of cells in S-phase and an increase in the proportion of cells in G2/M-phase compared with controls (Fig. 5c). These data suggest that Ddx5 is specifically required in undifferentiated spermatogonia to maintain the expression of genes important for cell cycle and its loss results in cell cycle arrest at both G1 and G2 checkpoints.

CCNA2 regulates G1/S and G2/M transitions of the cell cycle, and is strongly expressed in spermatogonia[54]. Expectedly, given decreased Ccna2 expression observed upon Ddx5 ablation, we observed a decrease in CCNA2 level in TAM-treated cultured Ddx5TAM-KO spermatogonia compared with controls (Fig. 5d, e and Supplementary Fig. 8a). However, we noted that despite arrest at both checkpoints, positive regulators of the cell cycle such as Ccnd1 and Ccnd2[55] were upregulated in TAM-treated Ddx5TAM-KO spermatogonia (Fig. 5b, c and Supplementary Fig. 8a). When analysing protein levels, we found that CCND1 and CCND2 were decreased upon Ddx5 ablation in contrast to respective mRNA levels (Fig. 5b, d, e), suggesting post-transcriptional regulation of these genes was perturbed. To investigate whether DDX5 was directly regulating cyclin expression post-transcriptionally, we performed RIP-qPCR in cultured wildtype spermatogonia using an antibody against DDX5. We analysed cyclin mRNAs that were differentially expressed according to RNA-seq (Fig. 5b). For each transcript, we determined fold enrichment versus pull down with immunoglobulin G (IgG) control and found Ccnb2 (~70-fold), Ccnd1 (~40-fold), Ccna2 (~30-fold), and Ccnd2 (~12-fold) to be the most strongly enriched transcripts (Fig. 5f).

Given our observation of increased Ccnd1 and Ccnd2 mRNA expression yet decreased levels of corresponding protein in Ddx5-ablated spermatogonia, we considered that loss of DDX5 function results in impaired nuclear export of these transcripts. Restricted localisation of DDX5 to nuclei in spermatogonia (Fig. 1e) together with binding of cyclin transcripts to DDX5 suggested direct regulation of cyclin expression by post-transcriptional processes other than translation. We assessed nuclear export of Ccnd1 and Ccnd2 mRNA transcripts by determining nuclear-cytoplasmic ratios at D2 post-TAM in cultured Ddx5TAM-KO spermatogonia. We confirmed significant accumulation of both Ccnd1 and Ccnd2 mRNA in the nucleus of Ddx5-ablated spermatogonia (Fig. 5g), indicating that DDX5 directly regulates export of these transcripts. In contrast, nuclear export of Ccna2 mRNA was not significantly disrupted upon Ddx5 loss (Supplementary Fig. 8b).

Ccna2 transcript is bound to DDX5 and both mRNA plus protein levels of CCNA2 are downregulated upon Ddx5 loss, suggesting a distinct mode of post-transcriptional regulation by DDX5. Consequently, we assessed stability of Ccna2 mRNA upon Ddx5 loss in Ddx5TAM-KO spermatogonia treated with the transcription inhibitor actinomycin D. However, there was no evident difference in Ccna2 mRNA stability between control and TAM-treated samples (Fig. 5h), suggesting that Ccna2 downregulation occurs as an indirect transcriptional consequence of Ddx5 ablation. Importantly, we observed a significant reduction in levels of Ccnb2 mRNA upon Ddx5 loss after 4 h of actinomycin D treatment, indicating that stability of Ccnb2 transcript was reduced (Fig. 5h). Given significant reduction in Ccnb2 expression upon Ddx5 loss (Fig. 5b and Supplementary Fig. 8a) and

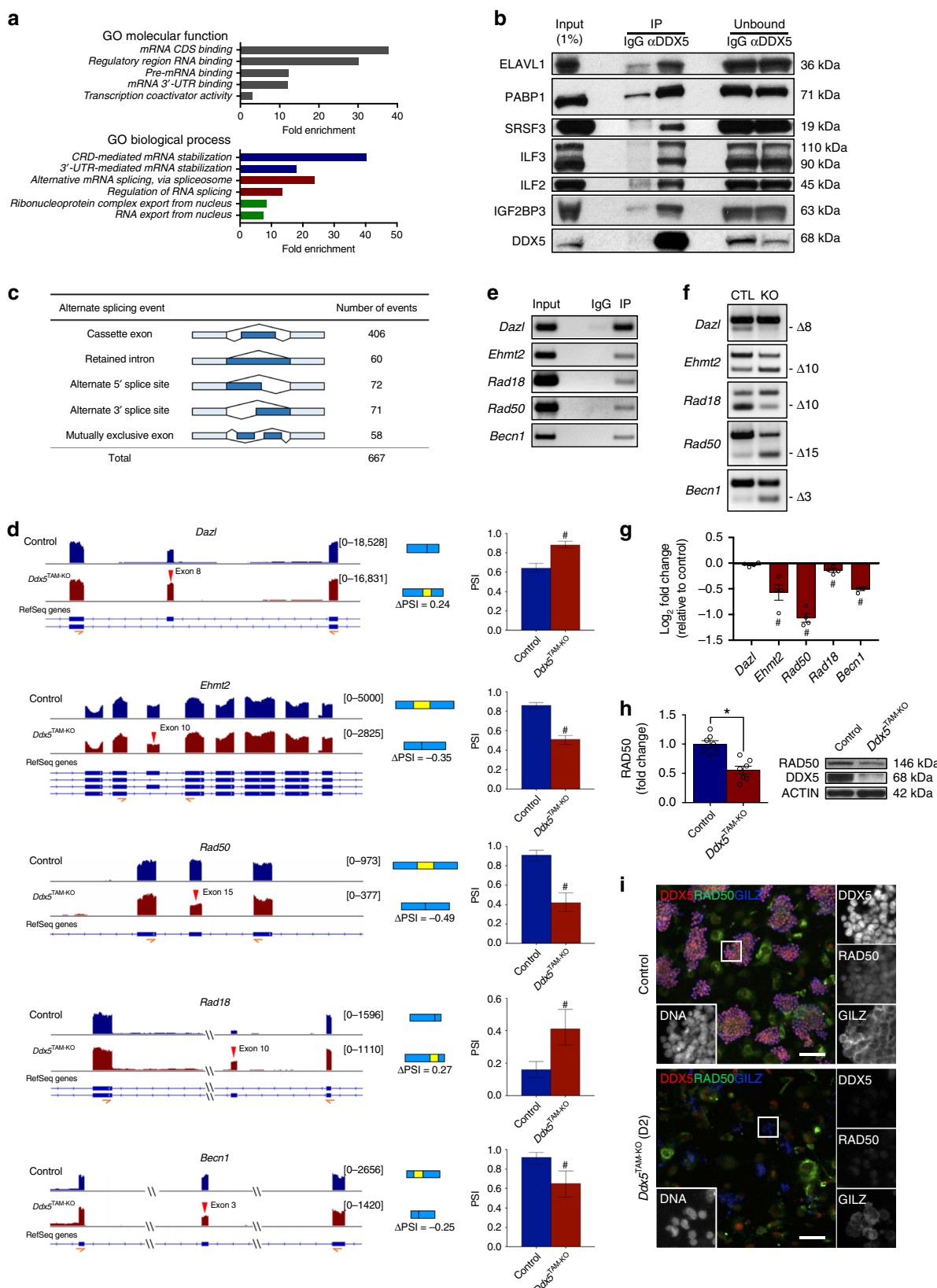

association of *Ccnb2* mRNA with DDX5, our results indicate that the stability of *Ccnb2* transcript is directly enhanced by DDX5. We used *Myc* as an assay control and observed mRNA stability comparable to previous reports in both control and TAM-treated cells[56] (Supplementary Fig. 8c).

These findings demonstrate that DDX5 is required for post-transcriptional regulation of cyclin mRNAs in undifferentiated spermatogonia. Furthermore, that cell cycle arrest observed upon *Ddx5* loss is, at least in part, a consequence of impaired export and stability of specific cyclin transcripts.

**Fig. 4** DDX5 is an essential regulator of splicing in undifferentiated spermatogonia. **a** Gene ontology (GO) term enrichment analysis of data obtained from DDX5 immunoprecipitation followed by mass spectrometry using wildtype murine cultured undifferentiated spermatogonia ($n = 3$ independent experiments). **b** DDX5 immunoprecipitation followed by western blot in wildtype cultured spermatogonia confirming the interaction of DDX5 with proteins involved in pre-mRNA splicing (SRSF3), maintenance of mRNA stability (ELAVL1, PABP1, ILF3, ILF2, IGF2BP3), mRNA export (PABP1, SRSF3, ILF2, ILF3, IGF2BP3) and translation (IGF2BP3). Representative of $n = 2$ independent experiments. **c** Summary of differential splicing analysis performed between control and *Ddx5*-ablated cultured spermatogonia. Numbers of predicted alternative splicing events in each category upon *Ddx5* deletion are indicated. **d** Visualisation of differential splicing analysis of RNA-sequencing data comparing control and *Ddx5*-ablated spermatogonia. Tracks are shown for selected candidate genes (left). Red arrows indicate differentially spliced exon. Schematics of alternative splicing events are shown (blue and yellow rectangles) (middle). Change in "percent spliced in" between conditions is shown as a value below splicing schematics (ΔPSI) and in bar charts (right). #: changes meet analysis cut-offs (ΔPSI >0.20, Bayes Factor ≥10). PSI ± 95% confidence interval shown. **e** RNA-immunoprecipitation using DDX5 antibody in cultured wildtype spermatogonia followed by PCR and gel electrophoresis for differentially spliced candidates depicted in **d**. Representative of $n = 3$ independent samples per condition. **f** PCR validation of differentially spliced candidates identified in **d**. CTL vehicle-treated control spermatogonia, KO *Ddx5*-ablated cultured spermatogonia. Differentially spliced exons are depicted by "Δexon number". Representative of $n = 3$ independent samples. Position of the PCR primers used are depicted in **d** in orange beneath schematic showing gene structure. **g** Bar chart of RNA-sequencing analysis for selected candidates. Fold change in *Ddx5*-ablated spermatogonia relative to vehicle-treated control. # denotes FDR <0.05. Mean ± SEM shown; $n = 4$ independent biological replicates per condition. **h** RAD50 protein levels by western blot in vehicle-treated control (Control) versus *Ddx5*-ablated cultured spermatogonia (*Ddx5*^TAM-KO). *$P < 0.05$; two-tailed unpaired *t*-test; mean ± SEM shown; $n = 4$ independent biological replicates per condition. Representative western blot shown on right. **i** RAD50 immunofluorescence of vehicle-treated control (top) and *Ddx5*-ablated (bottom) cultured spermatogonia at D2 post-treatment. Representative of $n = 4$ independent biological replicates per condition. Scale bars = 100 μm

**ILF3 associates with DDX5 and regulates CCND1 expression.** While investigating the role of DDX5 in cell cycle regulation in spermatogonia, we identified and confirmed interaction of DDX5 with ILF3 and its protein partners ILF2 and IGF2BP3[57,58] (Fig. 4b and Supplementary Data 3). ILF3 (interleukin enhancer binding factor 3, also known as NF90 or NFAR) is an RNA-binding protein important for transcriptional and post-transcriptional regulation of *Il2* in T cells and plays key regulatory roles in cell cycling[58–61].

In light of the identified functional connection between DDX5 and ILF3, we examined protein expression of ILF3 in male germline cells by IF. We observed ILF3 expression in all PLZF-positive cultured undifferentiated spermatogonia (Fig. 6a), as well as in vivo co-localisation with DDX5 in the nucleus of spermatogonia, spermatocytes, spermatids, and Sertoli cells of mouse seminiferous tubules (Fig. 6b). Previous studies have demonstrated that ILF3 is involved in post-transcriptional regulation of cyclin mRNAs[57,62]. Therefore, we performed ILF3 RIP-qPCR experiments in cultured wildtype spermatogonia to identify cyclin mRNAs bound to ILF3 and elucidate a potential co-regulatory role with DDX5. We found strong enrichment of *Ccnb2* (~1124-fold), *Ccna2* (~718-fold), *Ccnd1* (~84-fold), *Ccne1* (~73-fold), and *Ccnd2* (~62-fold) transcripts (versus IgG control), whereas others such as *Ccnd3* and *Ccna1* showed minimal enrichment or did not appear to be bound to ILF3, respectively (Fig. 6c).

ILF3 regulates the stability of *Ccne1* mRNA by direct binding to its 3'-UTR and post-transcriptionally regulates *Ccnd1* through interactions with IGF2BP3[57,62]. Given the DDX5/ILF3 interaction and known role of ILF3 in cyclin transcript regulation, we sought to determine whether loss of *Ilf3* in undifferentiated spermatogonia would result in a phenotype similar to that observed upon loss of *Ddx5*. We therefore used a lentiviral doxycycline (DOX)-inducible CRISPR/Cas9 system[63] to ablate *Ilf3* in cultured wildtype spermatogonia (referred to as *Ilf3*^−/− cells). At D4 post-DOX treatment, we confirmed loss of *Ilf3* by RT-qPCR and at the protein level by IF and western blot (Fig. 6d, f, g). In contrast to *Ddx5* loss, *Ilf3*^−/− spermatogonia persisted in culture and continued to proliferate. However, we observed a significantly reduced growth rate of *Ilf3*^−/− cells compared with DOX-treated wildtype cells when analysed D2 post-treatment (Fig. 6d, e). Analysis of gene expression in *Ilf3*^−/− spermatogonia by RT-qPCR did not find any significant change in expression of

analysed cyclin genes compared with controls (Fig. 6f). However, CCND1 protein was substantially decreased upon loss of *Ilf3* (Fig. 6g); a phenotype analogous to observed *Ccnd1* mRNA export defects upon *Ddx5* loss. However, we were unable to confirm consistent changes in *Ccnd1* nuclear-cytoplasmic ratio upon *Ilf3* loss, suggesting alternate post-transcriptional regulatory mechanisms are in play (Supplementary Fig. 9).

The difference in severity of cellular phenotypes and disruption in cyclin expression upon loss of *Ddx5* and *Ilf3* suggests that regulation of cyclin genes in undifferentiated spermatogonia and consequent effects on cell cycling are not dependent on a DDX5/ILF3 complex. Rather, DDX5-dependent post-transcriptional regulation is important for cyclin mRNA expression and cell cycle regulation whereas ILF3 appears dispensable, with the exception of *Ccnd1*.

**DDX5 and PLZF share DNA binding sites and co-regulate *Ilf3*.** Previous studies in other systems have shown that DDX5 interacts with and modulates activity of transcriptional regulators including MYOD, RUNX2, the tumour suppressor p53, and oestrogen receptor alpha[22]. Considering the ability of DDX5 to interact with transcription factor PLZF (Fig. 1b), we investigated whether DDX5 acts as a PLZF co-factor in undifferentiated spermatogonia. We therefore performed ChIP-sequencing (ChIP-seq) for DDX5 on cultured wildtype undifferentiated spermatogonia and identified 101 putative binding sites, of which 31 were found within gene promoter regions (Fig. 7a and Supplementary Data 8). Motif analysis of bound regions identified enrichment of the E-box-containing CLEAR element (5'-GTCACGTGAC-3'), a binding motif for the autophagy regulator TFEB[64] (Fig. 7b). Amongst our ChIP-seq analysis, we found DDX5 bound within the promoter regions of autophagy-related genes *Lamp1*, *Gns*, *Hexa*, *Tmem55b*, and *Atp6v1h* (Fig. 7c). Of these genes, only *Hexa* (1.12-fold downregulated), *Tmem55b* (1.13-fold upregulated), and *Atp6v1h* (1.15-fold upregulated) were differentially expressed according to RNA-seq analysis in cultured *Ddx5*-ablated spermatogonia compared with controls (false discovery rate <0.05) (Fig. 7d); however, these minor changes could not be confirmed by RT-qPCR (Supplementary Fig. 10d). Furthermore, we were unable to establish an interaction between DDX5 and TFEB by co-IP, indicating that DDX5 is unlikely to be a TFEB co-factor (Supplementary Fig. 10a). Therefore, while recruitment of DDX5 to CLEAR motifs is compelling, our data indicate that

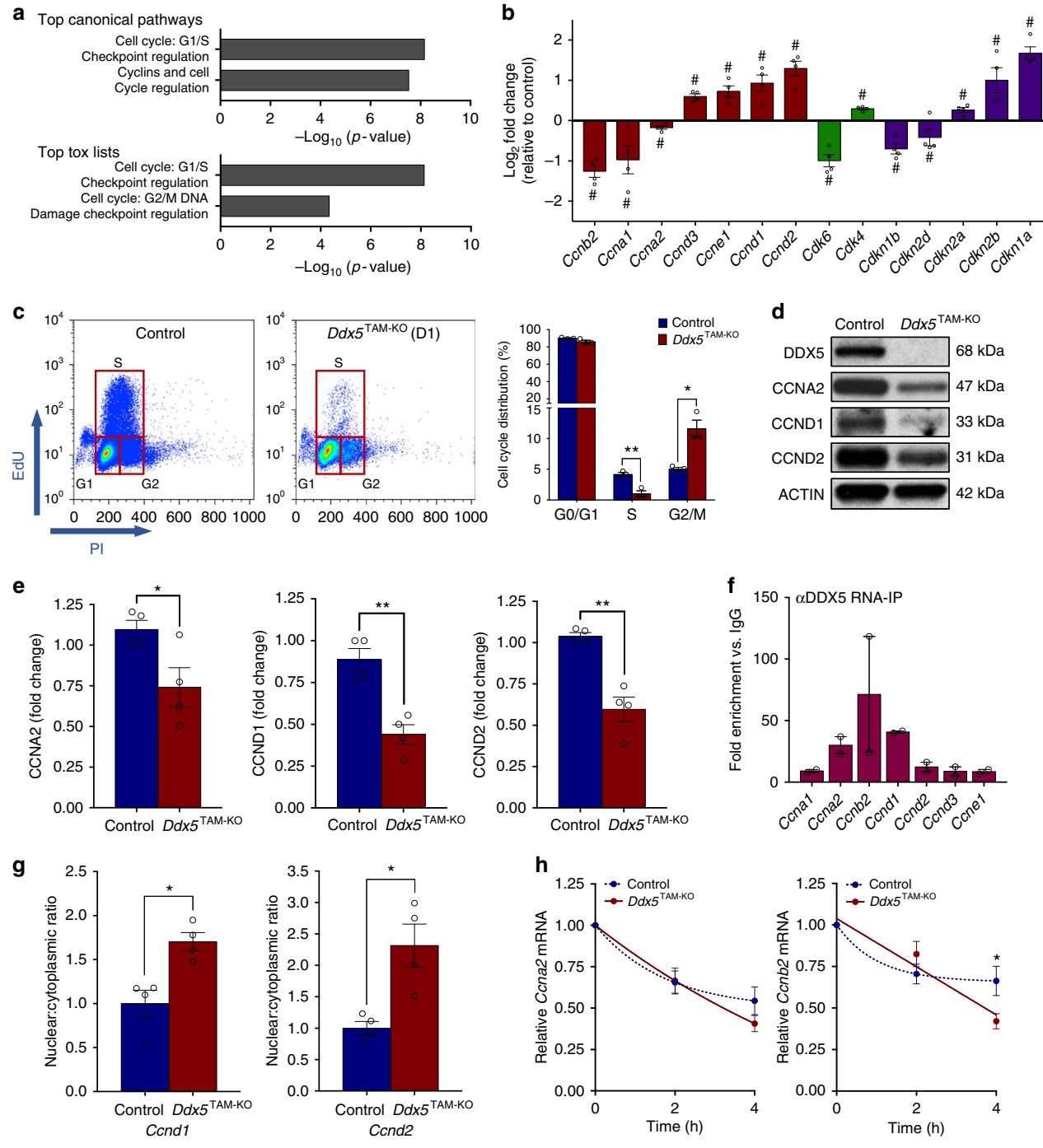

DDX5 is not an essential transcriptional regulator of autophagy-related genes in spermatogonia.

To investigate the potential role of DDX5 as a PLZF transcriptional co-factor in spermatogonia, we compared DDX5 ChIP-seq and RNA-seq datasets to identify DDX5-bound genes differentially expressed upon *Ddx5* loss. Forty-nine genes were potentially transcriptionally regulated by DDX5, with the number of upregulated and downregulated genes (25 versus 24, respectively) suggesting DDX5 may act as both an activator and repressor of transcription (Fig. 7e and Supplementary Fig. 10b, c). Next, we compared these 49 genes with a published PLZF ChIP-seq study[16] to identify shared binding sites. We found DDX5 and PLZF overlapping binding sites within promoter regions of *Amdhd2, Ilf3, Arhgap12, Tmem55b, 4933434E20Rik,* and *Zbtb25*

(Fig. 7f and Supplementary Data 8). Next, we determined changes in expression of these genes according to RNA-seq and found significant downregulation of *Amdhd2* (1.53-fold), *Ilf3* (1.44-fold), and *Arhgap12* (1.31-fold), and upregulation of *Tmem55b* (1.13-fold, as described earlier), *4933434E20Rik* (1.16-fold) and *Zbtb25* (1.24-fold) upon *Ddx5* loss. However, only changes in *Ilf3* and *Amdhd2* expression could be confirmed by RT-qPCR (Fig. 7d, g and Supplementary Fig. 10e). We examined expression of these DDX5-bound and PLZF-bound genes in undifferentiated spermatogonia isolated from testes of *Plzf*-null mice as described[11,12]. However, from this panel of genes only *Ilf3* was significantly downregulated in *Plzf*-null cells compared with wildtype controls (Fig. 7h). We identified DDX5 binding to the promoter of *Ilf3* as one of the top peaks by ChIP-seq and confirmed that this site

**Fig. 5** DDX5 regulates cell cycle progression in spermatogonia post-transcriptionally. **a** Pathway analysis of differentially expressed genes identified by RNA-sequencing comparing control and *Ddx5*-ablated cultured spermatogonia. **b** Differentially expressed cell cycle-related genes identified by RNA-sequencing comparing control and *Ddx5*-ablated cultured spermatogonia. Fold change in *Ddx5*-ablated spermatogonia shown relative to control. Mean ± SEM shown; $n = 4$ independent biological replicates per condition. # denotes FDR <0.05. **c** Representative flow cytometry plots (left) showing *Ddx5* loss results in cell cycle arrest in cultured undifferentiated spermatogonia at D1 post-ablation. Control: vehicle-treated control spermatogonia; *Ddx5*[TAM-KO]: tamoxifen-induced *Ddx5*-ablated spermatogonia. Representative of $n = 3$ independent experiments. Histogram (right) depicts quantification of cell proportions for each cell cycle phase determined by flow cytometry. *$P < 0.05$; **$P < 0.01$. Two-tailed unpaired *t*-test, $n = 3$ independent experiments per condition. **d** Representative western blot showing reduced CCNA2, CCND1 and CCND2 protein upon loss of *Ddx5* (*Ddx5*[TAM-KO]) compared with control in cultured undifferentiated spermatogonia at D2 post-tamoxifen treatment. Representative of $n = 3$ independent experiments. **e** Quantification of western blots represented in **d**. Mean ± SEM shown; *$P < 0.05$; **$P < 0.01$; Two-tailed unpaired *t*-test. $n = 4$ independent samples per condition. **f** RNA immunoprecipitation using DDX5 antibody in cultured wildtype undifferentiated spermatogonia followed by RT-qPCR for cyclin genes. Fold enrichment versus IgG control shown. Mean ± SEM shown; $n = 2$ independent experiments. **g** Nuclear-cytoplasmic ratio of *Ccnd1* and *Ccnd2* transcripts determined by RT-qPCR of control and *Ddx5*-ablated (*Ddx5*[TAM-KO]) cultured undifferentiated spermatogonia subcellular fractions. Fold change relative to control shown. Mean ± SEM shown; *$P < 0.05$; Mann–Whitney *U* test; $n = 4$ independent samples per condition. **h** Analysis of mRNA stability over time following inhibition of transcription by actinomycin D treatment in control versus *Ddx5*-ablated (*Ddx5*[TAM-KO]) cultured undifferentiated spermatogonia. mRNA expression determined by RT-qPCR (normalised to *Actb*) relative to initial levels at time 0 h. *$P < 0.05$; Two-way ANOVA with Bonferroni multiple comparisons test; mean ± SEM shown; $n = 4$ independent samples per condition

overlapped with a previously identified PLZF-binding site (Fig. 7i). In addition to downregulation of *Ilf3* mRNA, we found reduced levels of ILF3 protein at D2 in TAM-treated versus vehicle-treated *Ddx5*[TAM-KO] spermatogonia (Fig. 7j).

Finally, we performed siRNA-mediated *Plzf* knockdown in cultured spermatogonia (with non-targeting siRNA used as control) followed by DDX5 ChIP-qPCR to determine whether DDX5 recruitment to gene promoters was PLZF-dependent. First, we confirmed efficient *Plzf* knockdown by western blot (Supplementary Fig. 11a). Next, we performed DDX5 ChIP in control versus *Plzf* knockdown spermatogonia and determined enrichment by qPCR for peak regions identified by DDX5 and PLZF ChIP-seq[16] (Supplementary Fig. 11b). We selected candidate PLZF-DDX5 co-regulated genes *Ilf3*, *Amdhd2* and *Lamp1* for analysis, plus uniquely DDX5-bound and PLZF-bound genes *Zmym6* and *Tex13b*[16], respectively, as controls. However, we found no difference in DDX5 enrichment at *Ilf3*, *Amdhd2* and *Lamp1* promoters between control and *Plzf* siRNA conditions, suggesting that DDX5 recruitment to these target genes is PLZF-independent (Supplementary Fig. 11c).

We conclude that while DDX5 and PLZF are both bound at the *Ilf3* promoter and co-regulate *Ilf3* expression, other factors besides PLZF mediate DDX5 recruitment to this gene. Furthermore, our data indicates that DDX5 is a multifaceted regulator of ILF3 function given its association with the ILF3/ILF2/IGF2BP3 protein complex and ability to regulate *Ilf3* expression.

## Discussion

Spermatogenesis is a complex process dependent on the co-ordinated regulation of gene expression at transcriptional and post-transcriptional levels[1]. It has been demonstrated that the transcription factor PLZF is important for self-renewal of undifferentiated spermatogonia, the population of adult stem and progenitor cells that enable lifelong male fertility[11,14]. More recently, the role of various RNA-binding proteins during spermatogenesis has been uncovered, particularly during meiotic and post-meiotic stages of spermatogenesis where periods of transcriptional quiescence demand mechanisms for post-transcriptional control of gene expression[1]. However, few RNA-binding factors with specific functions in undifferentiated spermatogonia have been identified. NANOS2[19], DND1[65], and BCAS2[20] have been described as important RNA-binding proteins within undifferentiated spermatogonia and each has a unique role in post-transcriptional regulation, highlighting the complexity of this process and the need for further investigation.

In this study, we have identified an interaction between PLZF and the RNA helicase DDX5. We demonstrate that DDX5 is expressed throughout the male germline and its loss has catastrophic effects on spermatogenesis. Using a murine knockout model, we show that *Ddx5* ablation results in rapid loss of spermatogonia in vivo and in vitro. Moreover, we demonstrate that *Ddx5* ablation causes aberrant expression of genes involved in maintenance of spermatogonia, cell cycle, and apoptosis. *Ddx5* is particularly essential for spermatogonial function; the inducible Cre driver used to delete *Ddx5* in vivo is active in multiple tissues[38], yet TAM-treated *Ddx5*[TAM-KO] mice did not display any overt phenotype besides germline depletion.

While a specific role for DDX5 within the male germline has not been described previously, a study detailed *DDX5* expression in subsets of spermatogenic cells in humans[30]. Consistent with this study, we identified robust expression of DDX5 in PLZF-positive spermatogonia of humans and non-human primates although DDX5 was also detected in spermatocytes, spermatids, and somatic cell populations within the testis. We observed a ubiquitous expression of *Ddx5* in testis cells of mice and previous studies also identified DDX5 within other murine spermatogenic cell types[31], confirming its broad expression within the testis. Nevertheless, expression of *DDX5* in primate spermatogonia suggests that the essential role of DDX5 in mouse spermatogonial maintenance is conserved in other mammalian species.

In this study, we observed significant loss of spermatogonia upon ablation of *Ddx5* both in vivo and in vitro. In contrast, *Ddx5* appeared dispensable within fibroblasts and its loss potentially compensated for by upregulation of the related helicase DDX17[26]. Previous studies have identified functional redundancy between DDX5 and DDX17 in regulation of splicing, ribosome biogenesis, cell proliferation, and transcriptional co-activation[21–23,26,27]. However, studies employing individual knockdown of *Ddx5* and *Ddx17* have also provided evidence for non-overlapping roles[22]. For example, DDX5 is important for the p53-mediated DNA damage response in cancer cell lines, whereas *Ddx17* knockdown had no effect on this process[42]. Although *Ddx17* expression is detected in undifferentiated spermatogonia and upregulated at the transcript level upon *Ddx5* deletion according to RNA-seq, our findings support a critical role for DDX5 in spermatogonia that evidently cannot be compensated for by DDX17.

Through MS analysis, we have shown that DDX5 interacts with a number of proteins involved in post-transcriptional gene regulatory processes including pre-mRNA splicing, maintenance of mRNA stability, and mRNA export. Although the role of DDX5 in these processes is somewhat appreciated from other

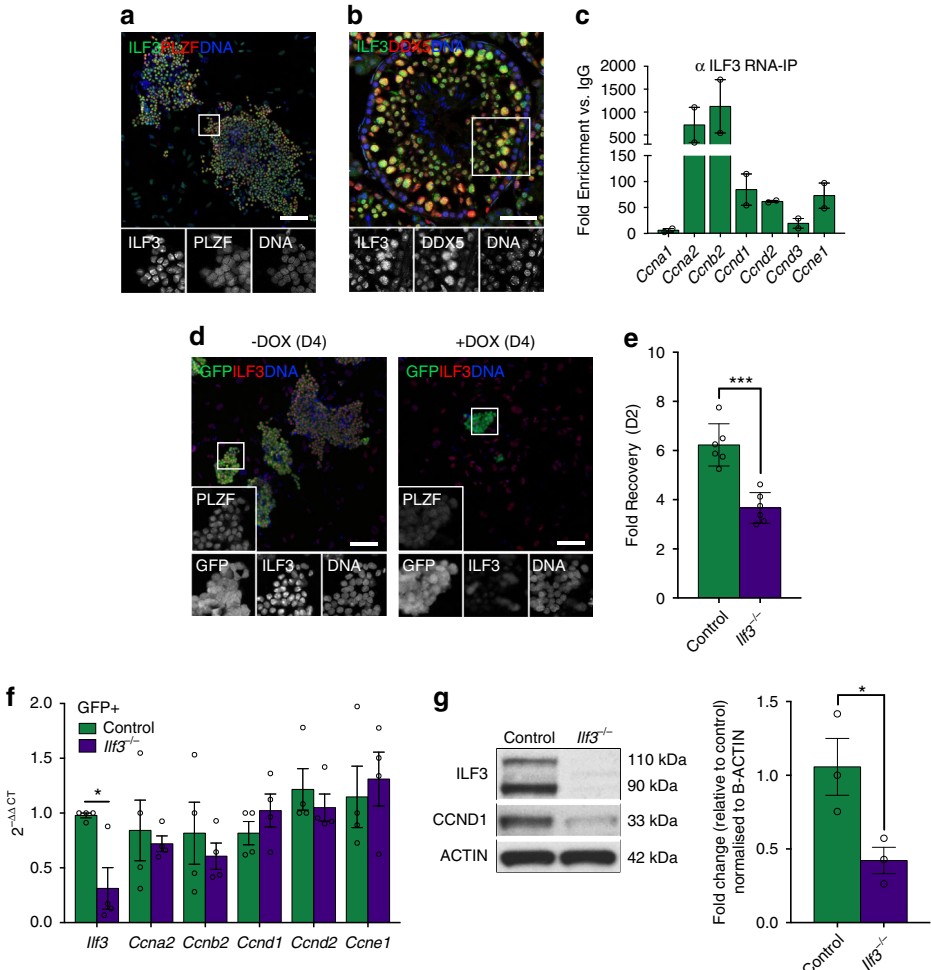

**Fig. 6** *Ilf3* loss reduces proliferation and disrupts cyclin expression in spermatogonia. **a** Immunofluorescence showing ILF3 is expressed in PLZF-positive cultured undifferentiated spermatogonia. Scale bar = 100 μm. **b** Immunofluorescence showing ILF3 is co-localised to DDX5-expressing cells within the murine seminiferous tubule. Scale bar = 50 μm. **c** RNA immunoprecipitation using ILF3 antibody in cultured wildtype undifferentiated spermatogonia followed by RT-qPCR for cyclin genes. Fold enrichment versus IgG control shown. Mean ± SEM shown; $n = 2$ independent experiments. **d** Doxycycline (DOX)-inducible CRISPR/Cas9-mediated *Ilf3* ablation in cultured undifferentiated spermatogonia shows efficient loss of ILF3 at D4 post-DOX treatment (+DOX) compared with control cells without DOX treatment (-DOX). sgRNA constructs targeting *Ilf3* are GFP-tagged, allowing visualisation of spermatogonia with an anti-GFP antibody. Scale bars = 100 μm. **e** Loss of *Ilf3* in cultured undifferentiated spermatogonia results in reduced cell numbers at D2 post-DOX treatment compared with DOX-treated wildtype control cells. \*\*\*$P < 0.001$. Two-tailed unpaired *t*-test, $n = 6$ replicates per condition. Mean ± SD shown. **f** RT-qPCR confirms downregulation of *Ilf3* upon CRISPR/Cas9-mediated ablation (*Ilf3*−/−) compared with control in cultured undifferentiated spermatogonia at D4 post-DOX treatment. Tested cyclin genes showed no significant difference. Cell samples were FACS-purified according to GFP to isolate undifferentiated spermatogonia prior to RNA extraction. \*$P < 0.05$; Mann–Whitney *U* test, $n = 4$ independent samples per condition Mean ± SEM shown. **g** Western blot depicting reduced CCND1 protein upon CRISPR/Cas9-mediated *Ilf3* ablation in cultured undifferentiated spermatogonia at D4 post-DOX treatment. Representative of $n = 3$ independent samples per condition (left). Quantification of CCND1 protein in control versus *Ilf3*-ablated (*Ilf3*−/−) spermatogonia (right). Mean ± SEM \*$P < 0.05$; Two-tailed unpaired *t*-test; $n = 3$ independent samples per condition

systems[21,24,29], it is well known that cell type-specific post-transcriptional regulation occurs and defects in ubiquitously expressed RNA-binding proteins can lead to disease phenotypes within specific tissues[66]. Here, we have identified an essential post-transcriptional regulator in spermatogonia that acts at multiple stages of RNA processing.

Our results support an essential role for DDX5 in splicing regulation within spermatogonia. It is known that splicing can be regulated in a tissue-specific manner and mediators of alternative splicing within the testis have been described[1,67]. We propose that DDX5 is central to the maintenance of an appropriate splicing programme through interactions with other known splicing regulators in spermatogonia and its loss leads to aberrant splicing of key spermatogenic and cellular genes. These splicing defects are likely to result in severely compromised gene expression or

the translation of inappropriate protein isoforms, which may account for the rapid loss of spermatogonia observed upon *Ddx5* ablation.

Interestingly, we found the important germ cell genes *Dazl* and *Ehmt2* to be aberrantly spliced in *Ddx5*-ablated spermatogonia, similar to a recent study that identified a role for BCAS2 in splicing within spermatogonia[20]. *Dazl* encodes an RNA-binding protein essential for germ cell maintenance and meiotic induction[46] and knockout mice are known to be infertile[48]. Our results indicate that *Ddx5* loss results in decreased levels of the *Dazl*-Δ8 isoform. To our knowledge, a role for *Dazl*-Δ8 in the male germline has not been defined. However, *Dazl*-Δ8 binds unique targets in mouse embryonic stem cells;[68] therefore, its loss may have important consequences in spermatogonia that warrants further investigation. Similarly, a specific role for *Ehmt2*-Δ10 in

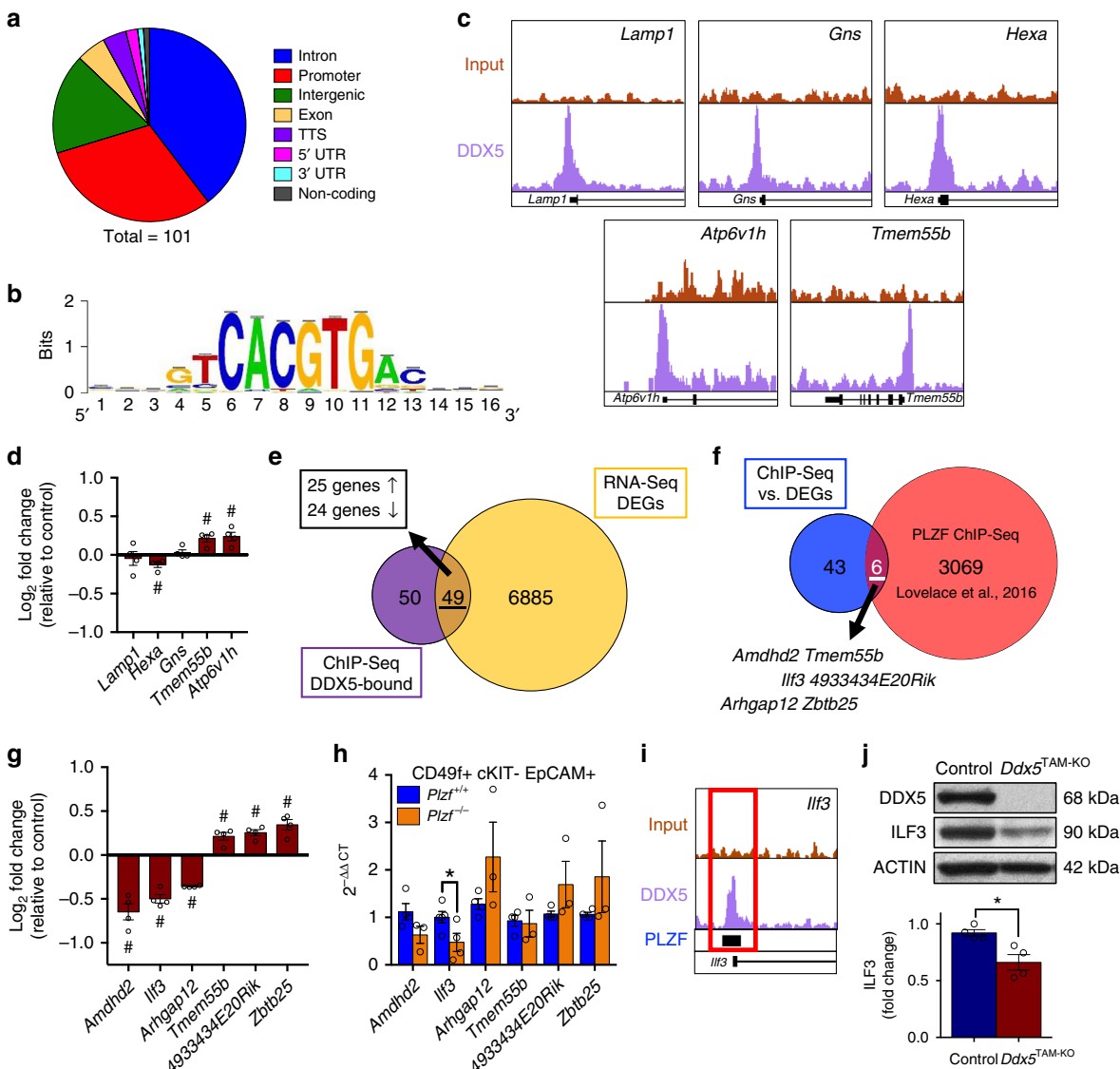

**Fig. 7** DDX5 regulates transcription of a subset of genes in spermatogonia. **a** Summary of binding locations for DDX5 ChIP-sequencing in undifferentiated spermatogonia. **b** Motif analysis of DDX5 ChIP-sequencing identifying enrichment of the autophagy-related and E-box-containing CLEAR consensus sequence (5′-GTCACGTGAC-3′). **c** DDX5 ChIP-sequencing in undifferentiated spermatogonia shows binding to the promoter region of autophagy-related genes *Lamp1*, *Hexa*, *Gns*, *Tmem55b* and *Atp6v1h*. **d** Comparison of identified autophagy-related genes shown in **c** by RNA-sequencing of control versus *Ddx5*-ablated cultured undifferentiated spermatogonia. Fold change versus control shown. # denotes FDR <0.05. Mean ± SEM shown; *n* = 4 independent biological replicates per condition. **e** Comparison of DDX5 ChIP-sequencing results in cultured wildtype undifferentiated spermatogonia (purple) versus differentially expressed genes identified by RNA-sequencing of control versus *Ddx5*-ablated cultured undifferentiated spermatogonia (yellow). DEGs: differentially expressed genes. **f** Comparison of DDX5-bound differentially expressed genes identified in **e** (blue) to a previously published PLZF ChIP-sequencing data set by Lovelace et al. (2016) (red). **g** Differentially expressed genes identified by RNA-sequencing of control versus *Ddx5*-ablated cultured undifferentiated spermatogonia that are also bound by PLZF according to previous ChIP-sequencing data (Lovelace et al., 2016). Fold change versus control shown. # denotes FDR <0.05. Mean ± SEM shown; *n* = 4 independent biological replicates per condition. **h** RT-qPCR shows significant downregulation of *Ilf3* in *Plzf*⁻/⁻ undifferentiated spermatogonia isolated by FACS from adult mouse testes (CD49f + c-KIT− EpCAM + cells). All other tested genes showed no significant difference. *$P$ < 0.05. Mann–Whitney $U$ test, $n$ = 4 $Plzf^{+/+}$ control and n = 3 $Plzf^{-/-}$ mice. **i** DDX5 and PLZF are present at the same binding site within the promoter region of *Ilf3*. Input (red) and DDX5 (purple) depict DDX5 ChIP-sequencing findings of this study. PLZF (blue with black bar) depict a previously identified PLZF binding site. **j** Western blot showing loss of ILF3 protein at D2 following tamoxifen-induced *Ddx5* ablation ($Ddx5^{TAM-KO}$) (top). Quantification of ILF3 protein in control versus $Ddx5^{TAM-KO}$ spermatogonia (bottom). Mean ± SEM shown; *$P$ < 0.05; Two-tailed unpaired $t$-test; $n$ = 4 independent samples per condition

male germ cells is unknown, although the role of *Ehmt2* as an essential epigenetic regulator during embryogenesis and spermatogenesis is defined[47]. A distinct role for *Ehmt2-Δ10* is described in neuronal differentiation[69], suggesting additional roles in other systems may yet be uncovered.

In addition to *Dazl* and *Ehmt2*, we found that DDX5 was important for the splicing of double-strand break (DSB) repair genes *Rad18* and *Rad50*. It is known that DSB repair is pivotal for meiotic homologous recombination[70] and both *Rad18* and *Rad50* mutant mice are infertile due to spermatogenic defects[49,50].

*Rad18* and *Rad50* are expressed in spermatogonia and spermatogenic failure in these models is suggested to be due to impaired spermatogonial stem cell maintenance and pre-meiotic defects, respectively[49,50]. Therefore, aberrant splicing of these genes in *Ddx5*-ablated spermatogonia leading to impaired protein function or expression likely results in deleterious effects. We also found aberrant splicing of *Becn1*, an important regulator of autophagy[51]. Although a specific role for *Becn1* in spermatogonia has not been defined, autophagy is a critical cellular process that is important throughout spermatogenesis during normal and pathological conditions[71].

Importantly, major spliceosome defects are suggested to be associated with non-obstructive azoospermia in humans and prevent spermatogonial differentiation in *Drosophila*[72]. Disruption of the pre-mRNA splicing machinery is known to result in p53 activation, p21 upregulation and cell cycle arrest in G1 phase[53], all of which were observed within *Ddx5*-ablated spermatogonia in our study. Therefore, the rapid and significant apoptosis of spermatogonia that occurs upon *Ddx5* loss may also be due to its role as core spliceosome component. However, we suggest that the negative effects of *Ddx5* deletion on cell proliferation are due to defects in specific DDX5-mediated post-transcriptional regulatory mechanisms.

We found that loss of *Ddx5* in spermatogonia results in cell cycle arrest at both G1/S and G2/M stages. Several cell cycle-related genes were aberrantly expressed, including strong upregulation of *Cdkn1a* (*p21*), a negative regulator of the G1/S checkpoint. Interestingly, we found a number of cyclin mRNA transcripts bound to DDX5 and showed that nuclear export and stability of a subset of these were regulated by DDX5. *Ccnd1* and *Ccnd2* transcripts were retained aberrantly in the nucleus upon *Ddx5* ablation and protein levels were significantly reduced. It has been shown that RNA-binding proteins can have roles in splicing and mRNA nuclear export of different transcripts within a single cell type that contributes to overall cellular function[56]. Besides their general requirement for G1/S-phase progression[54], CCND1 and CCND2 are dynamically regulated during spermatogenesis and are predominately expressed in specific subsets of spermatogonia. CCND2 has been shown to be important for long-term maintenance of spermatogonial stem cell self-renewal[73], whereas *Ccnd1* expression is predominantly associated with differentiating subsets of spermatogonia[37]. Therefore, we suggest that impaired spermatogonial function as a result of CCND1 and CCND2 loss may contribute to the overall *Ddx5*-knockout phenotype in addition to expected effects on G1/S-phase transition. Importantly, we demonstrate that DDX5 is required for efficient nuclear export of these transcripts and maintenance of CCND1 and CCND2 protein in spermatogonia.

We found that DDX5 bound to *Ccna2* and *Ccnb2* transcripts in spermatogonia and these genes were downregulated upon *Ddx5* ablation. *Ccna2* is expressed strongly in spermatogonia and is required for progression through G1/S and G2/M checkpoints, whereas CCNB2 functions only at the G2/M transition[54]. The binding of DDX5 to these transcripts and its absence at *Ccna2* and *Ccnb2* promoters by ChIP-seq coupled with differential expression upon *Ddx5* loss suggests a post-transcriptional regulatory role; however, we did not find defects in mRNA export or transcript stability of *Ccna2* in *Ddx5*-ablated spermatogonia. We suggest that *Ccna2* downregulation is an indirect transcriptional consequence but nevertheless can contribute to the G2/M arrest observed in spermatogonia upon *Ddx5* loss. We cannot exclude that other RNA-binding proteins are involved in maintaining the stability of particular transcripts, either independently or in association with DDX5. For example, we found DDX5 interacts with ELAVL1 in spermatogonia and this protein is known to be important for maintenance of mRNA stability through 3′-UTR-

mediated interactions[74]. In contrast, we found that the stability of *Ccnb2* mRNA was significantly reduced in *Ddx5*-ablated spermatogonia and *Ccnb2* was strongly downregulated compared with control spermatogonia. Given its central importance in G2/M transition, we propose that decreased *Ccnb2* transcript stability is a main contributor to G2/M cell cycle arrest we observe in *Ddx5*-ablated spermatogonia.

Notably, we found that DDX5 interacts with ILF3, a post-transcriptional regulator of cyclin mRNA transcripts[57,58,62], and demonstrated that ILF3 associates with cyclin mRNAs in spermatogonia. While expression of cyclins at the mRNA level appeared unaffected upon *Ilf3* deletion, CCND1 protein was substantially reduced and cell growth inhibited. This indicates that ILF3 is involved in post-transcriptional regulation of *Ccnd1* and plays a role in cell cycle regulation alongside DDX5. A specific role for ILF3 has not been described in spermatogonia; however, its role in cyclin regulation and cell cycle progression has been shown in other systems, with ILF3 acting independently or in concert with binding partners ILF2 and IGF2BP3[57–59,62].

Our data indicates that DDX5 is present in a complex with ILF3, ILF2, and IGF2BP3 and that together they bind a number of cyclin mRNA transcripts. However, whether this complex is absolutely required for regulation of expression of these mRNAs remains unclear as independent ablation of *Ddx5* and *Ilf3* did not produce overlapping effects on gene expression, with the exception of *Ccnd1*. Both DDX5 and ILF3 evidently regulate *Ccnd1* post-transcriptionally, although the role of DDX5 appears more significant. Given findings of this study and previously described functions of DDX5 and ILF3, our observation of increased or unchanged cyclin mRNA expression yet decreased protein levels suggests disruption of mRNA nuclear export or translation mechanisms. Defects in translation of cyclin mRNAs may be unlikely, as both DDX5 and ILF3 appear exclusively nuclear within spermatogonia. However, DDX5 and ILF3 evidently function through different post-transcriptional mechanisms; therefore, further studies are required to define the significance of DDX5-ILF3 interaction and precise post-transcriptional mechanisms by which ILF3 controls cyclin expression in spermatogonia.

Besides its known roles in RNA processing, previous studies have identified DDX5 as a co-factor of transcriptional regulators[21,22]. The ability of DDX5 to interact with PLZF indicated that DDX5 is a transcriptional co-factor for PLZF in spermatogonia. Through ChIP-seq experiments, we found that DDX5 is present at a number of gene promoters overlapping with previously identified PLZF binding sites[16], including that of *Ilf3*. Individual loss of *Ddx5* and *Plzf* in spermatogonia resulted in reduced *Ilf3* expression, suggesting a co-ordinated role in regulation of *Ilf3* expression. We propose that PLZF/DDX5-dependent expression of *Ilf3* is required in part to maintain proliferative capacity of undifferentiated spermatogonia through post-transcriptional regulation of cell cycle-related genes such as *Ccnd1*. Interestingly, despite robust interaction between PLZF and DDX5 in cultured undifferentiated spermatogonia, DDX5 is only recruited to a small subset of PLZF target genes. Molecular mechanisms mediating selective association of DDX5 with PLZF at specific target genes remain unclear and DDX5 recruitment to PLZF-target gene promoters was not strictly dependent on PLZF, suggesting its recruitment is mediated through other factors. For example, DDX5 can interact with RNA polymerase II and is suggested to function as a link between transcriptional and post-transcriptional processes[75]. Notably, we found RNA polymerase subunits within our list of DDX5 interacting proteins identified by IP-MS, suggesting this mechanism may be conserved in spermatogonia. This suggests that the described association of DDX5 with multiple transcriptional regulators across distinct

systems may occur due to a broader role in transcription initiation and/or elongation. Accordingly, DDX5 (in association with DDX17) is important for formation of the transcription initiation complex (including recruitment of RNA polymerase II) at promoters of skeletal muscle-related genes[27]. It is conceivable that this mechanism is conserved in other cell systems and dependent on tissue-specific transcriptional regulators that bind proximal promoter elements. Although PLZF was found unnecessary for recruitment of DDX5 to shared target gene *Ilf3*, such recruitment may be dependent on other transcriptional regulators that are yet to be characterised. Motif analysis of our DDX5 ChIP-seq dataset suggests that E-box binding proteins are likely candidates and numerous E-box binding motifs are found within the *Ilf3* proximal promoter. Nevertheless, our data indicate that PLZF and DDX5 interact at the *Ilf3* promoter and are both required for its appropriate expression in spermatogonia. Given that transcription regulation-independent roles for PLZF have been described[76], it is also possible that the interaction of PLZF with DDX5 is relevant for non-transcriptional PLZF functions. Notably, the majority of genes dysregulated upon loss of *Ddx5* in spermatogonia are not bound by either DDX5 or PLZF and are not differentially spliced according to our analyses. This indicates that loss of *Ddx5* results in many indirect transcriptional consequences. The even distribution of upregulated and downregulated genes upon *Ddx5* loss in SPCs precludes a global defect in mRNA stability.

The TFEB transcription factor is a master regulator of lysosome biogenesis and autophagy and binds target promoters through the CLEAR sequence motif[64]. Our ChIP-seq analysis found DDX5 bound to promoters of multiple TFEB target genes in cultured undifferentiated spermatogonia and the CLEAR sequence element was enriched at DDX5 binding sites. However, we were unable to confirm a specific role for DDX5 as a co-factor of TFEB and transcriptional regulator of autophagy-related genes. One exception could be *Amdhd2*, a PLZF target gene that was co-bound by DDX5 and is a candidate TFEB target[77]. *Amdhd2* is involved in amino sugar metabolism but despite potential regulation by TFEB, specific roles in autophagy or lysosome function are unexplored[78]. Moreover, while *Amdhd2* was significantly downregulated upon *Ddx5* deletion, whether DDX5 promoted *Amdhd2* expression by acting as a co-factor of TFEB rather than PLZF remains to be determined. Notably, we found that DDX5 regulates splicing of the autophagy-associated gene *Becn1* suggesting DDX5 may play a multifaceted role in autophagy. These findings can form the basis of future studies investigating additional mechanisms of autophagy regulation.

Considering our data demonstrating the role of DDX5 in splicing, mRNA export and transcript stability, as well as recruitment to multiple gene promoters, we propose a model whereby DDX5 maintains function of undifferentiated spermatogonia by regulating gene expression programmes through multiple distinct mechanisms (Fig. 8). Specifically, that DDX5 plays essential roles in post-transcriptional gene regulation as a *trans*-acting mediator of splicing and through regulation of mRNA stability and export independently and via interactions with other RNA-binding proteins such as ILF3. Furthermore, we propose that DDX5 and PLZF co-regulate transcription of *Ilf3*, thereby maintaining cell cycle progression through downstream effects on cyclin mRNA transcripts (Fig. 8). However, given that DDX5 recruitment to the *Ilf3* promoter does not appear to be dependent on PLZF, further study is required to confirm significance of the DDX5-PLZF interaction or identify other transcriptional regulators mediating DDX5 recruitment.

In conclusion, our study provides evidence of an essential role for *Ddx5* during in vivo spermatogenesis and in maintenance of spermatogonia. Moreover, through a combination of MS and

ChIP-seq approaches, we provide evidence for undefined transcriptional roles and identify DDX5 molecular targets. Altogether, our findings present DDX5 as a multidimensional regulator of gene expression in spermatogonia and an essential factor for male fertility. Given that mammalian spermatogenesis is an ideal model for the study of adult stem cells, an understanding of the important factors and mechanisms governing this process may have implications beyond reproductive biology.

## Methods

**Mouse maintenance and treatment**. *Ddx5*flox/flox and *Plzf* −/− mice were established in previous studies[11,39]. UBC-Cre^ERT2 mice[38] were obtained from the Jackson Laboratory. For in vivo gene deletion experiments, 8-week-old to 12-week-old male mice were injected with 2 mg tamoxifen (Sigma-Aldrich) dissolved in a 90% sesame oil (Sigma-Aldrich)/10% ethanol vehicle solution for 2 consecutive days[12]. All animal experimentation was conducted in accordance with institutional ethical requirements and approved by the Monash University Animal Ethics Committee.

**Whole mount and frozen section immunofluorescence**. For immunofluorescence staining of whole mount samples, testes were detunicated and seminiferous tubules washed in ice cold phosphate-buffered saline (PBS). Tubules were fixed with 4% paraformaldehyde (PFA) for 6 h and washed with PBS prior to blocking with 10% v/v foetal bovine serum (FBS) (In Vitro Technologies), 2% w/v bovine serum albumin (BSA) (Sigma-Aldrich) and 0.3% v/v Triton X-100 (Sigma-Aldrich) in PBS. Tubules were incubated overnight at 4 °C with primary antibodies diluted in PBS supplemented with 0.3% v/v Triton X-100 and 1% w/v BSA. Tubules were washed with PBS supplemented with 0.3% v/v Triton X-100. Tubules were then incubated for one hour at room temperature with Alexa Fluor-conjugated secondary antibodies diluted in PBS supplemented with 0.3% v/v Triton X-100 and 1% w/v BSA.

To prepare frozen sections, testes were fixed in 4% PFA for 24 h at 4 °C, washed in PBS and incubated in 30% w/v sucrose solution for 48 h at 4 °C. Testis samples were placed in Optimum Cutting Temperature compound (OCT) (Sakura Finetek) and frozen at -80 °C then sectioned using a cryostat. For frozen section immunofluorescence, 8 μm thick testis cross-sections were washed in PBS and blocked for 1 h at room temperature with 10% v/v FBS and 2% w/v BSA in PBS (block solution). Sections were incubated overnight at 4 °C with primary antibodies diluted in block solution. Sections were then washed in PBS and incubated for 1 h at room temperature with Alexa Fluor-conjugated secondary antibodies diluted in block solution. For both whole mount and frozen section samples, DAPI counterstain was added during secondary antibody incubations. Samples were then washed and mounted in VectaShield mounting medium (Vector Labs) and coverslipped. For human testis sections, standard paraffin histology processing with heat-mediated antigen retrieval was performed prior to immunofluorescence[79]. TO-PRO-3 Iodide (Thermo Fisher Scientific) was used as a DNA counterstain. All immunofluorescence analyses were performed using a Zeiss LSM780 confocal microscope (Carl Zeiss) at the Monash Micro Imaging facility. All image processing and analyses were performed using ImageJ. Primary antibodies were are follows: goat anti-DDX5 (1:250, Abcam, #ab10261); rabbit anti-DDX5 (1:500, Abcam, #ab21696); goat anti-GFRa1 (1:250, R&D Systems, #AF560); rat anti-GILZ (1:1000, eBioscience, #14-4033-82); goat anti-PLZF (1:500, R&D Systems, #AF2944); goat anti-c-KIT (1:250, R&D Systems, #AF1356); rabbit anti-ILF3 (1:500, Abcam, #ab92355); goat anti-SOX3 (1:250, R&D Systems, #AF2569); rabbit anti-VASA (1:250, Cell Signalling Technology, #8761); rabbit anti-SOX9 (1:1000, Millipore, #AB5535); rabbit anti-SCP3 (1:500, Abcam, #ab15093). All Alexa Fluor secondary antibodies against appropriate species were raised in donkey and used at 1:500 dilution (Thermo Fisher Scientific).

**Cell culture and immunofluorescence**. Murine undifferentiated spermatogonia cultures were established from 6-week-old mice following enzymatic digestion of whole testis extracts with collagenase and trypsin[12]. Undifferentiated spermatogonia were cultured on mitomycin-inactivated mouse embryonic fibroblast (MEF) feeder cells and maintained in StemPro-34 medium (Thermo Fisher Scientific) supplemented with 10 ng/ml GDNF, 10 ng/ml bFGF, 20 ng/ml EGF, 25 μg/ml insulin plus other additives[13]. For culture of *Ddx5*flox/flox; UBC-Cre^ERT2 MEFs, embryos were collected at E13.5 and MEFs isolated according to a standard protocol[80]. MEFs were maintained in DMEM (Thermo Fisher Scientific) supplemented with 10% v/v FBS, 1X non-essential amino acids (Thermo Fisher Scientific), 1× L-glutamine (Thermo Fisher Scientific) and 1× Penicillin/Streptomycin (Thermo Fisher Scientific). For gene knockout experiments, cells were treated with medium supplemented with 0.2 μM 4-hydroxytamoxifen (4-OHT) (Sigma-Aldrich) for four consecutive days. Following treatment, 4-OHT-supplemented medium was replaced with standard culture medium until the required collection time point. For subsequent analyses, cells were collected by dissociation with 0.5% v/v trypsin-EDTA (Thermo Fisher Scientific).

Immunofluorescence on cultured cells was performed using LabTek chamber slides coated with Geltrex Matrix (Thermo Fisher Scientific)[12]. Antibodies used are

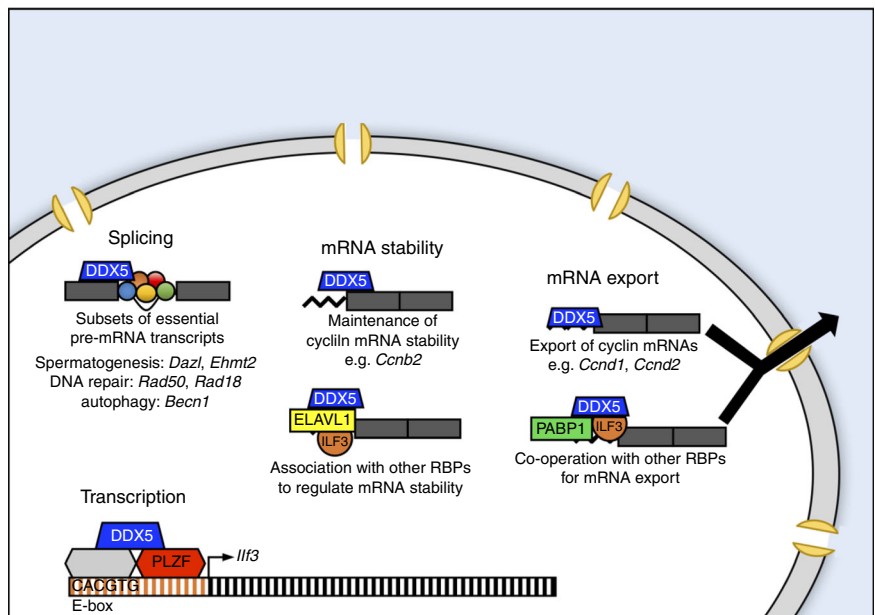

**Fig. 8** Model of DDX5-mediated gene regulation in undifferentiated spermatogonia. DDX5 plays an essential role in the post-transcriptional regulation of gene expression within undifferentiated spermatogonia. DDX5 is required for the appropriate splicing of key genes involved in spermatogenesis, DNA repair and autophagy, and is important for cell cycle regulation through maintenance of cyclin mRNA transcript stability and nuclear export. Post-transcriptional regulation by DDX5 is likely mediated through interactions with other RNA-binding proteins in a context-dependent manner. DDX5 also plays a transcriptional role, interacting with PLZF (and other potential binding partners such as E-box binding proteins) to co-regulate transcription of *Ilf3*, thereby maintaining cell cycle progression through downstream effects on cyclin mRNA transcripts

described above (see Whole mount and frozen section immunofluorescence), as well as: rabbit anti-cleaved CASP3 (1:500, Cell Signalling Technology, #9579); rabbit anti-SALL4 (1:2000, Abcam, #ab29112); chicken anti-GFP (1:5000, Abcam, #ab13970); rabbit anti-mCherry (1:2000, Abcam, #ab1647453); rabbit anti-DDX17 (1:250, Bethyl Laboratories, #A300-509A); and rabbit anti-RAD50 (1:250, Abcam, #ab124682).

**Apoptosis inhibitor experiments**. Undifferentiated spermatogonia were cultured and treated with 0.2 μM 4-OHT as described above (see Cell culture and immunofluorescence). To inhibit apoptosis, Z-VAD(OMe)-FMK (Z-VAD-FMK) (Cell Signalling Technology) reconstituted in DMSO was added at a final concentration of 50 μM in conjunction with 4-OHT treatment. An equivalent amount of DMSO was used as a vehicle control. Following 4-OHT treatment, medium was replaced with standard culture medium supplemented with 50 μM Z-VAD-FMK until collection for further analyses or fixation for immunofluorescence.

**Co-immunoprecipitation and western blot**. Cultured wildtype murine undifferentiated spermatogonia were used for all co-immunoprecipitation experiments. Co-immunoprecipitation was performed using protein G sepharose beads (GE Healthcare) according to manufacturer's instructions and western blots performed according to standard western blotting protocols[13,14]. Antibodies used for immunoprecipitation were as follows: Armenian hamster anti-PLZF (clone 13G10)[13] and goat anti-DDX5 (Abcam, #ab10261). Ten micrograms of antibody was used for each experiment. For western blots, antibodies used were as follows: Armenian hamster anti-PLZF (clone 9E12, 1:2000);[14] rabbit anti-DDX5 (1:2000, Cell Signalling Technology, #98775); rabbit anti-ILF3 (1:10000, Abcam, #ab92355); rabbit anti-ELAVL1 (1:1000, Abcam, #ab200342); rabbit anti-PABP1 (1:1000, Cell Signalling Technology, #4992); mouse anti-SRSF3 (1:2000, Sigma-Aldrich, clone 2D2); rabbit anti-ILF2 (1:1000, Abcam, #ab154791); rabbit anti-IMP3 (1:1000, Abcam, #ab177477); and rabbit anti-TFEB (1:1000, Bethyl Laboratories. #A303-673A). Horseradish peroxidase conjugated secondary antibodies were used (GE Healthcare and Jackson ImmunoResearch) in combination with Pierce ECL western blotting substrate (Thermo Fisher Scientific) for detection.

**Mass spectrometry and gene ontology analysis**. Lysates from cultured undifferentiated spermatogonia were prepared using lysis buffer containing 50 mM Tris-HCl pH 8.0, 150 mM NaCl, 5 mM EDTA pH 8.0, 0.5% v/v Nonidet P-40 (Sigma-Aldrich), 1 mM dithiothreitol (Sigma-Aldrich), and protease inhibitor cocktail (Roche)[76]. DDX5 and PLZF complexes were immunoprecipitated with goat anti-DDX5 antibody (Abcam, #ab10261) and goat anti-PLZF antibody (R&D Systems, #AF2944), respectively, coupled with Dynabeads using a Dynabead Coupling Kit (Thermo Fisher Scientific) according to manufacturer's instructions. Elution and

preparation of samples for mass spectrometry analysis was performed as described previously at the Bio21 Proteomics Facility[12]. Gene ontology analysis was performed on mass spectrometry data with PANTHER[43] v13.0 using PANTHER Overrepresentation Test (Fisher's Exact with FDR multiple test correction, FDR <0.05) and complete GO annotations for molecular function and biological process.

**Fluorescence-activated cell sorting**. Fluorescence-activated cell sorting (FACS) purification was performed on cultured cells to remove contaminating feeder cells prior to RNA extraction or subsequent analyses or western blot. Following culture and collection as described above, samples were washed in 2% v/v FBS in PBS and stained with phycoerythrin (PE)-conjugated EpCAM (1:300, eBioscience, clone G8.8) for 30 min at 4 °C prior to washing and resuspension in 2% v/v FBS in PBS. DAPI was added to final cell suspensions for live/dead cell discrimination. Spermatogonia were purified according to their expression of EpCAM[10]. For cultured cells expressing constructs tagged with GFP or tdTomato, spermatogonia were isolated based on endogenous fluorescence of these tags, without staining for EpCAM. Isolation of spermatogonia from testes of control and *Plzf* −/− mice was performed following staining with allophycocyanin (APC)-conjugated anti-cKIT (Biolegend, clone 2B8, #105812), PE-conjugated anti-CD49f (Biolegend, clone GoH3, #313612), PE-Cyanin7-conjugated anti-EpCAM (Biolegend, clone G8.8, #118216) and fluorescein isothiocyanate (FITC)-conjugated anti-CD9 (eBioscience, clone KMC8, #11-0091-82)[12]. FACS was performed at Monash University Flow-Core using a BD Influx Cell Sorter (BD Biosciences).

**RNA extraction and RT-qPCR**. RNA was isolated using TRIzol reagent (Thermo Fisher Scientific) and purified using a Direct-zol MiniPrep kit (Zymo Research) according to the manufacturer's instructions, including treatment with DNase I. cDNA was synthesised using Tetro cDNA synthesis kit (Bioline). qPCR was performed using SYBR Premix Ex Taq II (Takara) and run on a Light Cycler 480 (Roche) or Mic qPCR Cycler (Bio Molecular Systems). Primer sequences used were obtained from the Harvard PCR Primer Bank (http://pga.mgh.harvard.edu/primerbank/) and are included in Supplementary Table 1. Differences between conditions were calculated using the ΔΔCt method.

**RNA-sequencing and analysis**. RNA was isolated from FACS-purified cultured spermatogonia as described above. RNA quality was evaluated using Bioanalyzer and RNA quantity was determined using Qubit. Samples were prepared starting with 1 μg of total RNA and libraries generated according to the Illumina Stranded mRNA protocol using 12 cycles of amplification. Library size (~350 bp) was checked using Bioanalyzer and quantified with Qubit and qPCR. Based on qPCR results, a single equimolar pool was prepared and denatured with a final

concentration of 200 pM of library pool clustered in one lane of an Illumina HiSeq 3000 (100 bp paired-end) 8-lane flow cell using c-Bot. RNA-sequencing was performed at the Medical Genomics Facility, Monash Health Translation Precinct (MHTP). Raw sequencing reads in FASTQ format were filtered and trimmed with Trimmomatic[81] (v0.36, Phred score of 6 consecutive bases below 15, minimum read length of 36 nt), mapped to the complete mouse genome (GENCODE GRCm38 primary, annotation version vM15) with STAR[82] (v2.4.2a) followed by gene read counting with featureCounts[83] (v1.5.2, reverse stranded). Only genes with more than 5 sequencing reads and 2 counts per million of mapped reads in at least 4 library-sized normalised samples were considered for further analysis. Differential gene expression was performed with Degust webtool (http://degust.erc.monash.edu/) using *limma-voom*[84,85]. Genes with a false discovery rate (FDR) <0.05 were considered to be differentially expressed. Pathway analysis was performed with Ingenuity Pathway Analysis (IPA) (QIAGEN Inc., https://www.qiagenbioinformatics.com/products/ingenuity-pathway-analysis) using all differentially expressed genes (FDR <0.05).

Differential splicing analysis was performed using the MISO v0.5.4 probabilistic framework[45] (in the configuration where *filter_results* = TRUE) and the mm10 alternative event annotation (v2.0). For each alternative splicing event type, pooled treatment replicates were compared with combined control replicates using *miso_compare* and results filtered using *filter_events* (num-inc 1, num-exc 1, num-sum-inc-exc 10, delta-psi 0.20, bayes-factor 10).

**PCR validation of differential splicing analysis**. RNA was extracted from vehicle-treated and TAM-treated *Ddx5*[TAM-KO] cultured undifferentiated spermatogonia and cDNA synthesised as described above (see Cell culture and immunofluorescence and Apoptosis inhibitor experiments). Primers for validation of differentially spliced exons detected by MISO were designed using Primer3 (http://primer3.ut.ee). Primers were designed within constitutive exons flanking the exon of interest. Standard PCR for analysis by gel electrophoresis was performed using JumpStart REDTaq ReadyMix Reaction Mix (Sigma-Aldrich) according to the manufacturers' instructions and visualised by running on a 2% agarose gel. Primer sequences used are found in Supplementary Table 1. Uncropped gels are presented in Supplementary Fig. 12.

**Nuclear-cytoplasmic fractionation**. Vehicle-treated and TAM-treated *Ddx5*[TAM-KO] cultured undifferentiated spermatogonia were treated with Z-VAD-FMK as described above (see Cell culture and immunofluorescence and Apoptosis inhibitor experiments). Following dissociation with trypsin, cells were feeder depleted according to differential adhesion to cell culture dishes. For nuclear-cytoplasmic fractionation and RNA extraction, dissociated spermatogonia were pelleted by centrifugation and resuspended in PBS once to wash, then centrifuged again and supernatant removed. Cells were lysed by resuspension in hypotonic lysis buffer containing 10 mM HEPES pH 7.4 (Thermo Fisher Scientific), 10 mM NaCl, 3 mM $MgCl_2$ and 0.313% IGEPAL CA-630 (Sigma-Aldrich) in DNase/RNase-free deionised water. Samples were incubated for 5 min on ice and centrifuged for 8 min at $2000 \times g$ at 4 °C. The supernatant containing the cytoplasmic fraction was collected and the nuclear pellet was washed once in hypotonic lysis buffer before centrifuging again for 5 min at $2000 \times g$ at 4 °C. The supernatant was removed and resulting nuclear pellet and previously collected cytoplasmic fraction were processed for RNA extraction as described above (see RNA extraction and RT-qPCR). Nuclear-cytoplasmic ratios were calculated using the $\Delta\Delta$Ct method.

**mRNA stability assays**. Vehicle-treated and TAM-treated *Ddx5*[TAM-KO] cultured undifferentiated spermatogonia were treated with Z-VAD-FMK as described above (see Cell culture and immunofluorescence and Apoptosis inhibitor experiments). For mRNA stability assays, cells were treated with 10 μM actinomycin-D (Sigma-Aldrich) for up to four hours, with cell samples collected at 0, 2, and 4 h post-treatment. At each time point and following dissociation with trypsin, cells were feeder depleted according to differential adhesion to cell culture dishes prior to RNA extraction, cDNA synthesis and RT-qPCR analysis as described above (see RNA extraction and RT-qPCR).

**Western blotting**. For western blots, cells were lysed (following FACS purification for spermatogonia) and processed according to standard western blotting protocols[14]. For western blot analysis of *Ddx5*[TAM-KO] cultured spermatogonia, cells (including control samples) were treated with Z-VAD-FMK as described above in order to enough material for protein analysis (see Apoptosis inhibitor experiments). Antibodies used were as described above (see Co-immunoprecipitation and western blots), as well as: rabbit anti-DDX17 (1:1000, Bethyl Laboratories, #A300-509A); rabbit anti-CCND1 (1:1000, Novus Biologicals, #NB600-584); rabbit anti-CCND2 (1:1000, Santa Cruz Biotechnology, sc-593); mouse anti-Cyclin A (1:500, Sigma-Aldrich, #C4710); mouse anti-β-actin (1:2000, Sigma-Aldrich, #A2228); rabbit anti-RAD50 (1:1000, Abcam, #ab124682); and rabbit anti-ILF3 (1:10000, Abcam, #ab92355). Uncropped blots are presented in Supplementary Fig. 12.

**Cell cycle analysis and flow cytometry**. Analysis of cell cycling by flow cytometry was performed using Click-iT Plus EdU Pacific Blue Flow Cytometry kit (Thermo Fisher Scientific) according to manufacturer's instructions[12]. EdU incorporation was performed under standard cell culture conditions for 3 h. Cell fixation with 4% PFA and methanol permeabilisation was performed as described previously[12,17]. Intracellular staining was performed with Alexa 647-conjugated anti-PLZF antibody[13]. Preparation of whole testis extracts to isolate cells for flow cytometry analyses was performed as described previously[12,13,17]. Cells were stained with Alexa 647-conjugated anti-PLZF antibody[13], PE-conjugated anti-CD117 (c-KIT) antibody (1:250, eBioscience, clone 2B8, #12-1171-82) and goat anti-DDX5 (1:250, Abcam, #ab10261) with donkey anti-goat Alexa Fluor 488 (1:500, Thermo Fisher Scientific, #A-11055). Cells were analysed at Monash University FlowCore using an LSR Fortessa Flow Cytometer (BD Biosciences) and data processed with FlowJo software.

**Cloning of *Ddx5* constructs and lentiviral transduction**. DDX5 wildtype and DDX5 helicase mutant constructs were generated and described in a previous study by Bates et al.[42]. DDX5 constructs were subcloned into a PCCL-hPGK vector[86] modified to express a tdTomato fluorescent protein tag. First, the *P2A-tdTomato* sequence from pCDH1-EF1-Luc2-P2A-tdTomato vector (a gift from Kazuhiro Oka, Addgene plasmid #72486) was amplified by PCR using complimentary primers that included a BglII, BamHI and SalI recognition site at the 5′ end and XhoI recognition site at the 3′ end (F: 5′-GAA GAT CTG GAT CCG TCG ACG AAA GCG GAG CTA CTA A-3′; R: 5′-CGC CGC TCG AGT TAT TAC TTG TAC AGC TCG TC-3′). The PCR product was cloned by BglII/XhoI restriction digestion and ligation into BamHI/SalI sites of the PCCL-hPGK vector. As a result, original BamHI and SalI sites present in the PCCL-hPGK vector were destroyed and new BamHI and SalI sites created preceding *P2A-tdTomato* (new vector referred to as PCCL-hPGK-P2A-tdTomato). DDX5 constructs were then amplified by PCR using complimentary primers that included a BglII recognition site at the 5′ end and SalI site at the 3′ end (F: 5′-GGA AGA TCT GCC ATG GAA CAA AAA CTC ATC TCA G-3′; R: 5′-GCC GAC GTC GAC TTG GGA ATA TCC TGT TGG C-3′). The resulting PCR product was cloned by BglII/SalI restriction digest and ligation into the BamHI/SalI sites of PCCL-hPGK-P2A-tdTomato vector. All PCR, digestion and ligation reactions were performed using appropriate commercially available kits (New England Biolabs). PCR product purification was performed using PureLink PCR Purification kit (Thermo Fisher Scientific). All plasmid transformations were performed using One Shot TOP10 chemically competent *E. coli* according to manufacturer's instructions (Thermo Fisher Scientific) and colonies were selected based on ampicillin resistance. All DNA purification was performed using PureLink HiPure Plasmid Midiprep kits (Thermo Fisher Scientific). Sanger sequencing was performed to confirm successful cloning of each construct. For lentivirus production, HEK293T cells were transfected with cloned DDX5 constructs (10 μg) and packaging plasmids pMDL (5 μg), pRSV-rev (2.5 μg) and pVSV-G (3 μg) using Lipofectamine-2000 (Thermo Fisher Scientific). Lentiviral-containing supernatant was collected in spermatogonia culture medium 48 and 72 h after transfection. Undifferentiated spermatogonia cultures were infected with lentiviral-containing supernatant supplemented with 8 μg/ml polybrene (Millipore). Successfully infected cells were selected by FACS according to expression of tdTomato (see Fluorescence-activated cell sorting purification) and re-plated for subsequent analyses. Cells infected with PCCL-hPGK-tdTomato were used for control experiments.

**ChIP-sequencing and motif analysis**. Chromatin immunoprecipitation was performed using digested chromatin from two independently derived lines of cultured wildtype undifferentiated spermatogonia prepared with SimpleChIP Enzymatic Chromatin IP Kit (Cell Signalling Technology #9003)[12,14]. The antibody used for immunoprecipitation was: goat anti-DDX5 (Abcam ab10261 Lot# GR183991-1). Samples were quantified by Qubit and 20 ng of DNA was used as starting amounts for all libraries. Samples were sheared by Covaris to ~200–300 bp prior to library generation. Libraries were constructed using Ovation Ultralow System V2 using Nugen protocol M01379v1, 2014. Libraries were initially quantified by Qubit and size profile determined using Bioanalyzer followed by quantification by qPCR and pooled in equimolar ratio. Upon qPCR, 200 pM of a library pool was used for cluster generation with c-bot (Ilumina Protocol 15006165 v02 Jan 2016). Two lanes of Illumina HiSeq3000 (Ilumina Protocol 15066493 Rev A, February 2015) were used for 50 bp single end reads. ChIP-sequencing was performed at the Medical Genomics Facility, Monash Health Translation Precinct (MHTP) and data was processed by the Monash University Bioinformatics Platform. Raw reads from the ChIP-seq libraries were mapped to the UCSC mouse genome (mm10) using the bowtie2 tool. Good alignment rates were obtained across all libraries with over 97% mapped reads. Unmapped and low mapping quality reads were filtered out using samtools. ChIP-seq analysis was performed using the HOMER tool (http://homer.ucsd.edu/homer/). The first step of the HOMER ChIP-seq analysis creates a tag directory for each ChIP-seq library that generates quality control matrices and bedGraph files. Peak identification was performed with HOMER's *findPeaks* function using histone parameters. Peak annotation of individual samples and peaks common between 2 samples was performed with HOMER *annotatePeaks.pl*. Peak annotation determines the distance to the nearest Transcription Start Site (TSS) and assigns the peak to that gene followed by genomic annotation of the

region occupied by the centre of the peak. The list of peaks and associated genes are listed in Supplementary Data 8. Data is deposited in the GEO repository (GSE110347). Motif analysis was performed on ChIP-seq data using the RSAT peak-motifs tool (https://rsat01.biologie.ens.fr/rsat/peak-motifs_form.cgi).

**Plzf knockdown and ChIP-qPCR.** Enzymatically dissociated wildtype undifferentiated spermatogonia were seeded on to a mitotically-inactivated MEF feeder layer at $1 \times 10^6$ cells per well in a 6-well tissue culture plate with 2 ml final volume of culture medium (see Cell culture and immunofluorescence). siRNA knockdown was performed immediately following passage using ON-TARGETplus Mouse Zbtb16 siRNA SMARTpool (Horizon Dharmacon, #L-040219-01-0005) or control ON-TARGETplus Non-targeting Pool (Horizon Dharmacon, #D-001810-10-05). For transfection (per well), 150 µl Opti-MEM Reduced Serum Media (Thermo Fisher Scientific) was combined with 9 µl Lipofectamine RNAiMAX Reagent (Thermo Fisher Scientific) in one tube and 150 µl Opti-MEM Reduced Serum Media (Thermo Fisher Scientific) was combined with 60 pmol siRNA in a separate tube. Tubes were combined and incubated for 5 min at room temperature to form siRNA-lipid transfection complexes. After 5 min, 300 µl transfection complex mixture was added to each well. Cells were incubated for 18 h at 37 °C, 5% $CO_2$ then culture medium was replaced. Cells were incubated for an additional 24 h before collection for ChIP experiments. $4 \times 10^6$ cells were used for each ChIP. Knockdown efficiency was confirmed by western blot. Chromatin immunoprecipitation was performed using digested chromatin prepared using SimpleChIP Enzymatic Chromatin IP Kit (Cell Signalling Technology #9003)[12,14]. The antibody used for immunoprecipitation was: goat anti-DDX5 (Abcam ab10261 Lot# GR183991-1). Purified eluted DNA was used for subsequent qPCR analysis using primers designed within peak regions identified by HOMER as described above (see ChIP-sequencing and motif analysis), and control primers designed within intergenic regions upto 2 kb upstream of identified peaks. Primers were designed using Primer3 and are listed in Supplementary Table 2.

**RNA immunoprecipitation.** Cultured wildtype murine undifferentiated spermatogonia were used for all RNA immunoprecipitation (RNA-IP) experiments. Cells were cultured and collected as described above (see Cell culture and immunofluorescence). RNA immunoprecipitation was performed using an Imprint RNA Immunoprecipitation Kit (Sigma-Aldrich) according to the manufacturer's insructions. Five micrograms of antibody or rabbit IgG was used for immunoprecipitation. Antibodies used were: rabbit anti-DDX5 (Abcam, #ab21696) and rabbit anti-ILF3 (Abcam, #ab92355). RNA extraction was performed following RNA-IP using TRIZol (Thermo Fisher Scientific) and purified with Direct-zol Miniprep Kit (Zymo Research). cDNA synthesis was performed on total extracted RNA using Tetro cDNA Synthesis Kit (Bioline). qPCR was performed using SYBR Premix Ex Taq II (Takara) and run on a Mic qPCR Cycler (Bio Molecular Systems). Using cycle threshold values, fold enrichment versus IgG only pull-down was calculated for candidate genes. Standard PCR for analysis by gel electrophoresis was performed as described above (see PCR validation of differential splicing analysis). Primer sequences used are found in Supplementary Table 1. Uncropped gels are presented in Supplementary Fig. 12.

**Generation of inducible Ilf3⁻/⁻ spermatogonia by CRISPR/Cas9.** CRISPR/Cas9 targeting mouse Ilf3 was performed using the inducible lentiviral system described by Aubrey et al.[63]. sgRNAs to mouse Ilf3 were designed by CCTop software (https://crispr.cos.uni-heidelberg.de/). One sgRNA (5′-TCCCGTTACTGCCGTCTACCTCCA-3′) targeting an exonic sequence common to all known transcript variants of Ilf3 was selected. Oligonucleotides containing this sgRNA sequence were synthesised (Integrated DNA Technologies) including a 5′ "TCCC" 4 bp overhang for the complementary sequence and a ′5 "AAAC" 4 bp overhang for the reverse complementary sequence. Annealed oligonucleotides were cloned into BsmBI restriction site of the lentiviral vector for doxycycline (DOX)-inducible sgRNA expression (FgH1tUTG–eGFP expression) provided by Dr. Marco Herold. Sanger sequencing was performed to ensure sgRNA sequence was ligated into FgH1tUTG. Lentiviral production and transduction of cells were performed as described above. Cultured wildtype spermatogonia were initially infected with FUCas9mCherry and subsequently infected with the sgRNA construct. Cells were then sorted with BD Influx Cell Sorter (BD Biosciences) for mCherry (Cas9) and eGFP (sgRNA). Expression of sgRNA was induced by treatment with 2 µg/ml of doxycycline hyclate (Sigma-Aldrich D9891). Knockout of Ilf3 in spermatogonia was confirmed by immunofluorescence and qRT-PCR.

**Statistical analyses.** Assessment of statistical significance was performed using two-tailed unpaired t-tests, one-way ANOVA with Tukey multiple comparisons test or two-way ANOVA with Bonferroni multiple comparisons test for normally distributed data and two-tailed Mann-Whitney U tests or Kruskal–Wallis with Dunn's multiple comparisons tests for data that were not normally distributed. P-values less than 0.05 were considered statistically significant. All statistical analyses including normality tests were performed using GraphPad Prism v7. For mouse experiments, no statistical method was used to predetermine sample sizes and no specific randomisation or blinding methods were used.

## Data availability

PLZF and DDX5 mass spectrometry data have been deposited in the MassIVE repository under accession code MSV000083476 [https://doi.org/10.25345/C57P6V]. DDX5 RNA-seq data has been deposited in the Gene Expression Omnibus (GEO) database under accession code GSE127275 and DDX5 ChIP-seq data has been deposited in GEO under accession code GSE110347. All data are available within the article or Supplementary Information or from corresponding author upon reasonable request.

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

## Acknowledgements

We would like to thank Monash Animal Research Platform for animal care, Marco Herold for providing CRISPR/Cas9 lentiviral vectors and Dan Littman for providing mice. We also thank James Bourne for providing marmoset testis and Moira O'Bryan for human testis samples. We would like to thank Traude Beilharz, Eileen McLaughlin, and Antonella Papa for helpful discussions. We would like to thank the Melbourne Mass Spectrometry and Proteomics Facility of The Bio21 Molecular Science and Biotechnology Institute at The University of Melbourne for the support of mass spectrometry analysis. We acknowledge the facilities and technical assistance of Monash Micro Imaging, FlowCore, and Monash Health Translation Precinct Medical Genomics Facility. We would also like to acknowledge the support of Monash Bioinformatics Platform. This work was supported by the Australian Research Council (ARC) Stem Cells Australia Special Research Initiative and NHMRC Project Grant APP1078042. R.M.H. was supported by an ARC Future Fellowship. The Australian Regenerative Medicine Institute is supported by grants from the State Government of Victoria and the Australian Government.

## Author contributions

J.M.D.L., A.C., H.M.L., M-L.A. and R.M.H. conceived and designed the study. J.M.D.L., A.C., H.M.L., F.J.R. and R.M.H. performed experiments. F.V.F-P. and M-L.A. provided critical reagents. J.M.D.L., A.C., H.M.L., F.J.R. and R.M.H. analysed data. J.M.D.L., F.V.F-P. and R.M.H. wrote the paper.

## Additional information

**Competing interests:** The authors declare no competing interests.

