## [Peer Review File · Nature Communications]

Reviewers' comments:

Reviewer #1 (Remarks to the Author):

In the manuscript entitled « Ddx5 plays essential transcriptional and post-transcriptional roles in maintenance and function of germline cells », J. Legrand et al. generated a tamoxifen-inducible Ddx5 KO mice model. Ddx5 KO results in the depletion of undifferentiated spermatogonia. The authors show a physical interaction between DDX5 and PLZF, a transcription factor that is essential for self-renewal of undifferentiated spermatogonia in mice. Few (~100) genomic Ddx5-binding sites were identified by ChIP-seq. Among the DDX5-binding sites, eight could also be PLZF-binding sites (identified from a previously published PLZF ChIP seq). The *Ilf3* promoter seems to be targeted by both PLZF- and DDX5 and *Ilf3* expression is decreased in DDX5 KO mice. DDX5 is also shown to interact with the *Ilf3* protein and since *Ilf3* plays a role in mRNA export/stability, the authors hypothesize that DDX5 regulates the export and/or stability of some mRNAs. It is also shown that DDX5 interacts with many splicing factors, so the authors propose a role of DDX5 in regulating splicing in germline cells.

While the identification of an important role of DDX5 in maintenance and function of germline cells is very interesting (supported by the Ddx5 KO mice model), the analyses of the molecular functions of DDX5 in these processes are too superficial. The authors should focus on 1 or 2 molecular process(es) (e.g., transcription) and investigate more mechanistic details. For example, how does DDX5 work with PLZF? Is DDX5 recruited on promoters in a PLZF-dependent manner? More experiments are required to demonstrate that Ddx5 is a PLZF-transcriptional coregulator. Another puzzling question is why is DDX5 recruited on some autophagic gene promoters while Ddx5 KO had no effect on the expression of these genes?

The description of the GO term analysis from Ddx5 partners is too long and does not bring any new information regarding the nature of the already known DDX5 functions. The authors conclude that DDX5 plays a role in mRNA stability/export (?), which has already been reported elsewhere, but there is no further proof of that in this manuscript. The authors do not present any data showing a role of DDX5 in mRNA export or stability in germline cells. The results shown on Fig5g and 5h are quite confusing and, at this stage, it seems difficult to conclude on any clear function of Ddx5 on mRNA export and/or stability in this cellular model.

Finally, the authors observed that Ddx5 interacts with many splicing factors. They conclude from this observation that DDX5 plays a role in maintenance and function of germline cells in part because of its splicing regulatory activity (Figure 7). However, the interaction of Ddx5 with splicing factors is already well established and there is no data showing that indeed Ddx5 controls splicing of genes in germline cells. Did the authors check the effect of DDX5 KO on the splicing of *Ccna2*, *Ccnd1*, *CcnD2*...?

As it stands, the paper suffers from too many overstatements or shortcuts.

Reviewer #2 (Remarks to the Author):

The authors present a convincing case for a multifunctional and essential role for DDX5 in Spermatogonial maintenance in mice.

This is a novel finding and of interest to developmental and reproductive biologists and clinical andrologists.

The combination of inducible knockout, in vitro studies and cell lineage tracing revealed an intriguing alternative function for the RNA helicase DDX5 as a transcriptional co-regulator. This manuscript sheds light on some of the more obscure aspects of early Spermatogonial/Spermatogonial stem cell development and the authors may wish to comment on this work in that context.

Statistically valid observations have been made and as access to mice lines are readily available reproducibility of the work is feasible.

Questions for the authors

Could the authors comment on the predicted vs observed Molecular Mass of IP products in Fig 1 b ?

Did the authors confirm any other putative targets (e.g. SALL4) by IP from the mass spectroscopy lists?

Given functional redundancy in MEFs is DDX17 expressed in Spermatogonia?

CDKN1A functions primarily at the G1 checkpoint - could the authors speculate on the G2 arrest mechanism? is this likely to be an interaction with another CDKN1 e.g. CDKN1B ?

CCND1 - is a regulator of G1/S phase transition - did this disappear in the null Spermatogonia?

Reviewer #3 (Remarks to the Author):

In this study, Legrand and colleagues explored the role of an RNA helicase, Ddx5, in spermatogenesis using a variety of techniques. Based on the results, they conclude that Ddx5 plays an essential role in maintenance of the spermatogenic stem cell population via interaction with the transcription factor Plzf. At present, the role of Ddx5 and other RNA helicases in the spermatogenic lineage is undefined, thus the study addresses a gap in knowledge. Overall, the experimentation is technically sound. However, the results are not interpreted correctly, as discussed below, and should be re-assessed for accuracy rather than claiming a unique role in the stem cell compartment of the mouse male germline. Clearly, Ddx5 plays an important role in spermatogenesis, likely having a function in all germ cell subtypes and possibly Sertoli cells. This finding is unique and advances knowledge. It's unfortunate that the authors chose to over interpret the results to portray an importance in germline stem cells only rather than focus on the role of Ddx5 in spermatogenesis in general for which they have an abundance of data to support.

The authors' attestation that Plzf is essential for self-renewal for male germline stem cells is not necessarily accurate given that spermatogenesis still occurs in a good portion of seminiferous tubules of Plzf null mice. Clearly, it plays an important role but to claim an essential role is not supported by empirical evidence.

A major nuance of this study is that Ddx5 is clearly expressed by all spermatogenic cell types in mice as well as primates as demonstrated in Fig. 1 and the supplemental figure. Thus, to claim a unique role in undifferentiated spermatogonia (and presumably the rare stem cell component of this population) is not supported. Moreover, the expression of Ddx5 by Sertoli cells makes interpretation of the findings even more challenging.

The authors have made several claims of Ddx5 expression levels being different among cell types that are based on subjective observations rather than quantitative data. For example, in the supplemental data, they have claimed higher expression of Ddx5 in Plzf+ spermatogonia of human testes compared to other cells but this is based on a single image of immunofluorescent staining rather than actual measurements of Ddx5 abundance. Similarly, the claims of comparable levels of Ddx5 expression in different germ cell subtypes as presented in Fig. 1e is based solely on

subjective observations not quantitative data.

Empirical evidence supporting the claim the UBC-Cre transgene used in the studies is expressed in spermatogonia only is lacking. In fact, this same group's recent publication in Stem Cell Reports showed that the transgene is actually expressed in all spermatogonia as well as spermatocytes. Thus, using this model in the current study, Ddx5 expression was ablated in all pre-meiotic and meiotic germ cells not just spermatogonia. The phenotypic findings cannot be interpreted as a role in undifferentiated spermatogonia only as all germ cells were rendered deficient for Ddx5 following tamoxifen injection.

The data presented in Fig. 2 can only be interpreted as a role for Ddx5 in germ cells in general because it is expressed throughout the spermatogenic lineage and the Cre transgene is not expressed specifically in undifferentiated spermatogonia. Also, the claim that Ddx5 expression was not ablated from any Sertoli cells following tamoxifen injection is not supported by experimental evidence.

Data presented in Fig. 3-6 utilized cultures of spermatogonia that are a heterogeneous mix of subtypes including stem cell, progenitor, and depending on the methodology, differentiating spermatogonia. Thus, the data cannot be interpreted as Ddx5 playing a specific role in the stem cell component. Rather, the authors are encouraged to interpret the data as being relevant to all spermatogonial subtypes.

RESPONSE TO THE REVIEWERS

We would like to thank the reviewers for their constructive and positive comments that have enabled us to substantially improve our manuscript. As detailed below, we have addressed all reviewer concerns by performing additional experiments, refining previous data analyses and adjusting text to reflect these additions and changes. Our revised manuscript now includes a much more comprehensive assessment of mechanisms through which DDX5 regulates spermatogonial function and provides a detailed analysis of the role of DDX5 during spermatogenesis. We have performed additional experiments to substantiate our original conclusions, including new RNA-sequencing data comparing control and *Ddx5*-ablated spermatogonia (**Figure 3 and Supplementary Table 2**), differential splicing analysis (**Figure 4, Supplementary Figure S6 and Supplementary Table 6**), mRNA export and stability assays (**Figure 5 and Supplementary Figure S7**), further ChIP experiments (**Supplementary Figure S9**), flow cytometry and immunofluorescence (**Supplementary Figures S1-S4**). Combined, we provide a compelling dataset demonstrating a critical role for DDX5 in the maintenance of spermatogonia and male fertility. A point-by-point response to reviewers' comments is included below. We have highlighted all changes to the manuscript text in red font.

REVIEWER #1

*In the manuscript entitled « Ddx5 plays essential transcriptional and post-transcriptional roles in maintenance and function of germline cells », J. Legrand et al. generated a tamoxifen-inducible Ddx5 KO mice model. Ddx5 KO results in the depletion of undifferentiated spermatogonia. The authors show a physical interaction between DDX5 and PLZF, a transcription factor that is essential for self-renewal of undifferentiated spermatogonia in mice. Few (~100) genomic Ddx5-binding sites were identified by ChIP-seq. Among the DDX5-binding sites, eight could also be PLZF-binding sites (identified from a previously published PLZF ChIP seq). The *Ilf3* promoter seems to be targeted by both PLZF- and DDX5 and *Ilf3* expression is decreased in DDX5 KO mice. DDX5 is also shown to interact with the *Ilf3* protein and since *Ilf3* plays a role in mRNA export/stability, the authors hypothesize that DDX5 regulates the export and/or stability of some mRNAs. It is also shown that DDX5 interacts with many splicing factors, so the authors propose a role of DDX5 in regulating splicing in germline cells.*

*While the identification of an important role of DDX5 in maintenance and function of germline cells is very interesting (supported by the *Ddx5* KO mice model), the analyses of the molecular*

functions of DDX5 in these processes are too superficial. The authors should focus on 1 or 2 molecular process(es) (e.g., transcription) and investigate more mechanistic details.

We thank the reviewer for their constructive suggestions regarding our manuscript that have allowed us to substantially improve our understanding of DDX5 function in spermatogonia. We have now included a more comprehensive mechanistic analysis of the role of DDX5 in splicing, mRNA export, mRNA transcript stability and transcriptional regulation. Responses to specific concerns are below.

For example, how does DDX5 work with PLZF? Is DDX5 recruited on promoters in a PLZF-dependent manner? More experiments are required to demonstrate that Ddx5 is a PLZF-transcriptional coregulator.

In our original manuscript, we showed that DDX5 interacted with PLZF in undifferentiated spermatogonia and was present at the same location within the promoters of a subset of PLZF target genes. In this revised manuscript, from our newly included RNA-sequencing analysis we found that six of the eight originally suggested DDX5-PLZF co-regulated genes are differentially expressed upon *Ddx5* ablation (*Ilf3*, *Amdhd2*, *Arhgap12*, *Tmem55b*, *4933434E20Rik* and *Zbtb25*) (**Figure 7f,g**). However, as shown in our initial submission, only *Ilf3* was significantly downregulated upon loss of *Plzf* suggesting specific co-regulation of this gene by a DDX5-PLZF complex (**Figure 7h**). In order to determine whether recruitment of DDX5 to the *Ilf3* promoter occurred in a PLZF-dependent manner, we performed DDX5 ChIP in control versus *Plzf* siRNA-mediated knockdown spermatogonia followed by analysis (enrichment relative to input) by qPCR. We confirmed efficient knockdown of *Plzf* (**Supplementary Figure S9a**) and chose to analyse *Ilf3* and *Amdhd2* (top two downregulated DDX5-PLZF targets upon *Ddx5* loss), as well as *Lamp1* (top DDX5 binding peak by ChIP-seq and bound by PLZF but *not* differentially expressed upon *Ddx5* ablation), *Zmym6* (bound by DDX5 only and not PLZF) and *Tex13b* (bound by PLZF only¹ and not DDX5). We designed primers within peak regions identified by ChIP-seq and 1-2kb upstream of these regions as negative controls (**Supplementary Figure S9b**). By ChIP-qPCR, we found no difference in enrichment between control and *Plzf* knockdown samples, indicating that DDX5 recruitment to these gene promoters was not dependent on PLZF (**Supplementary Figure S9c** and described in **lines 444-454**). Although the relevance of DDX5-PLZF interaction as a mechanism of transcriptional control remains unclear, our data supports a model whereby DDX5 and PLZF interact at the promoter of *Ilf3* and are both required for regulation of its expression. However, our additional data indicate that recruitment of DDX5 is most likely mediated through other factors that are yet to be characterised.

Previous studies investigating skeletal myogenesis have shown that DDX5 interacts with RNA polymerase II and is required for formation of a transcription initiation complex at MYOD target

genes². In our study, we found RNA polymerase subunits amongst DDX5 interacting proteins identified by co-IP/mass spectrometry (**Supplementary Table 3**), suggesting this mechanism may be conserved in spermatogonia and other systems. DDX5 has been described as a transcriptional co-regulator of numerous factors that span multiple systems such as oestrogen receptor, androgen receptor, MYOD (myogenesis), RUNX2 (osteogenesis) and p53³. Therefore, a broader mechanism may exist where interaction of these factors with DDX5 leads to recruitment of proteins necessary for transcriptional initiation and/or elongation. Although DDX5 recruitment to gene promoters was not found to be dependent on PLZF, it is possible that other transcriptional regulators mediate DDX5 recruitment and are yet to be characterised. We postulate that these factors could include E-box binding proteins, given that motif analysis of our ChIP-seq findings indicate enrichment of the E-box binding motif (CACGTG core of the CLEAR consensus sequence) (**Figure 7b**). Interestingly, this motif is also present at multiple locations within the *Ilf3* promoter, suggesting regulation by additional transcription factors that may form part of a larger DDX5/PLZF-containing complex. Detailed discussion related to these findings are found in **lines 631-648**. We believe our data provides interesting insight into the role of DDX5 as a transcriptional regulator in spermatogonia and offers additional areas of investigation for future studies.

Another puzzling question is why is DDX5 recruited on some autophagic gene promoters while Ddx5 KO had no effect on the expression of these genes?

We agree with the reviewer that the presence of DDX5 on autophagy-related gene promoters is an intriguing finding. In our additional RNA-sequencing data, we found that autophagy-associated genes *Hexa*, *Tmem55b* and *Atp6v1h* were differentially expressed upon *Ddx5*-ablation in spermatogonia; however, these changes showed no overall trend of up- or downregulation. Given that DDX5 is described previously as a transcriptional co-regulator⁴, we postulated that DDX5 may be acting as a co-factor for the autophagy master regulator TFEB; however, we did not identify a DDX5-TFEB interaction in cultured undifferentiated spermatogonia by co-immunoprecipitation and western blot (**Supplementary Figure S8a**). Therefore, the role of DDX5 in transcriptional regulation of autophagy genes is yet to be understood fully and will be the foundation of future studies (discussed in **lines 651-663**). Interestingly, we noted that of the 101 genes bound by DDX5 according to our ChIP-sequencing data, 49 of these were differentially expressed upon *Ddx5* ablation (**Figure 7e** and **Supplementary Figure S8b,c**). Of those 49 genes, 25 were upregulated following loss of *Ddx5* with the remaining 24 downregulated. These findings suggest that DDX5 is both an activator and repressor of transcription that may function in a context dependent manner.

The description of the GO term analysis from Ddx5 partners is too long and does not bring any new information regarding the nature of the already known DDX5 functions.

We thank the reviewer for their suggestion regarding this results section. Given the additional data we now present, we have removed the detailed description of GO term analysis and integrated relevant GO term findings into appropriate sections of the revised manuscript. Specifically, a concise summary of GO term analysis is found in **lines 258-262** of the manuscript text and a summary figure displaying top enriched terms is presented in **Figure 4a**. Detailed results for GO term analysis can also be found in **Supplementary Table 4 and 5**.

Although the role of DDX5 in RNA processing is appreciated in other systems⁵⁻⁷, not all previously suggested functions of DDX5 may be important within the context of spermatogonial function and maintenance of spermatogenesis. For example, DDX5 has been described as an important regulator of microRNA biogenesis through its interactions with the Drosha complex in cancer cell lines⁸; however, we found no evidence for this role of DDX5 in spermatogonia and do not find association of DDX5 with microRNA processing machinery (**Supplementary Table 3**). In addition, DDX5 is known as a co-activator of p53 in various cancer cells, where it interacts with p53 protein and stimulates transcription of p53-target genes including *p21*⁹. Conversely, we found no evidence for this role of DDX5 in spermatogonia and provide evidence for the activation of a p53-mediated apoptotic response in *Ddx5*-ablated spermatogonia, illustrating cell type-dependent functions for DDX5.

The authors conclude that DDX5 plays a role in mRNA stability/export (?), which has already been reported elsewhere, but there is no further proof of that in this manuscript. The authors do not present any data showing a role of DDX5 in mRNA export or stability in germline cells. The results shown on Fig5g and 5h are quite confusing and, at this stage, it seems difficult to conclude on any clear function of Ddx5 on mRNA export and/or stability in this cellular model.

We appreciate the reviewer's concerns regarding our original conclusions on the role of DDX5 in mRNA export and stability. Accordingly, we have performed additional experiments and now provide data demonstrating that DDX5 is important for nuclear export of *Ccnd1* and *Ccnd2* mRNA (**Figure 5g**), as well as the maintenance of *Ccnb2* mRNA transcript stability (**Figure 5h**). These results are described in detail within manuscript text **lines 318-360**. We also show that these findings are not due to global defects in mRNA export and stability, as *Ccna2* export and *Myc* stability are unaffected by *Ddx5* ablation (**Supplementary Figure 7a,b**).

In this study, we found that loss of *Ddx5* in spermatogonia results in the significant upregulation of *Ccnd1* and *Ccnd2* mRNA expression (**Figure 5b**). Despite this upregulation of mRNA, we found

that CCND1 and CCND2 protein levels were significantly reduced in *Ddx5*-ablated spermatogonia (**Figure 5d,e**). Importantly, we found by RNA immunoprecipitation using an antibody against DDX5 followed by RT-qPCR (RIP-qPCR) that *Ccnd1* and *Ccnd2* mRNA transcripts (amongst others) were bound to DDX5 in spermatogonia (**Figure 5f**). This led us to hypothesise that DDX5 was directly regulating these mRNA transcripts in a post-transcriptional manner. We initially speculated that defects in mRNA nuclear export leading to decreased availability of mRNA for translation or defects in translation itself may account for the reduced CCND1 and CCND2 protein observed in *Ddx5*-ablated spermatogonia. As we observed DDX5 to be restricted to the nucleus of spermatogonia *in vivo* and *in vitro*, we excluded the possibility of significant translational roles and investigated nuclear export of transcripts in more detail. We therefore performed nuclear-cytoplasmic fractionation of control and *Ddx5*-ablated cultured spermatogonia and performed RT-qPCR analyses comparing nuclear versus cytoplasmic levels of *Ccnd1* and *Ccnd2* mRNA. From these results, we calculated nuclear-cytoplasmic ratios and confirmed that both *Ccnd1* and *Ccnd2* transcripts were unusually retained in the nucleus following *Ddx5*-ablation in spermatogonia (**Figure 5g** and **lines 332-341**). Consequently, less *Ccnd1* and *Ccnd2* mRNA is exported from the nucleus and available for translation resulting in decreased CCND1 and CCND2 protein and impacting cell cycle progression.

Markedly, not all cyclin mRNAs were found to have impaired nuclear export. For example, we found no significant difference in the nuclear-cytoplasmic ratio of *Ccna2* between control and *Ddx5*-ablated spermatogonia (**Supplementary Figure 7a** and **lines 342-343**). However, we found *Ccna2* mRNA expression to be downregulated upon loss of *Ddx5* in spermatogonia with a corresponding decrease in CCNA2 protein, and we found *Ccna2* mRNA bound to DDX5 by RIP-qPCR (**Figure 5b,d-f**). Given these findings, we hypothesised that *Ccna2* was being regulated by DDX5 through a different post-transcriptional mechanism, such as maintenance of mRNA stability. Therefore, we sought to assess stability of *Ccna2* upon DDX5 loss by measuring mRNA levels over time following inhibition of transcription. After treating control and *Ddx5*-ablated spermatogonia with the transcription inhibitor actinomycin D, we compared *Ccna2* mRNA levels at 2- and 4-hours post-treatment to levels prior to treatment by RT-qPCR. Expectedly, we saw a progressive decrease in *Ccna2* over time although there was no difference in mRNA levels between control and *Ddx5*-ablated conditions (**Figure 5h**). Thus, we suggest that the reduction in *Ccna2* mRNA and protein found upon *Ddx5* loss is an indirect transcriptional consequence or related to overall defects in cell cycling.

Interestingly, our new RNA-sequencing data revealed a significant reduction in *Ccnb2* expression upon *Ddx5* loss (**Figure 5b**). Furthermore, we also found *Ccnb2* mRNA bound to DDX5 (**Figure 5f**); therefore, we postulated that *Ccnb2* transcript stability may be regulated by DDX5. Accordingly, we analysed *Ccnb2* mRNA stability and confirmed significantly decreased levels of *Ccnb2* mRNA in *Ddx5*-ablated spermatogonia compared to control at 4 hours post-actinomycin D

treatment (**Figure 5h** and **lines 350-354**). These data indicate that DDX5 is important for the maintenance of *Ccnb2* transcript stability. CCNB2 is required for G2/M progression during the cell cycle¹⁰; thus, we suggest that the G2/M arrest that we observe upon *Ddx5*-ablation in spermatogonia is a result, at least in part, of impaired DDX5-mediated stabilisation of *Ccnb2*. A detailed discussion of these findings can be found in our revised manuscript (**lines 557-590**).

Finally, the authors observed that Ddx5 interacts with many splicing factors. They conclude from this observation that DDX5 plays a role in maintenance and function of germline cells in part because of its splicing regulatory activity (Figure 7). However, the interaction of Ddx5 with splicing factors is already well established and there is no data showing that indeed Ddx5 controls splicing of genes in germline cells.

We thank the reviewer for bringing this important point to our attention. In our original submission, we speculated that DDX5 plays an important role in splicing within spermatogonia based on its interaction with various splicing factors; however, as the reviewer correctly states, we did not provide additional insight into this process. In our revised manuscript, we now provide a substantial new dataset related to the role of DDX5 in splicing within spermatogonia that has resulted in the inclusion of a new figure (**Figure 4**). In our revised manuscript, we have performed differential splicing analysis using the MISO probabilistic framework¹¹ on RNA-sequencing data comparing control and *Ddx5*-ablated spermatogonia. We identified 667 differential splicing events upon loss of *Ddx5* including aberrant splicing of a number of genes involved in spermatogenesis, DNA repair and autophagy (**Figure 4c,d** and **lines 268-288**). We confirmed that candidate transcripts were bound to DDX5 by RIP-PCR (**Figure 4e**) and validated differential splicing events (exon inclusion/skipping) by PCR with primers designed in flanking constitutive exons and analysis by gel electrophoresis (**Figure 4f**). Amongst our interesting candidates, we found aberrant splicing of *Rad50* where exon 15 was significantly skipped upon *Ddx5* loss. RAD50 is important for DNA double-strand break repair and *Rad50* mutant mice have been shown previously to have spermatogenic defects resulting in sterility¹². To our knowledge, alternatively spliced isoforms lacking exon 15 have not been described. Through coding sequence analysis (**Supplementary Figure S6**), we determined that skipping of *Rad50* exon 15 introduces a premature termination codon and is likely to lead to nonsense-mediated decay of *Rad50* mRNA. This is corroborated by the significantly reduced expression of *Rad50* mRNA and protein in *Ddx5*-ablated spermatogonia (**Figure 4g,h,i** and **lines 289-296**). These findings demonstrate that DDX5 plays an important role in pre-mRNA splicing within spermatogonia and its loss results in splicing defects that are detrimental to spermatogonial maintenance and function.

It is well-known that tissue-specific regulation of splicing occurs and is essential for the maintenance of appropriate gene/isoform expression across tissues^{13,14}. Although a role for DDX5 in splicing is described in other systems^{5,15}, characterising its control of specific splicing programs within spermatogonia is essential for our understanding of gene expression regulation during spermatogenesis. Few *trans*-acting regulators of splicing with specific effects in spermatogonia have been described; therefore, we believe our study provides important novel insight in this area. An in-depth discussion of our additional findings can be found within the discussion text (**lines 505-556**).

Did the authors check the effect of DDX5 KO on the splicing of Ccna2, Ccnd1, CcnD2...?

Indeed, splice variants of *Ccnd1*, *Ccnd2* and *Ccna2* have been linked to developmental and pathological processes¹⁶⁻¹⁸. However, our RNA-sequencing and differential splicing analyses did not identify differences in the splicing of these cyclin transcripts. Rather, our analyses have identified a role for DDX5 in the nuclear export and stabilisation of specific cyclin mRNAs (see above).

As it stands, the paper suffers from too many overstatements or shortcuts.

We acknowledge the reviewer's initial concerns regarding our original submission. We sincerely hope that the additional mechanistic data that we have provided regarding the role of DDX5 in splicing, mRNA export, stability and transcription in our revised manuscript addresses these points.

REVIEWER #2

The authors present a convincing case for a multifunctional and essential role for DDX5 in Spermatogonial maintenance in mice. This is a novel finding and of interest to developmental and reproductive biologists and clinical andrologists.

The combination of inducible knockout, in vitro studies and cell lineage tracing revealed an intriguing alternative function for the RNA helicase DDX5 as a transcriptional co-regulator. This manuscript sheds light on some of the more obscure aspects of early Spermatogonial/Spermatogonial stem cell development and the authors may wish to comment on this work in that context.

Statistically valid observations have been made and as access to mice lines are readily available reproducibility of the work is feasible.

We thank the reviewer for their positive comments and suggestions regarding our manuscript, and for acknowledging the broader implications of our findings. In our manuscript, we have performed a detailed analysis of the role of DDX5 in adult spermatogonia using an inducible Cre/lox model. Certainly, we agree that our findings may provide insight into a role for DDX5 during development of the adult spermatogonial pool; however, given that we have not specifically performed *Ddx5*-ablation during developmental time points, we are cautious of making conclusions regarding spermatogonial development. Rather, we provide comprehensive evidence of the essential role of DDX5 during steady state maintenance of adult spermatogonia. Nevertheless, we predict that germline-specific *Ddx5* ablation during development would result in a similar phenotype to our observations in the adult. DDX5 is suggested to be a transcriptional co-factor for MYOD and RUNX2 during muscle and skeletal development^{2,19}, respectively; therefore, it is conceivable that DDX5 may also function in concert with similarly critical developmental regulators during establishment of the spermatogonial pool. A detailed investigation of the role of DDX5 during embryonic and postnatal development in general will form the basis of future studies. We have addressed the reviewer's additional questions in detail below.

Could the authors comment on the predicted vs observed Molecular Mass of IP products in Fig 1b?

In our reciprocal co-immunoprecipitation experiments to confirm PLZF-DDX5 interaction, our subsequent western blot analyses confirmed appropriate DDX5 and PLZF pulldowns with observed molecular weights of IP products corresponding to predicted molecular weights. We observed bands for PLZF at 75kDa (predicted at 75kDa) and DDX5 at approximately 68kDa (predicted at 68kDa). We have now added molecular weight annotations to **Figure 1b** and have described these findings within the results text (**lines 113-114**).

Did the authors confirm any other putative targets (e.g. SALL4) by IP from the mass spectroscopy lists?

From our PLZF co-immunoprecipitation (IP) and mass spectrometry experiments, we indeed found interaction with SALL4 as a positive control (**Supplementary Table 1**). We have not included IP and western blot confirmation of the PLZF-SALL4 interaction in this manuscript as we have shown this in previous studies^{20,21}. Besides demonstrating interaction of DDX5 with PLZF (**Figure 1b**), we chose to confirm interactions for our DDX5 co-immunoprecipitation experiments that covered the range of post-transcriptional regulatory processes identified by our gene ontology analyses (**Figure 4a** and **Supplementary Table 5**). We confirmed interaction of DDX5 with ELAVL1 (mRNA stability), PABP1 (mRNA stability/export) and SRSF3 (splicing/mRNA export), as well as ILF3 (mRNA stability/export) and its key partner proteins (**Figure 4b** and **lines 262-267** and **363-365**). A discussion of the implications of these interactions can be found within the manuscript text between **lines 582-616**.

Given functional redundancy in MEFs is DDX17 expressed in Spermatogonia?

In RNA-sequencing data that we now include, we found *Ddx17* to be modestly upregulated in *Ddx5*-deficient cultured spermatogonia compared to controls (**Figure 3d** and **Supplementary Table 2**). However, while DDX17 protein is readily detectable in cultured spermatogonia, it was not increased following DDX5 loss (**Figure 3c**). This contrasts with our observations in MEFs in which *Ddx5* ablation results in substantial upregulation of DDX17 at the protein level (**Figure 3b,c**). Besides this difference in modulation of *Ddx17* expression in spermatogonia and MEFs following *Ddx5* deletion, the response of these cell types to loss of DDX5 function is also very distinct. Specifically, MEFs could be maintained in culture following *Ddx5* ablation whereas rapid and significant loss of spermatogonia was observed by D2 after gene deletion (**Figure 3a,f,g**). Our data suggests that in MEFs DDX17 compensates for loss of DDX5 while in spermatogonia DDX5 is indispensable for cell function and little appreciable functional redundancy with DDX17 exists. Previous studies have suggested functional redundancy between DDX5 and DDX17 in some contexts (e.g. proliferation in HeLa cells); however, DDX5 and DDX17 are also shown to have distinct roles in other cell types with some conflicting evidence regarding redundancy in other systems (e.g. oestrogen-receptor co-activation)³. Here, we have demonstrated a critical role for DDX5 in spermatogonia that, to our knowledge, is unlike any phenotype observed in other systems upon loss of DDX5 function.

CDKN1A functions primarily at the G1 checkpoint - could the authors speculate on the G2 arrest mechanism? is this likely to be an interaction with another CDKNI e.g. CDKN1B?

As the reviewer correctly states, *Cdkn1a* is important for cell cycle regulation at the G1 checkpoint. We show G1 arrest in *Ddx5*-ablated spermatogonia (**Figure 5c**) that we conclude is a combined effect of strong *Cdkn1a* upregulation (**Figure 5b**) plus impaired nuclear export of *Ccnd1* and *Ccnd2* mRNA that results in reduced CCND1 and CCND2 protein (**Figure 5d-g**). In addition, we observed cell cycle arrest at the G2/M transition (**Figure 5c**) that we have concluded is due to aberrant expression of G2-related cyclins *Ccna2* and *Ccnb2* (**Figure 5b,d-h**).

In our study, we found that *Ccna2* mRNA and protein levels are significantly reduced upon *Ddx5* ablation in spermatogonia (**Figure 5b,d**). *Ccna2* is known to be expressed in spermatogonia and is important for both G1/S and G2/M transitions¹⁰. We found that nuclear export and mRNA stability of *Ccna2* were unaffected upon *Ddx5* loss and concluded that indirect transcriptional effects of *Ddx5* loss were responsible for the observed downregulation. However, we also found that *Ccnb2* was strongly downregulated upon *Ddx5* loss in spermatogonia (**Figure 5b**) and that *Ccnb2* mRNA was bound to DDX5 (**Figure 5f**). Upon further investigation, we found that loss of *Ddx5* results in decreased stability of *Ccnb2* mRNA (**Figure 5h**), suggesting DDX5 directly regulates stability of this transcript. *Ccnb2* is known to be important for G2/M progression¹⁰ and therefore we concluded that the G2-arrest we observed upon *Ddx5* loss in spermatogonia can be a result of *Ccna2* and *Ccnb2* downregulation.

A detailed description of these analyses and in-depth discussion of these findings can be found within the manuscript text between **lines 302-404** and **lines 557-616**, respectively. These data are presented as part of **Figure 5**, **Figure 6** and **Supplementary Figure S7**.

CCND1 - is a regulator of G1/S phase transition - did this disappear in the null Spermatogonia?

We found that loss of *Ddx5* resulted in significant *upregulation* of *Ccnd1* mRNA; however, CCND1 protein levels were significantly *reduced* in *Ddx5*-ablated spermatogonia (**Figure 5b,d,e**). Further, *Ccnd1* mRNA was bound to DDX5 in spermatogonia (**Figure 5f**) and through analysis of nuclear-cytoplasmic mRNA ratios we demonstrated that loss of DDX5 results in abnormal nuclear retention of *Ccnd1* mRNA (**Figure 5g**). These findings indicate that DDX5 directly regulates the nuclear export of *Ccnd1* transcripts. Therefore, loss of *Ddx5* results in reduced CCND1 protein and consequent G1/S arrest in spermatogonia. These findings are presented in the aforementioned figures, described specifically in manuscript text **lines 332-343** and discussed between **lines 557-573**.

REVIEWER #3

In this study, Legrand and colleagues explored the role of an RNA helicase, Ddx5, in spermatogenesis using a variety of techniques. Based on the results, they conclude that Ddx5 plays an essential role in maintenance of the spermatogenic stem cell population via interaction with the transcription factor Plzf. At present, the role of Ddx5 and other RNA helicases in the spermatogenic lineage is undefined, thus the study addresses a gap in knowledge. Overall, the experimentation is technically sound. However, the results are not interpreted correctly, as discussed below, and should be re-assessed for accuracy rather than claiming a unique role in the stem cell compartment of the mouse male germline. Clearly, Ddx5 plays an important role in spermatogenesis, likely having a function in all germ cell subtypes and possibly Sertoli cells. This finding is unique and advances knowledge. It's unfortunate that the authors chose to over interpret the results to portray an importance in germline stem cells only rather than focus on the role of Ddx5 in spermatogenesis in general for which they have an abundance of data to support.

We thank the reviewer for their constructive comments and for acknowledging our demonstration of key roles for DDX5 during spermatogenesis. We agree with the reviewer that the role we describe for DDX5 is not solely restricted to the self-renewing stem cell population and therefore, we have adjusted our manuscript to reflect this broader interpretation of our data. In addition, we have included additional experiments and reanalysed our data to provide a more comprehensive understanding of DDX5 during spermatogenesis and present a robust model of DDX5 function in control of spermatogonial gene expression (**Figure 8**). We have addressed specific concerns as detailed below.

The authors' attestation that Plzf is essential for self-renewal for male germline stem cells is not necessarily accurate given that spermatogenesis still occurs in a good portion of seminiferous tubules of Plzf null mice. Clearly, it plays an important role but to claim an essential role is not supported by empirical evidence.

We agree with the reviewer's concern regarding use of the term "essential" to describe the role of PLZF in male germline stem cells. We have now adjusted the text to state that "PLZF is *important* for the self-renewal of undifferentiated spermatogonia" (**lines 50-51, 105-107 and 461-462**).

A major nuance of this study is that Ddx5 is clearly expressed by all spermatogenic cell types in mice as well as primates as demonstrated in Fig. 1 and the supplemental figure. Thus, to claim a unique role in undifferentiated spermatogonia (and presumably the rare stem cell component of this population) is not supported. Moreover, the expression of Ddx5 by Sertoli cells makes interpretation of the findings even more challenging.

We acknowledge the reviewer's concerns regarding our claims that DDX5 has a unique role in undifferentiated spermatogonia and within the stem cell subpopulation. We agree that the data we present supports a broader role for DDX5 during spermatogenesis, rather than within specific subpopulations of undifferentiated spermatogonia. Thus, we have modified the manuscript text to reflect our revised interpretation that DDX5 is important for maintenance and function of *spermatogonia*. However, we have restricted our interpretation to *undifferentiated spermatogonia* where we have empirical evidence to support these claims through our *in vitro* analyses. Our *in vitro* culture method specifically supports undifferentiated and not differentiating spermatogonia^{20,22,23}. The bulk of cells in these cultures resemble progenitors and a minor fraction retains transplantable stem cell activity²⁰. Therefore, extending findings from our *in vitro* analyses to germ cell subsets other than *type A undifferentiated spermatogonia* would be an inaccurate interpretation given these culture conditions. We believe that our revised manuscript accurately details the role of DDX5 in the male germline while considering the limitations and features of our experimental systems.

Specifically: 1) we have changed the title of our manuscript to “DDX5 plays essential transcriptional and post-transcriptional roles in the maintenance and function of spermatogonia”; 2) We have adjusted the abstract text to reflect our revised conclusions; 3) We have adjusted the final sentence of our introduction to state “Our data uncovers an essential role for DDX5 within undifferentiated and differentiating spermatogonia, and characterises DDX5 as a critical regulator of male fertility” (**lines 101-103**); 4) We have made numerous adjustments to the discussion text (highlighted in the revised manuscript) to reflect our more accurate interpretation of DDX5 function in the male germline.

As detailed below, we have performed additional analyses that provide quantitative evidence to support these conclusions. This includes new flow cytometry data analysing whole testis samples of control versus *Ddx5*-ablated mice, as well as quantification of germ cell populations by immunofluorescence of testis sections. As the reviewer highlights, *Ddx5* is expressed in Sertoli cells; however, as described below, our model does not result in *Ddx5* ablation within Sertoli cells and our *in vitro* analyses confirm a critical cell autonomous role for DDX5 in spermatogonia. Therefore, we are confident that our findings can be accurately interpreted in the context of spermatogonia without confounding issues driven by defects in somatic cell populations (see below). Nevertheless, investigation of the role of DDX5 in Sertoli cells may form the basis of intriguing future studies.

The authors have made several claims of Ddx5 expression levels being different among cell types that are based on subjective observations rather than quantitative data. For example, in the supplemental data, they have claimed higher expression of Ddx5 in Plzf+ spermatogonia of human testes compared to other cells but this is based on a single image of immunofluorescent staining rather than actual

measurements of Ddx5 abundance. Similarly, the claims of comparable levels of Ddx5 expression in different germ cell subtypes as presented in Fig. 1e is based solely on subjective observations not quantitative data.

We acknowledge the concerns of the reviewer regarding the lack of quantitative data to define differences in DDX5 expression level between germ cell populations. Thus, we have revised results text under the title “DDX5 associates with PLZF and is expressed throughout the male germline”, where we have removed suggestions of differential DDX5 expression that are not based on quantitative results. Specifically, we have adjusted **lines 118-120** to state: “we examined expression of DDX5 in non-human primate (marmoset) and human testis sections and observed a comparable expression pattern in spermatogenic cells to that of mouse, including within PLZF-positive cells”. We have also revised the associated **Supplementary Figure S1**, where we now show additional images for DDX5 expression in both marmoset and human seminiferous tubule cross-sections. In addition, we have adjusted our in-text description of Figure 1e to state that “we observed DDX5 expression localised to the nuclei” (**lines 127-128**) of depicted populations, rather than make conclusions on expression differences. We have also adjusted the final sentence of this results section to read “As a result, we concluded that DDX5 is expressed throughout the male germline and in supporting somatic cells” (**lines 130-131**).

Empirical evidence supporting the claim the UBC-Cre transgene used in the studies is expressed in spermatogonia only is lacking. In fact, this same group’s recent publication in Stem Cell Reports showed that the transgene is actually expressed in all spermatogonia as well as spermatocytes. Thus, using this model in the current study, Ddx5 expression was ablated in all pre-meiotic and meiotic germ cells not just spermatogonia. The phenotypic findings cannot be interpreted as a role in undifferentiated spermatogonia only as all germ cells were rendered deficient for Ddx5 following tamoxifen injection.

In our revised manuscript, we have now included a quantitative analysis of *Ddx5* ablation in spermatocytes and round spermatids, as well as a quantitative comparison of these cell numbers between control and *Ddx5*-ablated mice by immunofluorescence analysis (**Supplementary Figure S3**). In addition, we have confirmed the loss of undifferentiated and differentiating spermatogonial populations by immunofluorescence (**Supplementary Figure S2**) and performed analysis of whole testis samples up to D14 following *Ddx5*-ablation by flow cytometry (**Supplementary Figure S4**).

In our analysis of *Ddx5* ablation, we found no significant difference in the number of spermatocytes or round spermatids that were positive for DDX5 and no significant difference in their numbers between control and TAM-treated *Ddx5*^{TAM-KO} testes at D7 (**Supplementary Figure S3b,c**,

and **lines 149-154**). These findings demonstrate that in this UBCCre^{ERT2};Ddx5^{flox/flox} mouse model, tamoxifen-induced Cre-mediated gene ablation is not evident in spermatocytes and round spermatids. However, we did find that *Ddx5* ablation resulted in loss of both undifferentiated and differentiating spermatogonia (**Supplementary Figure S2** and **Supplementary Figure S4**). This result is consistent with strong activity of the UBCCre^{ERT2} transgene within the spermatogonial pool²⁰. By immunofluorescence, we found that *Ddx5* ablation resulted in the loss of both GFR α 1-positive and SOX3-positive spermatogonia (representing stem and progenitor cell-enriched populations, respectively^{24,25}) by D7 post-ablation, and very few *Ddx5*-deleted c-KIT-positive spermatogonia remained (**Supplementary Figure S2** and **lines 143-149**). Accordingly, when analysing testis cell suspensions from control and *Ddx5*-ablated mice by flow cytometry, we found that undifferentiated (PLZF⁺ c-KIT⁻), early differentiating (PLZF⁺ c-KIT⁺) and late differentiating (PLZF^{low} c-KIT⁺) spermatogonia were all significantly depleted by D5 post-ablation (**Supplementary Figure S4a,b** and **lines 157-162**). Importantly, *Ddx5* ablation resulted in loss of all spermatogonia and therefore we have adjusted our manuscript text to reflect this revised interpretation. Given that we have shown specific loss of spermatogonia without directly impacting other germ cell subsets, we have stated that “Combined, our data indicate that *Ddx5* plays specific roles in maintenance of spermatogenesis and its loss results in the rapid and profound depletion of adult spermatogonia” (**lines 179-180**). We have made appropriate adjustments throughout the discussion text to reflect these findings. We thank the reviewer for highlighting this important point that has allowed us to broaden our conclusions on the important role for DDX5 in the male germline.

The data presented in Fig. 2 can only be interpreted as a role for Ddx5 in germ cells in general because it is expressed throughout the spermatogenic lineage and the Cre transgene is not expressed specifically in undifferentiated spermatogonia. Also, the claim that Ddx5 expression was not ablated from any Sertoli cells following tamoxifen injection is not supported by experimental evidence.

As described above, we have now shown that in this model *Ddx5* ablation results in loss of essentially all spermatogonia without affecting other germ cell populations; therefore, we have interpreted our findings presented in Figure 2 as a role for *Ddx5* in all spermatogonia. In addition, we now provide quantitative evidence demonstrating a lack of *Ddx5* ablation in Sertoli cells. In **Supplementary Figure S3a**, we show that 100% of scored Sertoli cells were positive for DDX5 in both control and TAM-treated *Ddx5*^{TAM-KO} testes. There was also no difference in the numbers of Sertoli cells following *Ddx5* ablation. These findings are described in **lines 149-154**. Given these additional analyses plus detailed assessment of the function of DDX5 in cultured spermatogonia, we conclude that our data demonstrates an essential cell autonomous role for DDX5 within spermatogonia.

Data presented in Fig. 3-6 utilized cultures of spermatogonia that are a heterogeneous mix of subtypes including stem cell, progenitor, and depending on the methodology, differentiating spermatogonia. Thus, the data cannot be interpreted as Ddx5 playing a specific role in the stem cell component. Rather, the authors are encouraged to interpret the data as being relevant to all spermatogonial subtypes.

As correctly highlighted by the reviewer, the majority of cells in these cultures do not generate stable colonies upon transplantation and resemble committed progenitors^{20,26}. However, established groups often refer to spermatogonia grown under these conditions as “germline stem (GS) cells”,^{27,28}. We have previously confirmed that cells cultured under these conditions are uniformly positive for markers of stem and progenitor cells, including GFR α 1, PLZF, SALL4 and E-Cadherin and that transplantable stem cell activity is retained^{20,22}. Very few cells in growing cultures are c-KIT⁺²³. Experiments that use these cultures are therefore of relevance for both stem and progenitor cells, *i.e.* undifferentiated spermatogonia. In light of the reviewer’s suggestion, we have now interpreted our findings based on *in vitro* experiments as relevant to undifferentiated spermatogonia, rather than specifically to the stem cell compartment. Furthermore, our *in vivo* experiments demonstrate that loss of *Ddx5* results in the depletion of all spermatogonial subtypes. Overall, the combination of our *in vivo* and *in vitro* analyses supports a role for DDX5 in both undifferentiated and differentiating spermatogonia and we believe that this is now reflected in our revised manuscript. In addition to our revised title, we have made appropriate changes to all relevant sections of the abstract, introduction, results and discussion. We thank the reviewer for these important suggestions that have enabled us to improve interpretations of the data.

REFERENCES

1. Lovelace DL, Gao Z, Mutoji K, Song YC, Ruan J, Hermann BP. The regulatory repertoire of PLZF and SALL4 in undifferentiated spermatogonia. *Development* **143**, 1893-1906 (2016).
2. Caretti G, *et al.* The RNA helicases p68/p72 and the noncoding RNA SRA are coregulators of MyoD and skeletal muscle differentiation. *Developmental cell* **11**, 547-560 (2006).
3. Fuller-Pace FV. The DEAD box proteins DDX5 (p68) and DDX17 (p72): multi-tasking transcriptional regulators. *Biochim Biophys Acta* **1829**, 756-763 (2013).
4. Fuller-Pace FV, Ali S. The DEAD box RNA helicases p68 (Ddx5) and p72 (Ddx17): novel transcriptional co-regulators. *Biochemical Society transactions* **36**, 609-612 (2008).
5. Dardenne E, *et al.* RNA helicases DDX5 and DDX17 dynamically orchestrate transcription, miRNA, and splicing programs in cell differentiation. *Cell Rep* **7**, 1900-1913 (2014).
6. Li H, *et al.* RNA Helicase DDX5 Inhibits Reprogramming to Pluripotency by miRNA-Based Repression of RYBP and its PRC1-Dependent and -Independent Functions. *Cell stem cell*, (2017).
7. Zonta E, Bittencourt D, Samaan S, Germann S, Dutertre M, Auboeuf D. The RNA helicase DDX5/p68 is a key factor promoting c-fos expression at different levels from transcription to mRNA export. *Nucleic acids research* **41**, 554-564 (2013).
8. Suzuki HI, Yamagata K, Sugimoto K, Iwamoto T, Kato S, Miyazono K. Modulation of microRNA processing by p53. *Nature* **460**, 529-533 (2009).
9. Bates GJ, *et al.* The DEAD box protein p68: a novel transcriptional coactivator of the p53 tumour suppressor. *EMBO J* **24**, 543-553 (2005).
10. Wolgemuth DJ, Manterola M, Vasileva ANA. Role of cyclins in controlling progression of mammalian spermatogenesis. *The International journal of developmental biology* **57**, 159-168 (2013).
11. Katz Y, Wang ET, Airoidi EM, Burge CB. Analysis and design of RNA sequencing experiments for identifying isoform regulation. *Nature methods* **7**, 1009-1015 (2010).
12. Roset R, *et al.* The Rad50 hook domain regulates DNA damage signaling and tumorigenesis. *Genes & development* **28**, 451-462 (2014).
13. Grosso AR, *et al.* Tissue-specific splicing factor gene expression signatures. *Nucleic Acids Res* **36**, 4823-4832 (2008).
14. Matlin AJ, Clark F, Smith CW. Understanding alternative splicing: towards a cellular code. *Nat Rev Mol Cell Biol* **6**, 386-398 (2005).
15. Kar A, *et al.* RNA helicase p68 (DDX5) regulates tau exon 10 splicing by modulating a stem-loop structure at the 5' splice site. *Molecular and cellular biology* **31**, 1812-1821 (2011).

16. Honda A, Valogne Y, Bou Nader M, Brechot C, Faivre J. An intron-retaining splice variant of human cyclin A2, expressed in adult differentiated tissues, induces a G1/S cell cycle arrest in vitro. *PloS one* **7**, e39249 (2012).
17. Paronetto MP, *et al.* Alternative splicing of the cyclin D1 proto-oncogene is regulated by the RNA-binding protein Sam68. *Cancer research* **70**, 229-239 (2010).
18. Sun Q, Zhang F, Wafa K, Baptist T, Pasumarthi KB. A splice variant of cyclin D2 regulates cardiomyocyte cell cycle through a novel protein aggregation pathway. *Journal of cell science* **122**, 1563-1573 (2009).
19. Jensen ED, *et al.* p68 (Ddx5) interacts with Runx2 and regulates osteoblast differentiation. *J Cell Biochem* **103**, 1438-1451 (2008).
20. Chan AL, *et al.* Germline Stem Cell Activity Is Sustained by SALL4-Dependent Silencing of Distinct Tumor Suppressor Genes. *Stem Cell Reports* **9**, 956-971 (2017).
21. Hobbs RM, *et al.* Functional antagonism between Sall4 and Plzf defines germline progenitors. *Cell stem cell* **10**, 284-298 (2012).
22. La HM, *et al.* GILZ-dependent modulation of mTORC1 regulates spermatogonial maintenance. *Development* **145**, (2018).
23. La HM, *et al.* Identification of dynamic undifferentiated cell states within the male germline. *Nature communications* **9**, 2819 (2018).
24. Nakagawa T, Sharma M, Nabeshima Y, Braun RE, Yoshida S. Functional hierarchy and reversibility within the murine spermatogenic stem cell compartment. *Science* **328**, 62-67 (2010).
25. Suzuki H, *et al.* SOHLH1 and SOHLH2 coordinate spermatogonial differentiation. *Developmental biology* **361**, 301-312 (2012).
26. Nagano MC, Yeh JR. The identity and fate decision control of spermatogonial stem cells: where is the point of no return? *Current topics in developmental biology* **102**, 61-95 (2013).
27. Kanatsu-Shinohara M, *et al.* Long-term proliferation in culture and germline transmission of mouse male germline stem cells. *Biol Reprod* **69**, 612-616 (2003).
28. Kanatsu-Shinohara M, Shinohara T. Spermatogonial stem cell self-renewal and development. *Annual review of cell and developmental biology* **29**, 163-187 (2013).

Reviewers' comments:

Reviewer #3 (Remarks to the Author):

The authors have made substantial revisions to the manuscript including altering the interpretations to reflect the more general role of DDX5 in spermatogonia. They have also added a significant amount of new data exploring the mechanism of action. My major concerns from the original version have been appropriately addressed.

Reviewer #4 (Remarks to the Author):

As far as I can see, I feel the authors have responded in details to all original comments made by the referees, and they have significantly improved their manuscript. I have however a few further questions and criticisms on some specific points, which need to be addressed prior to publication.

In figure 3c, I agree that the global level of DDX17 is not affected by Ddx5 ablation in spermatogonia, yet the ratio between p72 and p82 isoforms seems to be modified. Please comment.

The authors sometimes draw overstated conclusions from separated observations. For example, lanes 395-396, CCND1 level is decreased by Ilf3 ablation, but no evidence is shown that this is due to the same export defect as in DDX5 KO cells.

The title in lane 406 is also overstated, as it seems to claim that several genes are regulated by both PLZF and DDX5, while this is demonstrated for only 1 gene. Besides, gene expression defects induced by DDX5 ablation are only shown through the RNA-seq results, which cannot be considered as a bona fide demonstration, as many false positives results can arise from the bioinformatic analysis. The authors should validate their in silico predictions by RT-qPCR. This is especially true for figures 3e and 7g, less crucial for figure 4g.

Negative controls are also missing in some analyses, for example to show the effect of DDX5 on expression of genes not bound by DDX5 or bound only by PLZF (figure7), or for RIP experiments (figures 4e, 5f, 6c).

Lane 629, the authors cannot speak about a physical interaction as this can be indirect, for example mediated by RNA and/or DNA.

Lane 675, the proposal made by the authors is not fully supported by their results, as the effect of both factors on Ilf3 expression was shown in different contexts, and no combined ablation or silencing of DDX5/PLZF was tested. As this was also observed for only 1 gene, I would recommend the authors to be more cautious.

In lanes 1484-85, for quantifications of western-blot, 2 different statistical tests are used for the different proteins. Please explain or correct.

Cyril F. Bourgeois

RESPONSE TO THE REVIEWERS

We would like to thank the reviewers again for their positive comments regarding our revised manuscript and for bringing their additional concerns to our attention. We have addressed these remaining points through the inclusion of additional data and by making modifications to the text as detailed below. In particular, we have included RT-qPCR data that supports our RNA-seq findings (**Supplementary Figures 5a, 5b, 6d, 7a, 9d**), a more detailed analysis of mRNA export upon *Ilf3* ablation (**Supplementary Figure 8**), provided additional control data (**Supplementary Figure 6a-c**) and have revised the manuscript text accordingly to address specific points raised by the additional reviewer. We believe these revisions have improved our manuscript further and continue to support a critical role for DDX5 in the maintenance of spermatogonia and male fertility. A point-by-point response to reviewers' comments is included below. We have highlighted all changes to the manuscript text in red font.

REVIEWER #3

The authors have made substantial revisions to the manuscript including altering the interpretations to reflect the more general role of DDX5 in spermatogonia. They have also added a significant amount of new data exploring the mechanism of action. My major concerns from the original version have been appropriately addressed.

We take this opportunity to thank the reviewer for their positive comments and for acknowledging the work we have done to explore the mechanisms underlying DDX5 function in spermatogonia. We are pleased to have appropriately addressed the reviewer's concerns.

REVIEWER #4

As far as I can see, I feel the authors have responded in details to all original comments made by the referees, and they have significantly improved their manuscript. I have however a few further questions and criticisms on some specific points, which need to be addressed prior to publication.

We would like to thank the reviewer for their time in assessing our revised manuscript, and for acknowledging our detailed response and significant improvements related to our original submission. We appreciate the reviewer's concerns given their expertise on this topic and have addressed these through the addition of RT-qPCR validation, additional mRNA export analysis, additional controls and modifications to the manuscript text. Responses to specific points are detailed below.

In figure 3c, I agree that the global level of DDX17 is not affected by *Ddx5* ablation in spermatogonia, yet the ratio between p72 and p82 isoforms seems to be modified. Please comment.

We have now quantified the ratio of p72/p82 isoforms of DDX17 in control and *Ddx5*-ablated spermatogonia (**Fig. A**). We did not find any statistically significant difference in p72/p82 ratio between conditions following analysis of four independent biological replicates per condition ($p=0.3745$). Although *Ddx17* appears to be modestly upregulated in *Ddx5*-ablated spermatogonia according to RNA-seq analysis, we were unable to confirm differential expression between conditions when assessing *Ddx17* by RT-qPCR (**Supplementary Figure 5a** and **lines 216-218**). Nevertheless, these data continue to support our initial findings that “levels of DDX17 protein were unchanged” upon *Ddx5* ablation in spermatogonia and conclusion that “DDX17 cannot compensate for DDX5 function in undifferentiated spermatogonia” (**Figure 3b-d** and **lines 218-221**). Given our additional quantification, we conclude that there is no difference in p72/p82 ratio upon loss of *Ddx5* in spermatogonia.

The authors sometimes draw overstated conclusions from separated observations. For example, lanes 395-396, *CCND1* level is decreased by *Ilf3* ablation, but no evidence is shown that this is due to the same export defect as in *DDX5* KO cells.

We appreciate the reviewer bringing this to our attention and we apologise for this overstatement. We have now performed an additional analysis of *Ccnd1* mRNA export upon *Ilf3* ablation by assessing nuclear-cytoplasmic ratio in the same manner as performed for DDX5-related analyses. We were unable to confirm consistent changes in *Ccnd1* nuclear-cytoplasmic ratio at D2 following CRISPR/Cas9-mediated *Ilf3* ablation in undifferentiated spermatogonia (**Supplementary Figure 8** and **lines 408-411**), suggesting ILF3 regulates *Ccnd1* expression through alternate post-transcriptional mechanisms. We have taken care to state this explicitly within the results text (**lines**

408-411) and within the discussion text (**lines 630-632**) to avoid overstatement and for clarity to the reader.

Although our data indicates that ILF3 does not regulate *Ccnd1* through mRNA export mechanisms, we maintain that post-transcriptional regulation of this gene is impacted upon *Ilf3* ablation given unchanged *Ccnd1* mRNA yet reduced CCND1 protein (**Figure 6f,g**). Previous studies have found that ILF3 is present in a complex with IGF2BP3 (also known as IMP-3) in cancer cell lines and regulates *Ccnd1* and *Ccnd3* through post-transcriptional mechanisms¹. IGF2BP3 has been shown to protect *Ccnd1* and *Ccnd3* transcripts from miRNA-mediated translational repression that occurs as a result of AGO2 recruitment¹. In addition, ILF3 is known to be found in complex with ILF2 in HeLa cells and together are suggested to regulate cell proliferation². Thus, we suggest that defects in post-transcriptional regulation of *Ccnd1* upon loss of *Ilf3* may occur as an indirect consequence of ILF3/IGF2BP3/ILF2 complex disruption. We have provided a detailed discussion regarding these findings in **lines 607-634** of the manuscript text. In addition, we have now included text within **lines 630-632** of the discussion to clarify that DDX5 and ILF3 function through distinct mechanisms in this context.

The title in lane 406 is also overstated, as it seems to claim that several genes are regulated by both PLZF and DDX5, while this is demonstrated for only 1 gene.

We agree with the reviewer's concerns regarding the title of this results section; therefore, we have now changed this title to read "*DDX5 and PLZF share common promoter region binding sites and co-regulate Ilf3 expression*" (**line 421**). We believe this title more accurately reflects the findings described within this results section.

Besides, gene expression defects induced by DDX5 ablation are only shown through the RNA-seq results, which cannot be considered as a bona fide demonstration, as many false positives results can arise from the bioinformatic analysis. The authors should validate their in silico predictions by RT-qPCR. This is especially true for figures 3e and 7g, less crucial for figure 4g.

We have now added validation by RT-qPCR of gene expression data obtained through RNA-seq analysis. We have provided RT-qPCR confirming the differential expression of key genes that we have described in the manuscript text for Figures 3e, 7g and 4g, as suggested by the reviewer (**Supplementary Figures 5b; 6d; and 9d, and lines 224-226; 235-236; 301-302; and 452**). In addition, we have also included RT-qPCR validation of key spermatogonial genes that we refer to in manuscript text line 210 (**Supplementary Figure 5a and lines 211-212**, related to Figure 3d).

Furthermore, we felt it important to include RT-qPCR validation of key cell cycle-related genes that we have focused on in our manuscript (**Supplementary Figure 7a**, related to Figure 5b).

As the reviewer will see, all RT-qPCR results that we have provided are in agreement with key conclusions that we have derived from RNA-seq analyses, besides some p53-related genes. Nevertheless, we have provided clear evidence to demonstrate that loss of spermatogonia following *Ddx5* ablation is a result of apoptosis (**Figure 3e-g** and **lines 222-237**). We are confident that this occurs through a p53-mediated response, as we demonstrate strong induction of the p53-target genes *Cdkn1a* (*p21*)^{3,4}, *Puma*⁵ and *Gadd45b*⁶. Moreover, disruption of pre-mRNA splicing machinery has been shown previously to result in p53 activation⁷ (discussed in **lines 309-310** and **567-571**).

We appreciate the reviewer's concerns regarding RNA-sequencing analyses and agree that false positives (and false negatives) can occur. In our study, we are confident in the accuracy of our analysis due to the robust methods and strict criteria we have applied in order to avoid these outcomes. For our experiment, we used four independent biological replicates per condition with a sequencing depth of >30 million reads per sample. Our analysis was performed using the trimmed mean of M-values (TMM) method⁸ for normalisation and false discovery rate (FDR) method⁹ for multiple test correction under conservative conditions. These are well-established and reliable techniques that are routinely used for analysis of RNA-sequencing datasets. Given the stringent conditions applied in our analysis and number of biological replicates used, coupled with validation of key genes by RT-qPCR, we believe that we have provided a reliable RNA-sequencing dataset in this study.

Negative controls are also missing in some analyses, for example to show the effect of DDX5 on expression of genes not bound by DDX5 or bound only by PLZF (figure7), or for RIP experiments (figures 4e, 5f, 6c).

We apologise for the omission of some details regarding negative controls. Regarding the effect of DDX5 loss on expression of genes *not bound by DDX5* or *bound only by PLZF*:

- Of the genes not bound by DDX5 that were detected in our RNA-seq analysis (n=13421 genes), 3419 were downregulated (25.5%), 3466 were upregulated (25.8%) and the remainder were unaffected (48.7%) upon loss of *Ddx5*. Of the 4160 genes identified by Lovelace *et al.*¹⁰ that were bound by PLZF only, 827 were downregulated (19.9%), 609 were upregulated (14.6%) and the remainder were unaffected (65.5%) upon loss of *Ddx5*. These findings are in accordance with the proportions of up- and down-regulated genes upon *Ddx5* loss that are seen for DDX5-PLZF shared targets (described in **lines 442-445**). Although we have described DDX5 as a regulator of gene expression through post-transcriptional mechanisms and proposed a transcriptional role in spermatogonia, it is evident that the majority of genes

dysregulated upon *Ddx5* loss in SPCs occurs as a result of indirect effects. We have now added discussion of this point in the manuscript text between **lines 668-672**, as part of our broader discussion of the role of DDX5 in transcriptional regulation (**lines 635-682**).

- We have now included *Pou5f1* as an example of a negative control gene that is described previously as bound to PLZF¹⁰, but is *not* bound by DDX5 *or* differentially expressed upon *Ddx5* ablation. We have also included *Sall4* as a gene that is not bound by DDX5 (**Supplementary Table 8**) or PLZF¹⁰ and is not significantly affected by *Ddx5* ablation. We have included these results in **line 214** and **Supplementary Figure 5a**, in addition to the RNA-seq data found in **Supplementary Table 2**.

Regarding negative controls for RNA-IP experiments:

- We now show that the housekeeping gene *Hprt* is not bound to DDX5 as a negative control for DDX5 RNA-IP. We have included this as part of **Supplementary Figure 6c** and described in **lines 286-287**. We apologise for not including this in our revised submission and thank the reviewer for bringing this to our attention. Furthermore, we have also highlighted that *Ccnal* is *not* bound by ILF3 (**Figure 6c**) and thus serves as a negative control for this RIP experiment (**lines 391-392**).
- In addition, we have now included data to demonstrate that some genes identified in our splicing analysis as being differentially spliced upon *Ddx5* ablation in SPCs were not detectably bound to DDX5, suggesting indirect regulation of their splicing by DDX5. These data are included in **Supplementary Figure 6a-c** and described in **lines 284-286**.

These additional data continue to support our conclusions regarding the role of DDX5 as a selective regulator of gene expression through both transcriptional and post-transcriptional mechanisms in SPCs.

Lane 629, the authors cannot speak about a physical interaction as this can be indirect, for example mediated by RNA and/or DNA.

We agree with the reviewer regarding use of the term “*physical interaction*” to describe DDX5 association with PLZF. Therefore, we have now modified the wording to describe a “*robust interaction*” of PLZF-DDX5 to avoid the possibility of overinterpretation (**line 647**). We believe this accurately reflects our findings, given our reciprocal co-IP/WB and co-IP/MS findings.

Lane 675, the proposal made by the authors is not fully supported by their results, as the effect of both factors on Ilf3 expression was shown in different contexts, and no combined ablation or silencing of DDX5/PLZF was tested. As this was also observed for only 1 gene, I would recommend the authors to be more cautious.

We appreciate the reviewer's concerns regarding our proposed mechanism of DDX5/PLZF co-regulation of *Ilf3*. Therefore, we have now modified the discussion text where the reviewer has suggested. In **lines 696-698**, we now state: "*Further, we propose that DDX5 and PLZF co-regulate transcription of Ilf3, thereby maintaining cell cycle progression through downstream effects on cyclin mRNA transcripts*". We believe this is now a more accurate interpretation of our results given that recruitment of DDX5 to the *Ilf3* promoter is not dependent on PLZF, yet individual ablation of DDX5 and PLZF results in *Ilf3* downregulation (**Figure 7g,h** and **Supplementary Figure 9d**). Given the severity of *Ddx5*-knockout phenotype in SPCs, we predict that combined ablation or silencing of *Ddx5* and *Plzf* would not yield additional mechanistic insight. Moreover, the phenotypes of *Ddx5*- and *Plzf*-knockout¹¹ in SPCs are dissimilar and we believe that combined PLZF/DDX5 ablation is beyond the scope of our current study. However, in light of increasing interest in this area, we have included additional discussion where we suggest areas for further investigation that can form the basis of future studies regarding transcriptional regulation by DDX5 (**lines 698-701**).

Overall, we agree with the reviewer's suggestion that a more cautious interpretation of these findings is required and we are confident that this is now reflected in our revised manuscript.

In lanes 1484-85, for quantifications of western-blot, 2 different statistical tests are used for the different proteins. Please explain or correct.

For statistical analysis of our data, where the data were determined to be normally distributed according to the Shapiro-Wilk normality test¹² (where $\alpha=0.05$), we used two-tailed unpaired *t*-tests as a parametric statistical test. However, where the data were determined to be *not* normally distributed ($P<0.05$, Shapiro-Wilk normality test where $\alpha=0.05$), we used the non-parametric Mann-Whitney *U* test, as this does not assume a normal distribution for testing of the null hypothesis and when comparing independent samples¹³. A description of these methods is included in the manuscript text between **lines 1017-1022**.

For quantification of western blots in Figure 5e, the data for CCND1 were found to be not normally distributed ($P=0.0477$, $W=0.76$, $\alpha=0.05$); therefore, the Mann-Whitney *U* test was performed to test the null hypothesis. Data for CCND2 and CCNA2 were determined to be normally distributed ($P=0.7794$, $W=0.9601$, $\alpha=0.05$); therefore, two-tailed unpaired *t*-tests were used to test the null hypothesis.

REFERENCES

1. Deforz E, Vargas TR, Kropp J, Vandamme M, Pinna G, Polesskaya A. IMP-3 protects the mRNAs of cyclins D1 and D3 from GW182/AGO2-dependent translational repression. *Int J Oncol* **49**, 2578-2588 (2016).
2. Guan D, *et al.* Nuclear factor 45 (NF45) is a regulatory subunit of complexes with NF90/110 involved in mitotic control. *Molecular and cellular biology* **28**, 4629-4641 (2008).
3. Aubrey BJ, Kelly GL, Janic A, Herold MJ, Strasser A. How does p53 induce apoptosis and how does this relate to p53-mediated tumour suppression? *Cell Death Differ* **25**, 104-113 (2018).
4. el-Deiry WS, *et al.* WAF1, a potential mediator of p53 tumor suppression. *Cell* **75**, 817-825 (1993).
5. Nakano K, Vousden KH. PUMA, a novel proapoptotic gene, is induced by p53. *Mol Cell* **7**, 683-694 (2001).
6. Kim YA, *et al.* Gadd45beta is transcriptionally activated by p53 via p38alpha-mediated phosphorylation during myocardial ischemic injury. *J Mol Med (Berl)* **91**, 1303-1313 (2013).
7. Allende-Vega N, Dayal S, Agarwala U, Sparks A, Bourdon JC, Saville MK. p53 is activated in response to disruption of the pre-mRNA splicing machinery. *Oncogene* **32**, 1-14 (2013).
8. Robinson MD, Oshlack A. A scaling normalization method for differential expression analysis of RNA-seq data. *Genome Biol* **11**, R25 (2010).
9. Benjamini Y, Hochberg Y. Controlling the False Discovery Rate - a Practical and Powerful Approach to Multiple Testing. *J R Stat Soc B* **57**, 289-300 (1995).
10. Lovelace DL, Gao Z, Mutoji K, Song YC, Ruan J, Hermann BP. The regulatory repertoire of PLZF and SALL4 in undifferentiated spermatogonia. *Development* **143**, 1893-1906 (2016).
11. Costoya JA, *et al.* Essential role of Plzf in maintenance of spermatogonial stem cells. *Nature genetics* **36**, 653-659 (2004).
12. Shapiro SS, Wilk MB. An Analysis of Variance Test for Normality (Complete Samples). *Biometrika* **52**, 591-& (1965).
13. Fay MP, Proschan MA. Wilcoxon-Mann-Whitney or t-test? On assumptions for hypothesis tests and multiple interpretations of decision rules. *Stat Surv* **4**, 1-39 (2010).

REVIEWERS' COMMENTS:

Reviewer #4 (Remarks to the Author):

I appreciate the effort made by the authors to respond to all my comments. Some have been correctly addressed, but I still have concerns regarding 3 specific points. I want to stress that it does not question the overall quality of the work and the conclusions of the manuscript, that deserves to be published, but that will help the reader in fully appreciating the many details of this work.

1. I am not yet utterly convinced about the lack of switch between the p82 and p72 isoforms in Figure 3c. In their rebuttal letter, the authors provided a quantification of western-blot from 4 biological replicates, showing that, as I had noticed in the figure, there is indeed a small increase in the p72/p82 ratio, yet the difference did not appear as significant. In that case, why did the authors not show a blot that would appear less misleading, instead of the picture they maintained in the figure ?

2. My point regarding the validation of RNA-seq data: I appreciate that the authors have validated or tried to validate some of the gene expression changes predicted by their RNA-seq data, and that they went beyond my initial recommendations. However, I think that showing (in several figures) genes that are predicted to be altered with Log2 fold change < 0.5 does not make much sense if they are not validated by RT-qPCR.

For example, the authors say in lanes 448-452: "we determined changes in expression of these genes according to our RNA-seq data and found significant downregulation of Amdhd2 (1.53-fold), Ilf3 (1.44-fold) and Arhgap12 (1.31-fold), and upregulation of Tmem55b (1.13-fold, as described earlier), 4933434E20Rik (1.16-fold) and Zbtb25 (1.24-fold) upon Ddx5 loss (Fig. 7d,g and Supplementary Figure 9d)." However, in Supplementary Figure 9d, they show only the RT-qPCR validation of Amdhd2 and Ilf3 that are the 2 most deregulated genes. What about the others? I would anticipate that considering the very low predicted effects, those genes will be very difficult to validate by RT-qPCR.

Despite a nice RNA-seq design (4 replicates, fairly deep sequencing) and a good analysis pipeline, I am wondering why the authors do not consider those predictions with more caution (maybe by defining a different threshold). In fact, in the only example where the authors tried quasi-systematically to validate the 7 RNA-seq predictions by RT-qPCR (Fig. 3e and Supp. Fig. 5b), they validated only 2 changes out of 5 (the last 2 genes were not tested or were not shown). They even failed to validate one gene, Noxa, which had a predicted Log2(fold change) of about 1. To me, this is a clear evidence that RNA-seq predictions with low fold change cannot be fully trusted, and I would suggest the authors to remove such genes from their figures or to validate them.

3. Finally, my last question about the legend of Figure 5e (related to the Western-blot shown in Figure 5d) unexpectedly gave rise to a long dissertation about the normality of the data distribution for the 3 quantified proteins. However, calculating the normality of the distribution on such a small series of samples (n=4) is meaningless, and the authors should opt for the same test for all 3 proteins. In such a situation (a measurement of an unsaturated signal), assuming the possibility that the distribution is gaussian is perfectly acceptable, and thus a parametric test can be used.

Cyril F. Bourgeois

RESPONSE TO THE REVIEWERS

We would like to thank the reviewer for their positive and constructive comments. We have now addressed all remaining points through the inclusion of additional data and modifications to the text as detailed below. Specifically, we have now included quantification of p72/p82 DDX17 isoforms as part of the Supplementary Information (**Supplementary Figure 5**), included additional results for RNA-sequencing validation by RT-qPCR (**Supplementary Figures 10d** and **10e**), and removed RNA-sequencing data that were of low fold change and not validated by RT-qPCR. All changes to the manuscript text are identifiable through tracked changes in Microsoft Word.

REVIEWER #4

I appreciate the effort made by the authors to respond to all my comments. Some have been correctly addressed, but I still have concerns regarding 3 specific points. I want to stress that it does not question the overall quality of the work and the conclusions of the manuscript, that deserves to be published, but that will help the reader in fully appreciating the many details of this work.

We thank the reviewer for acknowledging the quality of our work and we appreciate their comments that have allowed us to improve our manuscript. We have addressed remaining concerns through inclusion of additional data, further RT-qPCR validation of RNA-sequencing analysis, and modifications to the manuscript text. Responses to specific points are detailed below.

1. I am not yet utterly convinced about the lack of switch between the p82 and p72 isoforms in Figure 3c. In their rebuttal letter, the authors provided a quantification of western-blot from 4 biological replicates, showing that, as I had noticed in the figure, there is indeed a small increase in the p72/p82 ratio, yet the difference did not appear as significant. In that case, why did the authors not show a blot that would appear less misleading, instead of the picture they maintained in the figure?

We appreciate the reviewer's concerns regarding p72/p82 isoforms of DDX17. Although the quantification appears to show a small increase in p72/p82 ratio upon loss of *Ddx5* in spermatogonia, this difference was not found to be statistically significant. We believe the western blot presented in Figure 3c is representative of this *slight but non-significant* increase that the reviewer highlights. Therefore, in order to improve clarity to the reader and allow full interpretation regarding this finding, we have included additional text within the results that highlight these data (**lines 250-252**) and now provide the quantification figure (originally supplied as Figure A in our previous Response to Reviewers) within the Supplementary Information (**Supplementary Figure 5**).

2. My point regarding the validation of RNA-seq data: I appreciate that the authors have validated or tried to validate some of the gene expression changes predicted by their RNA-seq data, and that they went beyond my initial recommendations. However, I think that showing (in several figures) genes that are predicted to be altered with Log₂ fold change < 0.5 does not make much sense if they are not validated by RT-qPCR.

For example, the authors say in lanes 448-452: "we determined changes in expression of these genes according to our RNA-seq data and found significant downregulation of *Amdhd2* (1.53-fold), *Ilf3* (1.44-fold) and *Arhgap12* (1.31-fold), and upregulation of *Tmem55b* (1.13-fold, as described earlier), *4933434E20Rik* (1.16-fold) and *Zbtb25* (1.24-fold) upon *Ddx5* loss (Fig. 7d,g and Supplementary Figure 9d)." However, in Supplementary Figure 9d, they show only the RT-qPCR validation of *Amdhd2* and *Ilf3* that are the 2 most deregulated genes. What about the others? I would anticipate that considering the very low predicted effects, those genes will be very difficult to validate by RT-qPCR.

Despite a nice RNA-seq design (4 replicates, fairly deep sequencing) and a good analysis pipeline, I am wondering why the authors do not consider those predictions with more caution (maybe by defining a different threshold). In fact, in the only example where the authors tried quasi-systematically to validate the 7 RNA-seq predictions by RT-qPCR (Fig. 3e and Supp. Fig. 5b), they validated only 2 changes out of 5 (the last 2 genes were not tested or were not shown). They even failed to validate one gene, *Noxa*, which had a predicted Log₂(fold change) of about 1. To me, this is a clear evidence that RNA-seq predictions with low fold change cannot be fully trusted, and I would suggest the authors to remove such genes from their figures or to validate them.

We understand the reviewer's concerns regarding the presentation of RNA-seq data with low fold change. Therefore, we have now included additional data within **Supplementary Figure 10d** and **10e** showing further validation of RNA-seq findings by RT-qPCR. In addition, we have highlighted these findings within the manuscript text (**lines 622-623** and **639-640**). We have also removed genes identified by RNA-seq that were not assessed by RT-qPCR in **Figure 3e** and have modified the manuscript text accordingly (**line 293**).

We have chosen not to completely remove RNA-seq findings that were tested by RT-qPCR but *not* validated using this method. While RT-qPCR has proven to be a robust technique for analysis of gene expression, like RNA-seq, it is not without limitations. Factors such as PCR reaction efficiency, choice of housekeeping gene, total background RNA, and target transcript abundance can all impact RT-qPCR results¹. While we have taken the necessary measures to eliminate variability due to these factors, they must be considered when assessing RT-qPCR data in any study. Indeed, RNA-sequencing is dependent on bioinformatics techniques such as mapping, alignment and normalisation that each have nuances depending on the algorithm used; however, RNA-seq is a genome-wide, high

throughput technique that produces highly reproducible results capable of detecting subtle changes in gene expression, even amongst genes that are expressed at low levels within a sample². We have only selected robust and widely-used techniques for our data analysis pipeline³⁻⁷. Despite being unable to validate some differentially expressed genes identified by RNA-seq, changes in others with low fold changes such as *Ccna2* (1.15-fold downregulated) were successfully confirmed using RT-qPCR (Supplementary Figure 7a). Thus, we still include genes from the RNA-seq data with lower fold-change values in our analysis (providing they meet the standard FDR<0.05 cut-off) to enable readers to have a complete overview of transcriptional consequences of *Ddx5* loss in spermatogonia. However, we have taken care to explicitly state instances where RT-qPCR was unable to detect these changes, to avoid potential for misinterpretation by the reader.

Overall, we continue to agree with the reviewer's point that RNA-seq data should be interpreted with caution where only small differences are detected. We believe that the changes we have made to figures and manuscript text now appropriately describe our findings, while allowing further interpretation by the reader at their discretion.

3. Finally, my last question about the legend of Figure 5e (related to the Western-blot shown in Figure 5d) unexpectedly gave rise to a long dissertation about the normality of the data distribution for the 3 quantified proteins. However, calculating the normality of the distribution on such a small series of samples (n=4) is meaningless, and the authors should opt for the same test for all 3 proteins. In such a situation (a measurement of an unsaturated signal), assuming the possibility that the distribution is Gaussian is perfectly acceptable, and thus a parametric test can be used.

We thank the reviewer for their advice concerning the use of statistical tests in Figure 5e. As suggested, we have now assumed a Gaussian distribution for all data presented in Figure 5e and have performed two-tailed unpaired t-tests. Where previous statistical analysis for CCND1 using the Mann-Whitney U test returned a P-value of 0.0286, re-analysis using a two-tailed unpaired t-test returned a lower P-value of 0.0021. Thus, we have modified **Figure 5e** to reflect these revised analyses and adjusted the legend of Figure 5 (**line 1887-1888**).

REFERENCES

1. Levesque-Sergerie JP, Duquette M, Thibault C, Delbecchi L, Bissonnette N. Detection limits of several commercial reverse transcriptase enzymes: impact on the low- and high-abundance transcript levels assessed by quantitative RT-PCR. *BMC Mol Biol* **8**, 93 (2007).
2. Wang Z, Gerstein M, Snyder M. RNA-Seq: a revolutionary tool for transcriptomics. *Nat Rev Genet* **10**, 57-63 (2009).

3. Bolger AM, Lohse M, Usadel B. Trimmomatic: a flexible trimmer for Illumina sequence data. *Bioinformatics* **30**, 2114-2120 (2014).
4. Dobin A, *et al.* STAR: ultrafast universal RNA-seq aligner. *Bioinformatics* **29**, 15-21 (2013).
5. Law CW, Chen Y, Shi W, Smyth GK. voom: Precision weights unlock linear model analysis tools for RNA-seq read counts. *Genome Biol* **15**, R29 (2014).
6. Liao Y, Smyth GK, Shi W. featureCounts: an efficient general purpose program for assigning sequence reads to genomic features. *Bioinformatics* **30**, 923-930 (2014).
7. Ritchie ME, *et al.* limma powers differential expression analyses for RNA-sequencing and microarray studies. *Nucleic Acids Res* **43**, e47 (2015).